# Minute amounts of helicase-deficient truncated RECQL4 are sufficient for DNA replication

Paula Armina V Buco [ID] [1,2,3,10], Wilson Castillo-Tandazo [2,3,10], Alistair M Chalk [ID] [2,3], Courtney Pilcher [ID] [2,4], Jessica K Holien [2,4], Jörg Heierhorst [ID] [2,3], Tiong Y Tan [5,6], Amnon Koren [ID] [7], Monique F Smeets [ID] [2,11] & Carl R Walkley [ID] [1,2,3,8,9,11 ✉]

## Abstract

RECQL4, a RecQ family helicase, is essential for DNA replication and genome stability. Mutations in RECQL4 cause severe human disorders yet we do not fully understand its functions, particularly regarding ATP-dependent helicase activity. To understand RECQL4's functions further, we performed a genome-wide forward genetic screen using a murine model harbouring patient-like RECQL4 mutations. We identify KLHDC3, a substrate-binding subunit of the Cullin-RING ligase E3 complex, loss as the most significant rescue allele. KLHDC3 loss restores proliferation and replication in RECQL4-deficient cells by stabilizing trace levels of a truncated RECQL4 fragment containing the N-terminal 480 amino acids, lacking the helicase and C-terminal regions. This RECQL4 fragment forms after Cre-mediated recombination of the *Recql4^fl* allele and contains a neo-degron sequence specific for KLHDC3. Although this mechanism does not apply to human mutations, it demonstrates that minimal RECQL4 levels, without any ATPase domain/activity, are sufficient to support DNA replication. This demonstrates that RECQL4 is an essential and non-redundant regulator of DNA replication and cell viability and that this activity does not require its ATP-dependent helicase activity.

Keywords Recql4; Rothmund-Thomson Syndrome; RecQ; DNA Replication
Subject Category DNA Replication, Recombination & Repair

## Introduction

The orderly and accurate duplication of DNA during each cell division cycle is essential for development and homeostasis of all organisms. Despite the fundamental importance of DNA replication, there are significant knowledge gaps about the specific roles of numerous replication proteins. One such protein is RecQ like helicase 4 (RECQL4). RECQL4 is essential for DNA replication across multicellular organisms from Drosophila to mammals (Chu and Hickson, 2009). Although RECQL4 was long considered the mammalian homologue of the essential yeast DNA replication factor Sld2, it has been proposed that a structurally unrelated protein DONSON has evolved to carry out the Sld2-like function during mammalian DNA replication (Cvetkovic et al, 2023; Evrin et al, 2023; Hashimoto et al, 2023; Kingsley et al, 2023; Lim et al, 2023). As a result, the role and function of RECQL4 in normal DNA replication remains a major knowledge gap in our understanding of this fundamental cellular process. This knowledge will be directly relevant to understanding human disease, as mutations in RECQL4, like those of the other mammalian RecQ helicases BLM, WRN and RECQL5, are associated with human disease (Chu and Hickson, 2009; Hickson, 2003; Lu and Davis, 2021).

RECQL4 is a member of the RecQ family of helicases with genome integrity functions, similar to the enzymes mutated in Bloom and Werner syndromes (Hickson, 2003; Kitao et al, 1999a; Kitao et al, 1999b; Wang et al, 2003; Wang et al, 2001; Wang et al, 2002). Bi-allelic compound heterozygous mutations in *RECQL4* have been reported in three rare autosomal recessive human genetic syndromes: RAPADILINO Syndrome (OMIM #266280), Baller-Gerold Syndrome (BGS, OMIM #218600) and Rothmund-Thomson Syndrome (RTS, OMIM #268400). A significant body of work has reported that RECQL4 is required for DNA replication (Sangrithi et al, 2005) and is involved in various DNA damage repair pathways (Fielden et al, 2025; Jin et al, 2008; Lu et al, 2017; Petkovic et al, 2005; Singh et al, 2010; Thakur et al, 2025). Mouse models that lack RECQL4 protein expression led to early-to-mid-gestational lethality (Ichikawa et al, 2002; Lu et al, 2015; Smeets et al, 2014). In contrast, mice with a targeted biochemical mutation that specifically abolished ATP-dependent helicase activity (murine p.K525A, homologous to human p.K508A) were born at Mendelian frequency, fertile, not spontaneously cancer prone and had a normal lifespan under standard housing conditions (Castillo-

[1]Centre for Innate Immunity and Infection Diseases, Hudson Institute of Medical Research, Clayton, Victoria 3168, Australia. [2]St Vincent's Institute of Medical Research, Fitzroy, Victoria 3065, Australia. [3]Department of Medicine, St Vincent's Hospital, University of Melbourne, Fitzroy, Victoria 3065, Australia. [4]School of Science, STEM College, RMIT University, Melbourne, Victoria 3083, Australia. [5]Victorian Clinical Genetics Services, Murdoch Children's Research Institute, Melbourne, Victoria 3052, Australia. [6]Department of Paediatrics, University of Melbourne, Melbourne, Victoria 3052, Australia. [7]Department of Molecular and Cellular Biology, Roswell Park Comprehensive Cancer Center, Buffalo, NY 14263, USA. [8]Department of Molecular and Translational Sciences, Monash University, Clayton, Victoria 3168, Australia. [9]Drug Discovery Biology, Monash Institute of Pharmaceutical Sciences, Faculty of Pharmacy and Pharmaceutical Sciences, Monash University, Parkville, Victoria 3052, Australia. [10]These authors contributed equally: Paula Armina V Buco, Wilson Castillo-Tandazo. [11]These authors contributed equally as senior authors: Monique F Smeets, Carl R Walkley. ✉E-mail: carl.walkley@hudson.org.au

Tandazo et al, 2019). This result demonstrated that the ATP-dependent helicase activity of RECQL4 was not essential in vivo for DNA replication or murine homeostasis, consistent with analysis in human cell line models (Padayachy et al, 2024).

In contrast to these findings that RECQL4's role in DNA replication is independent of its ATPase activity, a recent single-molecule analysis in reconstituted Xenopus egg extracts proposed that the helicase activity of RECQL4 was required to evict DONSON from the Cdc45/Mcm2-7/GINS (CMG) complex to allow DNA replication (Terui et al, 2024). The authors of this recent study noted that while they considered that they had successfully depleted endogenous xRECQL4 from the extracts, the extract still retained some origin firing activity potentially indicative of trace amounts of RECQL4 remaining (Terui et al, 2024). At present it is not clear how these seemingly contradictory findings from mouse models and human cell lines can be reconciled with the recent single-molecule analysis in Xenopus.

Herein we have used forward genetics in murine cells to identify suppressor mutations that rescued DNA replication in the presence of RECQL4 mutation. This approach led to the identification of KLHDC3, the substrate-binding subunit of the Cullin-RING ligase (CRL) E3 that facilitates ubiquitin-mediated destruction of proteins with specific C-terminal degron motifs (Pilcher et al, 2025; Scott et al, 2024a). Loss of KLHDC3 normalised cell proliferation and DNA replication rates in *Recql4* mutated cells. This occurred by KLHDC3 loss leading to stabilisation of trace levels of a truncated RECQL4 protein containing a neo-degron specific for KLHDC3, formed after Cre-mediated recombination of the *Recql4*[fl] allele. While this rescue mechanism is restricted to this specific model and not generally applicable to RECQL4 mutation, it demonstrates that very low levels of truncated RECQL4—containing only the N-terminal 480 amino acids in-frame including its Sld2-like domain but lacking the ATP-dependent helicase domain and entire C-terminal portion of the protein—was capable of supporting DNA replication in vivo in mammalian cells. These results provide an orthogonal system to our previous work with a knock-in allele of a ATP-binding mutant (Castillo-Tandazo et al, 2019) demonstrating that the ATPase activity and helicase domain of RECQL4 are not essential for DNA replication in vivo in mouse models.

# Results

## Genome-wide loss of function screen to identify suppressors of RECQL4 mutation phenotypes

Our previous work using an allelic series of *Recql4* mutant mouse models had identified that in vivo loss-of-function RECQL4 mutations resulted in a fully penetrant bone marrow (BM) failure phenotype (Castillo-Tandazo et al, 2021; Castillo-Tandazo et al, 2019; Smeets et al, 2014). This did not occur in mice engineered to have an ATPase dead RECQL4 either as a germ-line mutation (*Recql4*[K525A/K525A]) or upon acute expression (*R26*-CreER[T2] *Recql4*[Δ/K525A]) (Castillo-Tandazo et al, 2019), indicating that the BM failure was not related to RECQL4's ATP-dependent helicase function. In contrast, mouse models acutely expressing a truncated but mislocalised (p.R347X) or truncated and unstable (p.G522EfsX6) RECQL4 protein developed BM failure (Castillo-Tandazo et al,

2021; Castillo-Tandazo et al, 2019). The p.R347X lacks the Zn knuckle domain and most of the region required for ssDNA and fork DNA binding (Castillo-Tandazo et al, 2021; Castillo-Tandazo et al, 2019). The Recql4 p.R347X protein is stable both when expressed ectopically or from the endogenous allele, but mis-localised and excluded from the nucleolus (Castillo-Tandazo et al, 2021). In vivo, homozygous *Recql4*[R347X/R347X] mice are not viable and do not survive to birth (Castillo-Tandazo et al, 2021). The p.G522Efs product is unstable and we have not been able to detect any protein product either of native protein or when GFP-fused and expressed as a cDNA (Castillo-Tandazo et al, 2021; Castillo-Tandazo et al, 2019).

To establish a tractable model suitable for studying this phenomenon we generated immortalised myeloid cell lines from the BM of the different *Recql4* alleles crossed to the *Rosa26*-CreER[T2] allele (Castillo-Tandazo et al, 2019; Heraud-Farlow et al, 2024; Wang et al, 2006; Xu et al, 2022). In these cells, the floxed *Recql4*[fl] allele is deleted upon tamoxifen-induced activation of the CreER[T2] recombinase. We observed that a homozygous null allele (*Recql4*[fl/fl]) of RECQL4 resulted in rapid proliferation arrest and cell death upon Cre activation compared to *Recql4*[fl/+] heterozygous cells (that retained one WT allele) used as a control. However, we also observed that the floxed allele deletion in the *Recql4*[R347X/Δ] retained a certain level of proliferation and viability (Castillo-Tandazo et al, 2021; Ng et al, 2015). We therefore chose to perform a screen using the p.R347X allele.

To identify genes whose loss restored proliferation and viability to *Recql4* mutant cell lines, we engineered *R26*CreER[T2] *Recql4*[fl/+] and *R26*CreER[T2] *Recql4*[fl/R347X] cell lines to constitutively express Cas9 for a loss-of-function genome-wide screen using a murine Brie sgRNA library (Fig. 1A) (Doench et al, 2016; Heraud-Farlow et al, 2024; Xu et al, 2022). After selection of Brie library infected cells, tamoxifen was added to delete the floxed *Recql4* allele (day 0). The cells were then cultured with tamoxifen, counted and their DNA pellets collected at day 0, day 4 and day 9 and the sgRNA copy number was determined by sequencing (Fig. 1A,B). The sgRNA copy number was compared between Days 4 and 9 to Day 0, respectively (Appendix Fig. S1A–F). Surprisingly, we identified only a single candidate that was significantly enriched in the *Recql4*[Δ/R347X] cells compared to the *Recql4*[+/Δ] control cells, sgRNA against Kelch domain containing 3 (*Klhdc3*) (Fig. 1B; Dataset EV1). KLHDC3 is a substrate receptor of the Cullin2-RING ligase (CRL2) that recognises and facilitates ubiquitin-mediated destruction of proteins with specific C-terminal motifs via the recently discovered C-end degron (DesCEND) pathway (Koren et al, 2018; Lin et al, 2018; Scott et al, 2024a; Scott et al, 2024b; Timms et al, 2023; Zhang et al, 2023). Assessment of the screen data revealed no enrichment of loss-of-function alleles of any other KLHDC family member nor other proteins known to interact with KLHDC3 (such as the other components of the CRL E3 complex), indicating that the mechanism of rescue was specific to loss of KLHDC3.

Loss of KLHDC3 did not have an appreciable effect on the control cells (*Recql4* heterozygous; fl/+) and the sg*Klhdc3* guides were not enriched in the control genotype (Fig. 1B). The loss of KLHDC3 was first validated in the same two Cas9 cell lines (Appendix Fig. S2D–F) and then in three additional (non-Cas9 expressing) cell lines including a *Recql4*[fl/fl] cell line (Fig. 1C–G; Appendix Fig. S2A–C). We validated the result using 4 different sgRNA (the top 2 ranked sgRNA from the Brie library and 2 additional sgRNA not present in the library)

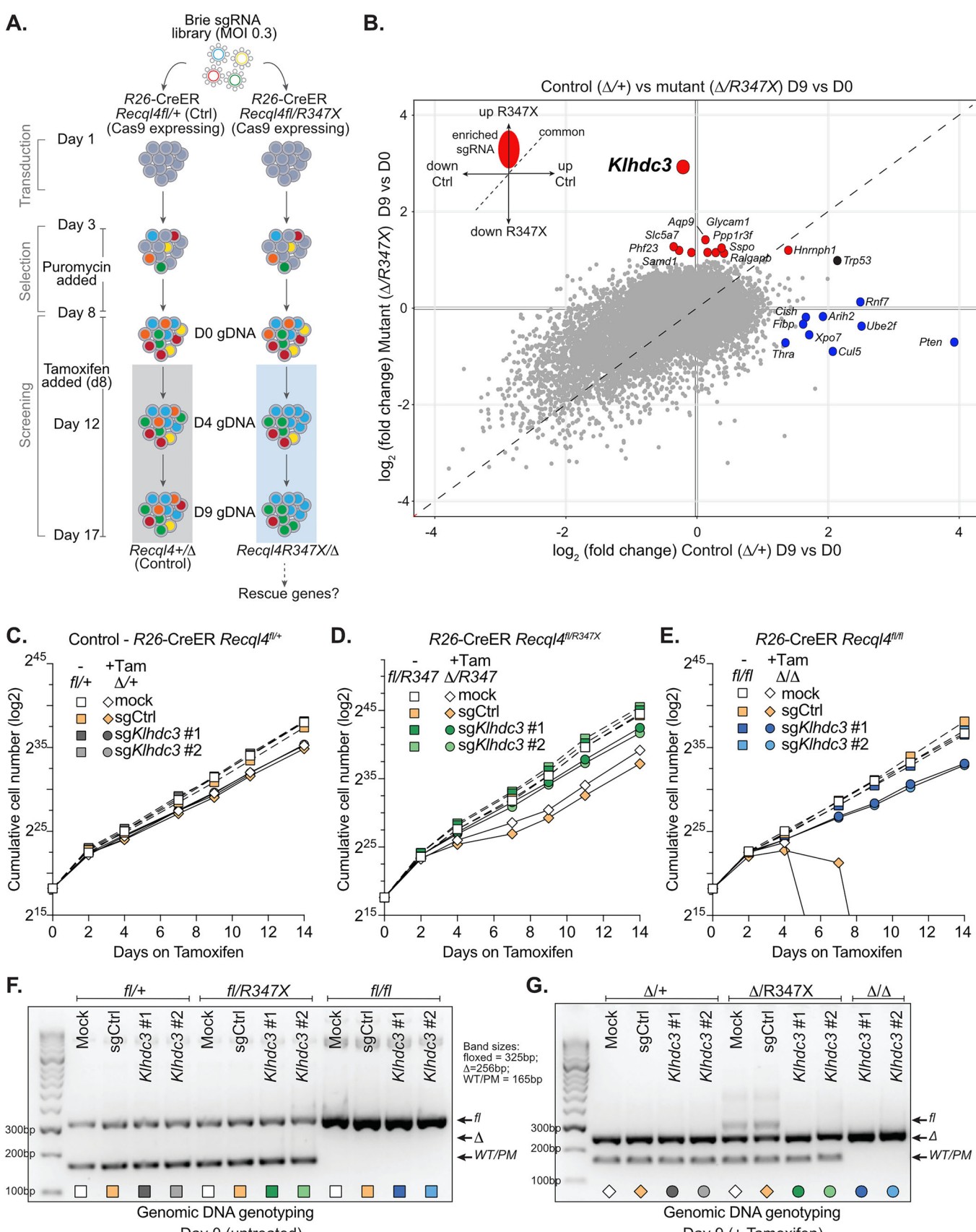

**Figure 1. Loss of *Klhdc3* rescues *Recql4* loss-of-function mutant myeloid cells.**

(A) Schematic outline of genetic screen designed to identify loss-of-function rescue alleles of proliferation defects induced by RECQL4 p.R347X mutation. Screen was performed on one Cas9 positive *R26*-CreER *Recql4*$^{fl/+}$ and on Cas9 positive *R26*-CreER *Recql4*$^{fl/R347X}$ cell line. (B) Summary of MaGeCK maximum likelihood estimation (mle) analysis of screen showing enrichment of sgRNAs targeting the indicated genes in the $\Delta/R347X$ cells compared with those in the $\Delta/+$ cells at day 9/day 0. Red dots, high-confidence indicate statistically significant enrichment in the $\Delta/R347X$ cells; blue indicated enriched sgRNA in the $\Delta/+$ cells. (C) Effect of loss of *Klhdc3* on control cells (*R26*-CreER *Recql4*$^{fl/+}$; become $\Delta/+$ cell following tamoxifen treatment applied at Day 0) proliferation. (D) Effect of loss of *Klhdc3* on RECQL4 p.R347X only expressing cells (*R26*-CreER *Recql4*$^{fl/R347X}$; become $\Delta/R347X$ cells following tamoxifen treatment applied at Day 0) proliferation. (E) Effect of loss of *Klhdc3* on Recql4-deficient cells (*R26*-CreER *Recql4*$^{fl/fl}$; become $\Delta/\Delta$ cells following tamoxifen treatment applied at Day 0) proliferation. (F) Genotyping PCR using genomic DNA of the *Recql4* locus at Day 0; each allele as indicated. (G) Genotyping PCR using genomic DNA of the *Recql4* locus at Day 9 of tamoxifen treatment; each allele as indicated. Source data are available online for this figure.

targeting different regions of *Klhdc3* in *R26*-CreER$^{T2}$ *Recql4*$^{fl/R347X}$ cell lines (Fig. 1C,D; Appendix Fig. S2A,B,D). To exclude survival and outgrowth of *R26*CreER$^{T2}$ *Recql4*$^{fl/R347X}$ cells that had not fully recombined the *Recql4*$^{fl}$ allele, we completed genotyping of the deletion efficiency in sgControl (sgCtrl) and sg*Klhdc3* cells. Strikingly, in sg*Klhdc3* cells, there was complete and stable recombination of the *Recql4*$^{fl}$ allele with complete loss of the full-length WT protein and stable expression of the p.R347X at ~28% of WT levels (Fig. 1F,G; Appendix Fig. S2E,F). Sequencing of the DNA confirmed this and that the *Klhdc3* mutation was homozygous and predicted to be deleterious (across all sgRNA used for *Klhdc3*) (Appendix Fig. S3A–C). The *Recql4*$^{\Delta/R347X}$ sg*Klhdc3* cells (referred to hereon as *Recql4*$^{\Delta/R347X}$ *Klhdc3*$^{\Delta/\Delta}$ cells) had a proliferation rate comparable to tamoxifen-treated control cells irrespective of the *Recql4* allele status.

These results prompted us to test if cells predicted to be null for RECQL4 (*R26*-CreER$^{T2}$ *Recql4*$^{fl/fl}$) would also be rescued by loss of KLHDC3. To our surprise, given the profound and rapid proliferation failure and cell death engendered by tamoxifen treatment of the *R26*-CreER$^{T2}$ *Recql4*$^{fl/fl}$ cells (Fig. 1E), the removal of KLHDC3 prior to tamoxifen treatment enabled the *Recql4*$^{\Delta/\Delta}$ *Klhdc3*$^{\Delta/\Delta}$ cells to survive and proliferate at the same rate as control cells (Fig. 1E–G; Appendix Fig. S2C). Therefore, this forward genetic screen identified that the loss of KLHDC3 specifically and uniquely allowed for sustained proliferation and viability of RECQL4-deficient or mutant expressing myeloid cells.

## Enhanced DNA replication licensing in the *Recql4*$^{\Delta/\Delta}$ *Klhdc3*$^{\Delta/\Delta}$ cells

We next sought to understand the characteristics of the rescued *Recql4*$^{\Delta/\Delta}$ *Klhdc3*$^{\Delta/\Delta}$ cells. We isolated and validated clones from two independently derived cell lines (made from different donor mouse bone marrows). These were confirmed as *Klhdc3* homozygous mutant and had undergone complete genomic recombination of the floxed *Recql4* allele with no RECQL4 protein detectable in whole cell western blot (Fig. 1G; Appendix Fig. S2E,G). Analysis of proliferation rates showed that the *Recql4*$^{\Delta/\Delta}$ *Klhdc3*$^{\Delta/\Delta}$ cells had a comparable proliferation rate to control cells, including *Klhdc3*$^{\Delta/\Delta}$ single mutant cells (Fig. 1C; Appendix Fig. S2A). We then assessed DNA replication rates using pulse labelling of asynchronous cultures of myeloid cells with 5-ethynyl-2′-deoxyuridine (EdU) incorporation (Fig. 2A). The EdU pulse labelling demonstrated that sg*Klhdc3* cells were indistinguishable from sgCtrl myeloid cells, indicating that loss of KLHDC3 alone did not change DNA replication and cell cycle phase entry rates perceptibly (Fig. 2A). Interestingly, while the overall cell cycle phase distribution of the *Recql4*$^{\Delta/\Delta}$ sg*Klhdc3* was not significantly different from either sgCtrl

or sg*Klhdc3*, there was a reproducible reduction in the overall intensity of the EdU signal in S phase of the cell cycle (Fig. 2A). This suggested a lower overall rate of EdU incorporation in the *Recql4*$^{\Delta/\Delta}$ *Klhdc3*$^{\Delta/\Delta}$ cells, with a more prominent reduction during the second half of S phase. A similar profile was also obtained upon deletion of *Recql4* in *Klhdc3* WT cells, although these cells fail to proliferate and die within 5–14 days (Fig. EV1A–C). To further confirm this change was due to the loss of RECQL4 and not an effect unique to the combination of *Recql4*$^{\Delta/\Delta}$ *Klhdc3*$^{\Delta/\Delta}$ loss, we re-expressed mCherry-mRECQL4 in the rescued cells and repeated the analysis (Fig. 2B). Re-expression of RECQL4 restored the EdU profiles and intensity/level of EdU signal in S phase to that of control cells. This demonstrates that the reduced EdU incorporation during S phase, despite a normal overall cell cycle phase distribution and entry time, was a specific result of RECQL4 loss (Fig. 2B). To gain more insight, we coupled EdU and Ki67 staining to obtain a more fine-grained analysis of cell cycle state and DNA replication rates (Fig. 2C). When compared to control cell lines, the *Recql4*$^{\Delta/\Delta}$ *Klhdc3*$^{\Delta/\Delta}$ cells had evidence of continuing DNA synthesis (EdU incorporation) during mitosis (Fig. 2C). Taken together, these results indicate that the *Recql4*$^{\Delta/\Delta}$ *Klhdc3*$^{\Delta/\Delta}$ cells have a level of replication stress, albeit seemingly tolerated, with incomplete DNA replication during S phase, but then utilise the Mitotic DNA Synthesis pathway (MiDAS) to complete replication and sustain largely normal proliferation and viability (Bhowmick et al, 2023; Bhowmick et al, 2016; Minocherhomji et al, 2015).

Based on these results, we used a FACS-based method to measure the amount of the replication origin licensing factor MCM3 bound to DNA during the different cell cycle phases (Fig. 2D) (Clijsters et al, 2019; Matson and Cook, 2020; Matson et al, 2017). From this analysis we found that loss of KLHDC3 alone did not change the level of MCM3 loading compared to WT cells (Fig. 2D). In contrast, the *Recql4*$^{\Delta/\Delta}$ *Klhdc3*$^{\Delta/\Delta}$ cells showed increased MCM3 loading intensity during G1 and early S phase compared to control cells (Fig. 2D). In conclusion, we have identified that loss of KLHDC3 specifically and uniquely rescued the cellular effects of the loss or mutation of the core essential protein RECQL4. This was associated with a mild DNA replication stress and enhanced replication licensing, allowing cells to maintain an overall proliferation and viability rate comparable to RECQL4 WT cells.

## Loss of *Klhdc3* in vivo protects against bone marrow failure induced by loss of *Recql4*

Next, we sought to determine if the rescue observed in the immortalised myeloid cells extended to in vivo phenotypes. To test this, we generated two *Klhdc3* mouse models: a *Klhdc3*$^{fl/fl}$ allele

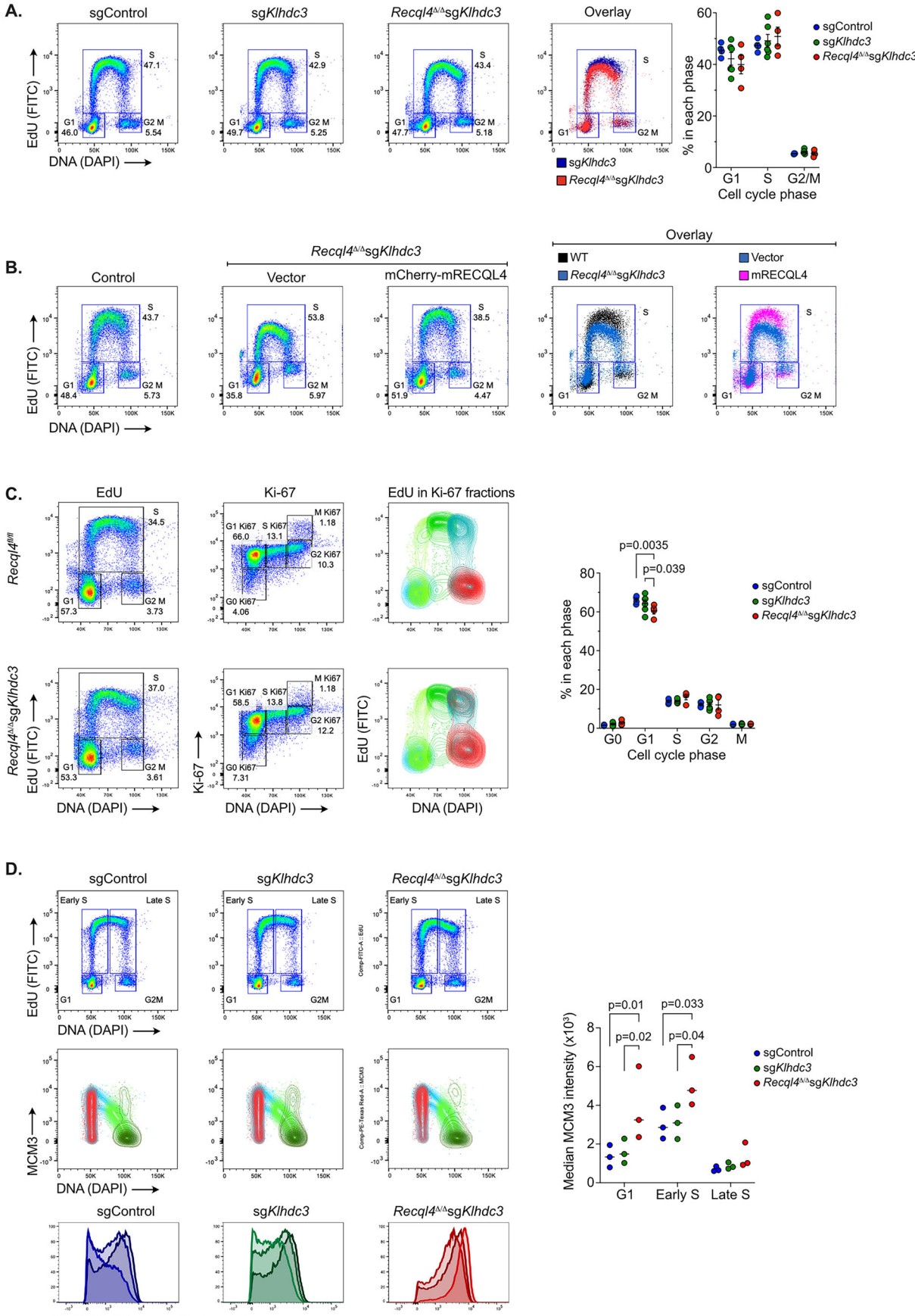

◄ **Figure 2.   Enhanced DNA replication licensing in the *Recql4*$^{\Delta/\Delta}$ *Klhdc3*$^{\Delta/\Delta}$ cells.**

(A) Representative flow cytometry plots assessing DNA replication rates following pulse labelling of asynchronous cultures of myeloid cells with EdU incorporation and DAPI (DNA stain). Genotypes assessed as sgControl (WT); sg*Klhdc3* and *Recql4*$^{\Delta/\Delta}$ sg*Klhdc3*. *Recql4*$^{\Delta/\Delta}$ sg*Klhdc3* reproducibly had a lower maximum intensity of EdU staining. Quantitation of the cell cycle distribution from independent replicates shown; *n* at least 4 per genotype. (B) Re-expression of mCherry-RECQL4 protein in the *Recql4*$^{\Delta/\Delta}$ sg*Klhdc3* cells. Control is sgControl (WT); vector is *Recql4*$^{\Delta/\Delta}$ sg*Klhdc3* reconstituted with empty vector and *Recql4*$^{\Delta/\Delta}$ sg*Klhdc3* mCherry-RECQL4 is cells reconstituted with full-length mouse RECQL4 protein. Overlays of the Edu/DAPI plots are provided. *n* = 2 for 185 K1B12 and *n* = 2 for 186 K1E1 clone (repeated 3 times). (C) Analysis of cell cycle distribution using combined EdU, DAPI and Ki-67 staining. Representative flow cytometry plots for control (*Recql4*$^{fl/fl}$; non-tamoxifen treated cells retaining *Recql4*) and *Recql4*$^{\Delta/\Delta}$ sg*Klhdc3* cells. First panel shows EdU vs DAPI, second panel shows Ki-67 vs DAPI and third panel shows the distribution of EdU within the fractions identified by Ki-67 staining. The *Recql4*$^{\Delta/\Delta}$ sg*Klhdc3* cells show a persistence of DNA replication in the G2 and M phase of the cell cycle. Quantitation of the cell cycle phases provided; *n* = 4 per genotype. (D) Analysis of chromatin bound MCM3 in sgControl (WT); sg*Klhdc3* and *Recql4*$^{\Delta/\Delta}$ sg*Klhdc3* cells. Populations were fractionated based on cell cycle distribution based on Edu/DAPI into G1, Early S, Late S and G2/M phases as indicated (top row). The levels of chromatin loaded MCM3 were then assessed in each phase (middle row) and then specifically in the G1 phase (bottom row). The median MCM3 loading in G1 (based on fluorescence intensity) of three cell lines each genotype is shown as histogram overlays; *n* = 3 cell lines per genotype. Data expressed as mean ± sem from independent experiments; *n* ≥ 3 except panel (B) where representative FACS plots are shown; Statistical significance determined from ANOVA with multiple-comparisons correction calculated using Prism software. Source data are available online for this figure.

(Fig. 3A) and a germ-line deficient *Klhdc3*$^{+/-}$ (phenotype described in detail elsewhere) (Buco et al, 2026). These were generated on a C57BL6/J background using endonuclease-mediated recombination. Both models result in the deletion of exons 4 and 5, with the germ-line deficient (*Klhdc3*$^{+/-}$) allele arising as a result of recombination and deletion of the region encompassing the loxP flanked region during targeting. We tested in vivo rescue in two contexts.

Firstly, we tested whether the loss of *Klhdc3* would rescue the fully penetrant in vivo BM failure phenotype previously characterised in *Recql4* deficient and point mutant mice (Fig. 3B) (Castillo-Tandazo et al, 2019; Smeets et al, 2014). We generated cohorts of *R26*-CreER$^{ki/+}$ *Recql4*$^{fl/fl}$ *Klhdc3*$^{fl/fl}$ and mixed *Recql4/Klhdc3* wild type and heterozygous floxed controls (littermates, co-housed and treated similarly, both males and females used), and fed them with tamoxifen containing food for up to 28 days as previously described (Fig. 3B) (Castillo-Tandazo et al, 2019; Smeets et al, 2014). All mice lost weight initially, associated with the introduction of the tamoxifen diet and independent of genotype (Fig. 3C). In contrast to control genotypes that fully recovered weight after the first 2 weeks of treatment, the single mutant *Recql4*$^{\Delta/\Delta}$ mice had a steady and progressive weight loss throughout the tamoxifen treatment period and had to be euthanised at day 21 (Ethical End Point; EEP) (Fig. 3C). In accordance with our previous work, several of the *Recql4*$^{\Delta/\Delta}$ mice exhibited pale feet and tail tips, consistent with the development of severe anaemia that accompanies the BM failure phenotype (Castillo-Tandazo et al, 2019; Smeets et al, 2014). In contrast, the *Recql4*$^{\Delta/\Delta}$*Klhdc3*$^{\Delta/\Delta}$ double mutant mice regained weight similar to control genotypes at day 28 and did not display clinical signs of anaemia. Assessment of the DNA from BM cells demonstrated that the *Recql4* alleles had complete deletion while *Klhdc3* was fully recombined suggesting that loss of KLHDC3 was able to protect from BM failure induced by a loss of *Recql4* in vivo (Fig. 3D). Analysis of the peripheral blood confirmed severe anaemia (reduced red blood cell numbers and haematocrit; HCT) in the single mutant *Recql4*$^{\Delta/\Delta}$ mice, which was prevented by concomitant loss of *Klhdc3* (Fig. 3E,F). Similarly, total cellularity of the BM, spleen and thymus was rescued in the *Recql4*$^{\Delta/\Delta}$*Klhdc3*$^{\Delta/\Delta}$ double mutant mice (Fig. 3G). *Recql4*$^{\Delta/\Delta}$ mutant mice had reduced cellularity in the myeloid lineage and erythro-myeloid progenitors, which again was completely rescued in *Recql4*$^{\Delta/\Delta}$*Klhdc3*$^{\Delta/\Delta}$ mice to comparable levels as in control mice (Fig. 3H,I).

Finally, we also looked at embryonic development. Homozygous *Recql4*$^{-/-}$ (a germ-line deleted version of the *Recql4*$^{fl/fl}$ allele used herein) is lethal before embryonic day 10.5 (E10.5) (Smeets et al, 2014). When *Recql4*$^{+/-}$ and *Klhdc3*$^{+/-}$ alleles were crossed, viable and macroscopically normal *Recql4*$^{-/-}$*Klhdc3*$^{-/-}$ embryos were detected at the expected Mendelian ratio at E12.5, although no *Recql4*$^{-/-}$*Klhdc3*$^{-/-}$ animals were identified postnatally (Fig. EV2) (Smeets et al, 2014). Collectively, these results demonstrate that KLHDC3 deletion completely prevents the development of BM failure in adult conditional RECQL4-deleted mice in vivo and significantly extended the survival of *Recql4*$^{-/-}$ embryos.

## KLHDC3 is a specific rescue allele and involves its E3 activity

Having established that loss of *Klhdc3* protected against the phenotypes associated with a loss or mutation of *Recql4* both in cell lines and in vivo, we sought to understand mechanistically how this occurred. We generated a series of KLHDC3 mutant constructs to re-express in the rescued *Recql4*$^{\Delta/\Delta}$*Klhdc3*$^{\Delta/\Delta}$ mutant cell lines. We made a wild-type KLHDC3, a dominant negative (DN) point mutated KLHDC3, with mutations in the Elongin B/C box and Cullin 2 binding domain that render it unable to participate in the degradation of substrates, and a C-terminal deleted mutant than lacks the Elongin and Cullin 2 interaction domains entirely (Fig. 4A) (Mahrour et al, 2008). The re-expression of the WT but not the mutant KLHDC3 proteins resulted in death of the *Recql4*$^{\Delta/\Delta}$*Klhdc3*$^{\Delta/\Delta}$ mutant cell lines without any appreciable effect on control cells (Fig. 4B). This demonstrates that the activity of KLHDC3 as a substrate recognition receptor that recognises and facilitates ubiquitin-mediated destruction of proteins with specific C-terminal degron motifs is the specific function accounting for the observed rescue.

Having established that the rescue was specific to the substrate recognition capacity of KLHDC3, we sought to identify the key substrate(s) that were being stabilised in the absence of KLHDC3 that could explain the rescue of *Recql4*$^{\Delta/\Delta}$. As computational prediction of KLHDC substrates based degron-like C-termini is notoriously unreliable (Yeh et al, 2021), we turned to genetic screening. We posited that deletion of candidate KLHDC3 substrate(s) in the *Recql4*$^{\Delta/\Delta}$*Klhdc3*$^{\Delta/\Delta}$ mutant cell lines should revert the rescue and result in specific lethality due to the absence of RECQL4 (Fig. 4C). Cas9 expressing *Recql4*$^{\Delta/\Delta}$*Klhdc3*$^{\Delta/\Delta}$

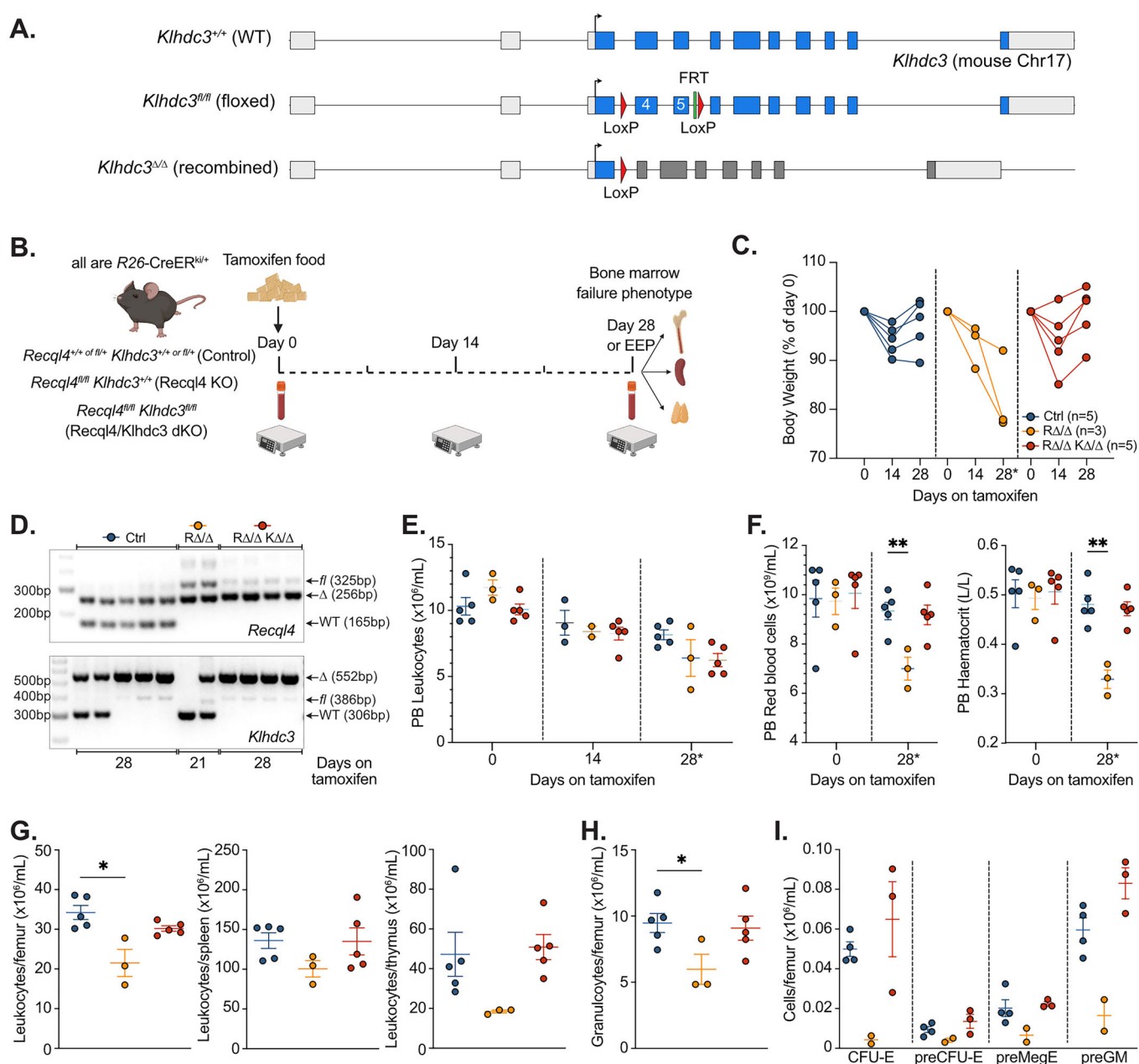

**Figure 3. Loss of KLHDC3 prevents bone marrow failure in RECQL4-deficient mice in vivo.**

(A) Schematic outline of *Klhdc3* conditional allele; exons 4 and 5 are flanked by loxP elements. (B) Schematic outline of experiment. All animals are *R26*-CreER$^{T2Ki/+}$ and were fed tamoxifen containing food ad libitum from Day 0. Control mice are of mixed genotypes containing wild-type or heterozygous alleles of either *Recql4* or *Klhdc3* and are littermates of the *Recql4* KO (*Recql4$^{fl/fl}$ Klhdc3$^{+/+}$*) and *Recql4/Klhdc3* dKO (*Recql4$^{fl/fl}$ Klhdc3$^{fl/fl}$*). Both male and female mice were used for all genotypes. All animals were assessed on day 0 and day 14 and then either at day 28 or EEP (ethical endpoint based on health assessment) which ever came first. (C) Body weights of mice of each indicated genotype; each line represents an individual. (D) Genotyping of genomic DNA from whole bone marrow cells for *Recql4* and *Klhdc3* recombination, respectively. Days on tamoxifen indicate day of collection. (E) Peripheral blood leukocyte counts at indicated day for each genotype cohort. Day 28* indicates either day 28 or EEP. (F) Peripheral blood red blood cell counts and haematocrit either day 28 or EEP. (G) Total leukocyte counts in per femur, spleen and thymus, respectively, either day 28 or EEP. (H) Granulocyte numbers per femur in the bone marrow at either day 28 or EEP. (I) Analysis of myelo-erythroid progenitors in the bone marrow of each genotype either day 28 or EEP. Populations assessed are within the Lineage$^-$c-Kit$^+$Sca-1$^-$ population of the bone marrow and are the colony-forming unit erythroid (CFU-E); pre-colony-forming unit erythroid (pre-CFU-E); pre-megakaryocyte erythroid progenitor (preMegE) and pre-granulocyte macrophage progenitor (preGM). Data pooled and presented from 3 independent cohorts of animals. Data expressed as mean ± sem with data pooled from two independent experiments; *n* as indicated by each dot; $P < 0.05$, **$P < 0.01$ as indicated with statistical comparisons from ANOVA with multiple-comparisons correction calculated using Prism software. Source data are available online for this figure.

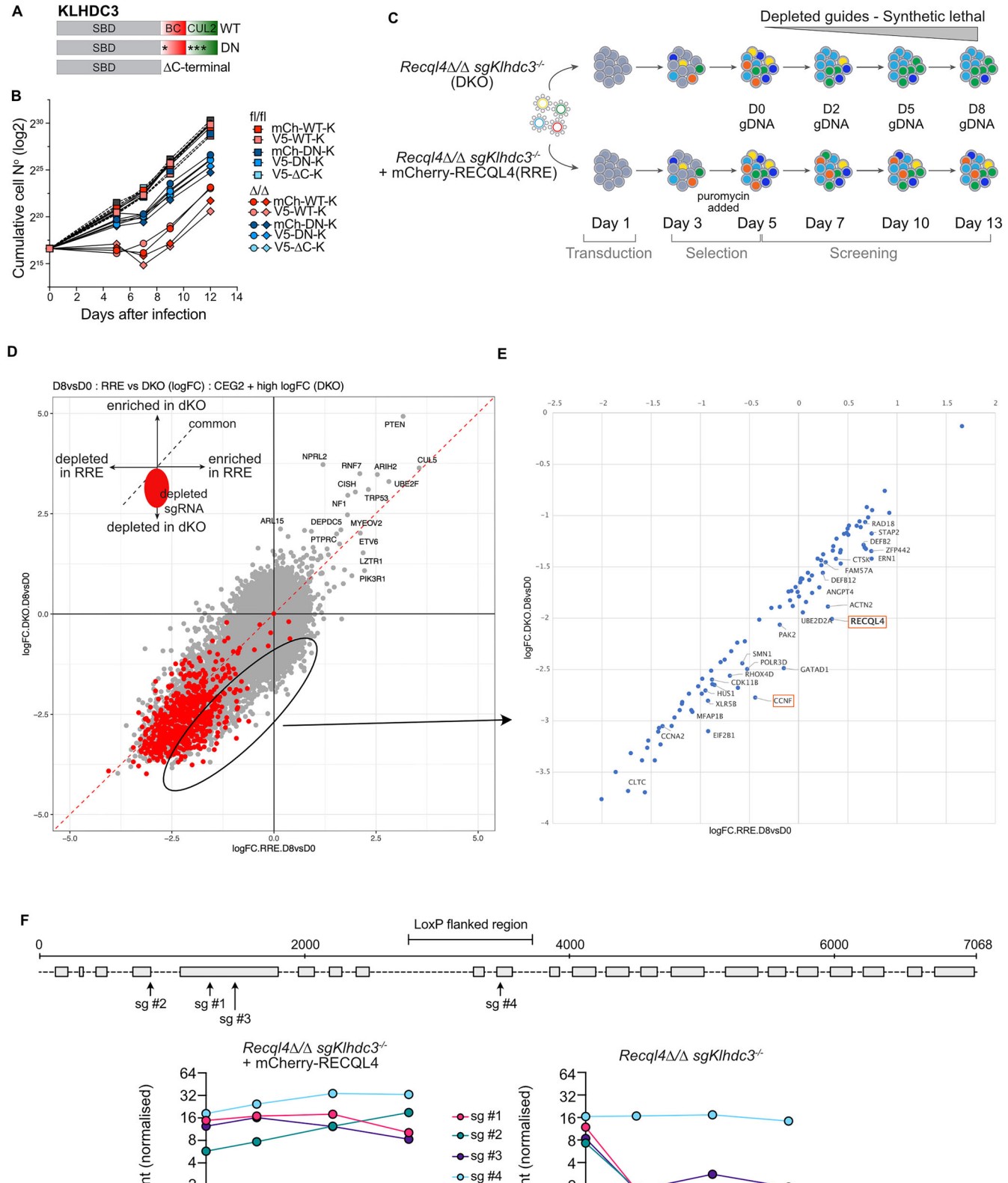

◀ **Figure 4. KLHDC3 is a specific rescue allele and its E3 activity is required.**

(A) Schematic outline of KLHDC3 constructs used (WT: wild type, DN: dominant negative, *: point mutations, Δ: deletion). (B) Re-expression of KLHDC3 or the DN or ΔC terminal mutant in Recql4fl/fl or Recql4Δ/Δ sgKlhdc3 cells. Data expressed as cumulative cell number after infection with the KLHDC3 construct indicated. Data representative of three experiments using 2-4 cell lines each genotype per experiment. Experiment performed with 2 different fl/fl cell lines (185 and 186), 2 different rescued clones (185 K1B12 and 186 K1E1), 5 different constructs (mCh WT and DN Klhdc3 and V5 WT, DN and delta Klhdc3) used in different combinations in 3 independent experiments. (C) Outline of genetic screen designed to identify if loss of individual substrates of KLHDC3 would result in lethality of Recql4Δ/Δ sgKlhdc3 cells (labelled as DKO – double knock-out) and to identify synthetic lethal interactions with Recql4Δ/Δ sgKlhdc3. Cells were compared to control Recql4Δ/Δ reconstituted with mCherry-RECQL4 (labelled as RRE – RECQL4 re-expressed). (D) Summary of MaGeCK maximum likelihood estimation (mle) analysis of screen showing depletion of sgRNAs targeting the indicated genes in the Recql4Δ/Δ sgKlhdc3 cells (DKO) compared with those reconstituted with mCherry-RECQL4 (RRE) at day 8/day 0. Red dots, high-confidence, indicate statistically significant depletion. Area of interest indicated and expanded into (E). (E) Expanded view of the sgRNAs displaying preferential depletion in the Recql4Δ/Δ sgKlhdc3 cells. (F) Schematic outline of where the individual sgRNA against Recql4 align relative to the region deleted in the Recql4fl/fl mice (marked as LoxP flanked region). Normalised individual sgRNA counts for each indicated sgRNA against Recql4 in the control (RRE) or Recql4Δ/Δ sgKlhdc3 (DKO) cells from the synthetic lethal screen. sgRNAs that are depleted target sequences 5′ to the LoxP flanked region that has been deleted in the Recql4Δ/Δ sgKlhdc3 cells. Source data are available online for this figure.

cells (referred to as DKO) and control cells complemented with WT RECQL4 (referred to as RRE; RECQL4 re-expression) were infected with the Brie sgRNA library. At day 0 and days 2, 5 and 8, cells were collected, and DNA was extracted and assessed for sgRNA representation to identify sgRNA that were specifically depleted in the Recql4Δ/ΔKlhdc3Δ/Δ cells compared to control cells re-expressing RECQL4. This resulted in a broader range of candidate genes than the initial screen (Fig. 1B; Dataset EV1), including many core essential genes and those with known involvement in cell cycle control and DNA replication (Fig. 4D,E; Dataset EV2). Moreover, four of these genes, Dna2, Gins4, Rfc3 and Timeless, were also recently identified in human dual-guide CRISPRi screening with RECQL4 sgRNAs in RPE-1 cells (Fielden et al, 2025), suggesting a conservation of genetic pathways across species (Appendix Fig. S4). As an example of identified candidates, we validated synthetic lethality with loss of Cyclin F (Ccnf) by showing that guides targeting Ccnf were more rapidly depleted from the Recql4Δ/ΔKlhdc3Δ/Δ cells compared to control cells (Fig. EV3). We also saw evidence consistent with a level of functional redundancy between RECQL4 and MCM10, both in the screen and with individual sgRNA experiments, consistent with a recent preprint (Terui et al, 2025). This demonstrated that the screen was robust in resolving synthetic lethal genetic interactions with the loss of RECQL4.

Remarkably, we also identified 3 of the 4 sgRNA targeting Recql4 amongst the most highly depleted candidates (Fig. 4F). The identification of Recql4 sgRNA as synthetic lethal interaction in the Recql4Δ/ΔKlhdc3Δ/Δ cells but not in the control was puzzling. The Recql4fl/fl allele used here phenocopies other reported null alleles both with respect to both germline and lineage-specific deletions (Lu et al, 2015; Ng et al, 2015). This floxed allele results in the loss of exons 9 and 10, with introduction of multiple in-frame stop codons expected to result in non-sense-mediated decay of the transcript and a functionally protein null state (Castillo-Tandazo et al, 2019; Smeets et al, 2014). Interestingly, the Recql4 sgRNAs that demonstrated synthetic lethality targeted Recql4 in exons 4 and 5, prior to the loxP sites that remove the protein coding region from exons 9 and 10 (Fig. 4F). This was confirmed by individually targeting exons 4, 5, 10 and 15 in Recql4fl/fl control, Klhdc3-/- and Recql4Δ/ΔKlhdc3Δ/Δ cells. The most plausible explanation for this finding was that the residual Cre/loxP deletion product of the Recql4fl allele itself might be the target of KLHDC3, and that stabilisation of this truncated protein product was sufficient for the observed rescue.

## KLHDC3 loss mediates rescue by stabilising the recombined RECQL4-truncation fragment

The recombined Recql4Δ mRNA transcript is predicted to retain the first 480 amino acids (aa) of RECQL4 (from exons 1–8) plus an additional frame-shifted 50aa (from exons 11 and 12), lacking the entire helicase domain and C-terminal region. Notably, the predicted C-terminal protein sequence of this 530aa protein was VPRGLGGRG*, approximating the previously characterised KLHDC3 recognition motif of RxxxxRG* (Fig. 5A) (Lin et al, 2015; Lin et al, 2018; Rusnac et al, 2018; Timms et al, 2023; Yeh et al, 2021; Zhang et al, 2023).

Western blotting completed after the initial identification of the rescue using whole cell lysate of the Recql4Δ/Δ Klhdc3Δ/Δ myeloid cells had not been able to detect a protein of the predicted size of the 530aa RECQL4 protein product of the Recql4Δ allele (Appendix Fig. S2F,G). We revisited this with an improved sample preparation method and additional protease inhibition and were able to detect a protein of ~58 kDa, the expected mass based on the sequence (Fig. 5B). This was detectable in whole cell lysates at between 5.5–7% of the levels of the endogenous full-length RECQL4. As the predicted protein is expected to retain the nuclear localisation signal as part of the in-frame portion of RECQL4, we assessed where the truncated protein product was located. Based on compartmental fractionation, the truncated protein was exclusively associated with the chromatin fraction of the cell, suggesting that the Recql4Δ protein product was retained on the DNA and potentially sufficient to compensate for the absence of full-length RECQL4 (Fig. 5B).

Having established conditions to detect the truncated protein product, we then sought to directly establish that KLHDC3 was stabilising the 530aa protein. We re-expressed WT and C-terminal truncated mutant of KLHDC3 described earlier in Recql4+/ΔKlhdc3Δ/Δ cells (derived from bone marrow from offspring of Recql4+/- x Klhdc3+/- crosses) (Smeets et al, 2014). The expression of WT KLHDC3, but not C-terminal truncated, led to the loss of the detected 530aa RECQL4Δ protein product (Fig. 5C). These results collectively establish that the rescue of Recql4Δ/Δ cells, and point mutant cells used in the screen which also had the Recql4fl allele in them, was most likely due to KLHDC3 loss leading to stabilisation of the remnant RECQL4 protein product resulting from recombination of the floxed allele.

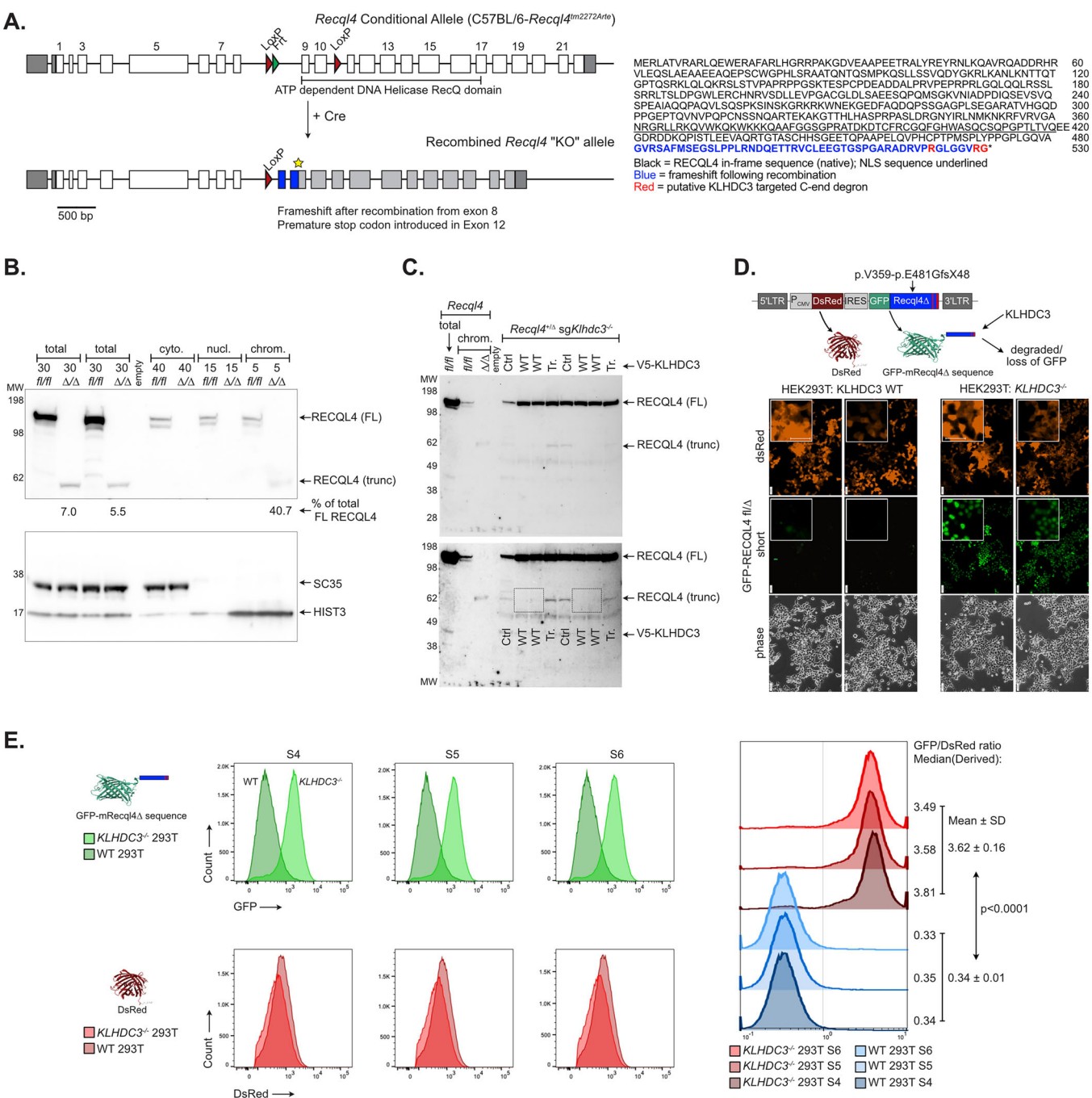

**A.** *Recql4* Conditional Allele (C57BL/6-*Recql4*tm2272Arte)

MERLATVRARLQEWERAFARLHGRRPAKGDVEAAPEETRALYREYRNLKQAVRQADDRHR 60
VLEQSLAEAAEEAQEPSCWGPHLSRAATQNTQSMPKQSLLSSVQDYGKRLKANLKNTTQT 120
GPTQSRKLQLQKRSLSTVPAPRPPGSKTESPCPDEADDALPRVPEPRPRLGQLQQLRSSL 180
SRRLTSLDPGWLERCHNRVSDLLEVPGACGLDLSAEESQPQMSGKVNIADPDIQSEVSVQ 240
SPEAIAQQPAQVLSQSPKSINSKGRKRKWNEKGEDFAQDQPSSGAGPLSEGARATVHGQD 300
PPGEPTQVNVPQPCNSSNQARTEKAKGTTHLHASPRPASLDRGNYIRLNMKNKRFVRVGA 360
NRGRLLRKQVWKQKWKKKQAAFGGSGPRATDKDTCFRCGQFGHWASQCSQPGPTLTVQEE 420
GDRDDKQPISTLEEVAQRTGTASCHHSGEETQPAAPELQVPHCPTPMSPLYPPGPLGQVA 480
**GVRSAFMSEGSLPPLRNDQETT**RVCLEEGTGSPGARADRV**P**RGLGGV**RG*** 530

Black = RECQL4 in-frame sequence (native); NLS sequence underlined
Blue = frameshift following recombination
Red = putative KLHDC3 targeted C-end degron

ATP dependent DNA Helicase RecQ domain
+ Cre
Recombined *Recql4* "KO" allele

500 bp

Frameshift after recombination from exon 8
Premature stop codon introduced in Exon 12

**B.**

RECQL4 (FL)
RECQL4 (trunc)

7.0    5.5    40.7    % of total FL RECQL4

SC35
HIST3

**C.**

V5-KLHDC3
RECQL4 (FL)
RECQL4 (trunc)

RECQL4 (FL)
RECQL4 (trunc)
V5-KLHDC3

**D.** p.V359-p.E481GfsX48

5'LTR  P_CMV  DsRed  IRES  GFP  Recql4Δ  3'LTR
KLHDC3
DsRed    GFP-mRecql4Δ sequence → degraded/loss of GFP

HEK293T: KLHDC3 WT          HEK293T: *KLHDC3*-/-

dsRed
GFP-RECQL4 fl/Δ short
phase

**E.**

GFP-mRecql4Δ sequence
*KLHDC3*-/- 293T
WT 293T

DsRed
*KLHDC3*-/- 293T
WT 293T

S4    S5    S6

GFP/DsRed ratio Median(Derived):

3.49
3.58    Mean ± SD
3.81    3.62 ± 0.16

p<0.0001

0.33
0.35    0.34 ± 0.01
0.34

*KLHDC3*-/- 293T S6    WT 293T S6
*KLHDC3*-/- 293T S5    WT 293T S5
*KLHDC3*-/- 293T S4    WT 293T S4

To assess the stability of the predicted *Recql4*Δ protein product, we fused its last 172aa of the *Recql4*Δ protein product (122aa of in-frame RECQL4 and 50aa of frameshifted sequence; from position 358 of the native protein; RECQL4Δ p.358-530aa) to an N-terminal GFP, and compared its stability to the constitutively expressed DsRed using the global protein stability (GPS) assay (Fig. 5D) (Yen and Elledge, 2008; Yen et al, 2008). The ratio of GFP to DsRed reflects of the stability of the C-end degron fused to GFP. When expressed in KLHDC3 wild-type HEK293T cells we could detect robust DsRed signal but no GFP by either microscopy (Fig. 5D) or flow cytometry (Fig. 5E). We then expressed the construct in

*KLHDC3*-/- HEK293T cells (Fig. EV4A) and obtained robust expression of GFP (Fig. 5D,E; independent replicates in Fig. EV4B). We confirmed the specificity of *KLHDC3*-/- HEK293T cells using previously characterised GPS-reporter constructs that are targeted by KLHDC3 or APPBP2, respectively (Fig. EV4C,D) (Huang et al, 2023; Lin et al, 2018; Yeh et al, 2021). This formally established that the transcript arising from the *Recql4*Δ allele, if translated, was forming a C-end degron recognised by KLHDC3 and specifically stabilised by loss of KLHDC3. These results collectively demonstrate that the rescue of *Recql4*Δ/Δ and point mutant cells, which also contain the *Recql4*fl allele, was due to KLHDC3 deletion-induced

**Figure 5. The recombined *Recql4* allele protein product is stabilised by loss of KLHDC3.**

(A) Analysis of the predicted protein product arising from recombination of the *Recql4^fl/fl^* allele. The predicted protein product remains in frame for the first 480 amino acids (indicated in black font) then is predicted to be frameshifted (indicated in blue font) with a stop codon at 530aa. The putative KLHDC3 degron motif (RxxxxxRG*) is highlighted in red font. (B) Western blot analysis of whole cell lysate or cytoplasmic, nuclear and chromatin fractions from cells of the indicated genotype assessing RECQL4 (upper panel) or SC35 or HIST3 (lower panel) expression. % of truncated RECQL4 as a percentage of full length (in fl/fl cell) as quantitated by iBright (Invitrogen/Thermo Fisher) image analysis software; fl/fl = *Recql4^fl/fl^*; Δ/Δ = *Recql4^Δ/Δ^* sg*Klhdc3*. Data representative of 2 independent replicates. (C) Re-expression of wild-type full length V5-tagged KLHDC3 (either empty vector; WT1, WT2 or ΔC terminal mutant (Tr)) in *Recql4^+/-^ Klhdc3^-/-^* cells. Experiment repeated twice with the same cell line: once without MG132 (lanes 5–8) and once with MG132 (lanes 9–12). Lanes 5 and 8: Ctrl (no vector), lanes 6, 7 and 10, 11: WT KLHDC3, and lanes 8 and 12: ΔC-term KLHDC3. (D) The C-terminal region of the *Recql4* deleted allele coding for p.V359-E481GfsX48 was cloned into the GPS-reporter plasmid. In this plasmid DsRed is constitutively expressed and GFP stability/expression is determined by the C-terminus of the fused RECQL4 deleted product. This reporter was expressed in KLHDC3 WT 293 T cells or *KLHDC3^-/-^* K2D10 293 T cells as indicated. Fluorescence signal was detected by live cell fluorescent imaging; images of RECQL4 p.V359-E481GfsX48 clone S4. Scale bar represents 50 μm. Inset images scale bar represents 50 μm. (E) Flow cytometric analysis of the GFP and DsRed expression of 3 independently generated RECQL4 p.V359-E481GfsX48 clones (indicated as S4, S5 and S6, respectively) transduced in KLHDC3 WT *KLHDC3^-/-^* 293 T cells. Stabilisation of GFP is quantitated by the GFP/DsRed derived median (right panel), data expressed as mean + /St dev; $p < 0.0001$ calculated using an unpaired t test in Prism. The left shift in the KLHDC3 WT 293 T cells indicative of destruction of the GFP-RECQL4 p.V359-E481GfsX48 GPS reporter. Source data are available online for this figure.

stabilisation of the remnant RECQL4 protein product resulting from recombination of the floxed allele.

## Genome-wide replication timing is maintained in the absence of the RECQL4 ATP Helicase domain and activity

A recent study suggested that, unlike what was previously reported for RECQL4 p.K525A ATP-binding mutant mice (Castillo-Tandazo et al, 2019), there was a requirement for the ATP-dependent helicase activity of xRECQL4 to support normal DNA replication in metazoans (Terui et al, 2024). This study used single-molecule resolution analysis in *Xenopus* oocyte extracts to conclude that the ATP-dependent activity of RECQL4 was required to efficiently remove DONSON from the Cdc45/Mcm2-7/GINS (CMG) helicase to allow DNA replication to proceed. The KLHDC3 loss mediated rescue of the myeloid cell lines allowed us to directly address this question, as the 530aa RECQL4^Δ^ protein product does not encode the ATP-dependent helicase region nor the entire C-terminal domains of RECQL4. Therefore, if the ATP-dependent helicase activity was essential, we should see significant changes in replication timing in the *Recql4^Δ/Δ^Klhdc3^Δ/Δ^* rescued cells. We directly assessed genome-wide DNA replication kinetics using whole-genome sequencing and the Timing Inferred from Genome Replication (TIGER) computational analysis method in WT and *Recql4^Δ/Δ^Klhdc3^Δ/Δ^* cells (Koren et al, 2021). TIGER analysis assesses DNA copy number and then corrects for alignability, GC sequence bias and copy number variations, then normalizes the copy number data to enable high-resolution, genome-wide mapping of replication timing profiles from bulk cultures of cells. We resolved high resolution (10 kb) maps of DNA replication timing in all cell lines assessed and from this were able to compare replication timing in the presence of full length RECQL4 (WT control cells) or when there was no ATP-dependent helicase region at all (*Recql4^Δ/Δ^Klhdc3^Δ/Δ^* cells) (Figs. 6A–D and EV5) (Koren et al, 2021). This analysis demonstrated that there were, at most, subtle differences between cells with and without the ATP-dependent helicase and C-terminal domains of RECQL4 (Fig. 6D). When assessed across individual chromosomes there was only a small variation apparent for the majority of chromosomes (Fig. EV5). There was more variability in the *Recql4^Δ/Δ^Klhdc3^Δ/Δ^* cells compared to the controls, potentially derived from sub-clonal variance that is apparent in the compound mutants (Figs. 6D and EV5). This

analysis, while not able to completely exclude true difference in replication timing, indicate that any difference at most is small. Therefore, we can conclude that the ATP-dependent helicase and C-terminal domains of RECQL4 are not essential for DNA replication to proceed in an ordered and efficient manner in the myeloid cell lines. In addition, the complete protection from BM failure in the *Recql4^Δ/Δ^Klhdc3^Δ/Δ^* animals further suggests that this also applies in vivo, consistent with our previous in vivo work demonstrating that the ATP-dependent helicase function of RECQL4 was not essential in mammals (Castillo-Tandazo et al, 2019).

## Discussion

Mutations in RECQL4 are associated with Rothmund-Thomson Syndrome (RTS) type II, characterised by both relatively benign pathologies, including low bone mass and immune compromise, and a highly elevated rate of cancer, particularly osteosarcoma and a range of other malignancies including haematological cancers (Larizza et al, 2010; Wang et al, 2001; Ashraf et al, 2025). Despite this association with severe human disease, the fundamental biology of RECQL4 and our understanding of its genetic interactions is very restricted compared to the related RecQ helicases such as BLM. RECQL4 was thought to be the mammalian homologue of the essential yeast protein Sld2. However, more recently it has been speculated that DONSON may be the mammalian Sld2 homologue (Cvetkovic et al, 2023; Evrin et al, 2023; Hashimoto et al, 2023; Kingsley et al, 2023; Lim et al, 2023). Recent studies have also suggested an interaction between the N-terminus of RECQL4 and MUS81, and a potential role for this interaction in alleviating replication stress and ensuring proper chromosome segregation (Ashraf et al, 2025). Therefore, significant gaps remain in our understanding of RECQL4's function. We sought to address this knowledge gap using forward genetic screens and human like mutations engineered in mouse cells and in vivo in murine models. While our study yielded an unexpected finding specifically related to models we used, it has allowed us to make significant advances in understanding both the genetic interactions and functions of RECQL4 protein in mammals.

We undertook a genome-wide suppressor screen to identify factors whose loss restored proliferative capacity and viability to cells with a non-viable RECQL4 truncating mutation (Castillo-

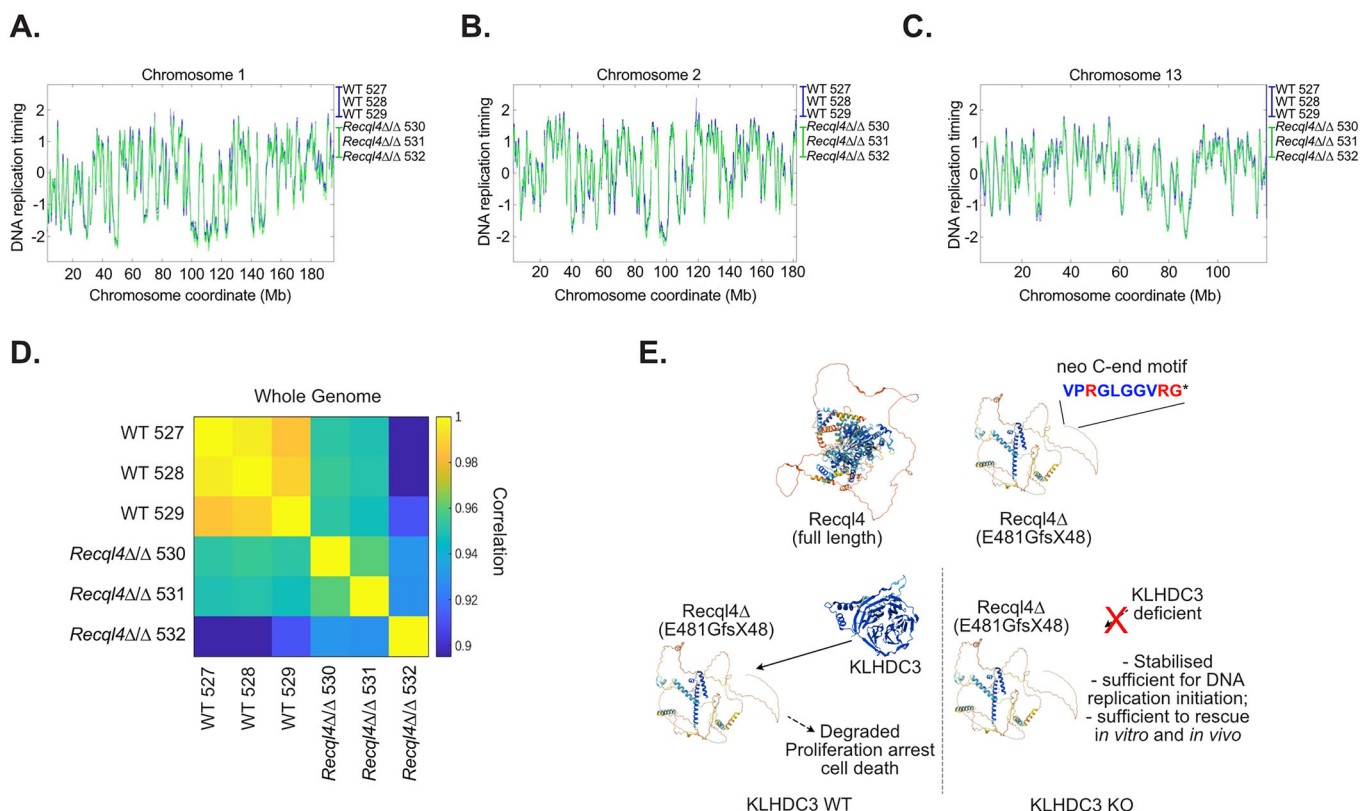

**Figure 6. Normal genome-wide DNA replication timing does not require the helicase domain of RECQL4.**

(A) Computed DNA replication timing across chromosome 1 with each genotype averaged to a single line per genotype (blue = WT; green = *Recql4^{Δ/Δ}* sg*Klhdc3* cells). (B) Calculated DNA replication timing for chromosome 2. (C) Calculated DNA replication timing for chromosome 13. (D) Correlation analysis of DNA replication timing across the whole genome; whole-genome sequencing performed on *n* = 3 independent cell lines per genotype. (E) Graphical summary of the results. In wild-type cells, the protein product from the *Recql4^{fl/fl}* allele recombination (*Recql4^{Δ/Δ}*; RECQL4 p.E481GfsX48) is unstable and subjected to targeting by KLHDC3 due to the formation of a specific neo C-end degron motif recognised by KLHDC3. This leads to ubiquitin mediated proteasomal degradation of this RECQL4 p.E481GfsX48 protein and failure of DNA replication. When KLHDC3 is deleted, the RECQL4 p.E481GfsX48 protein is stabilised and retained on chromatin where it is sufficient to enable near normal DNA replication and cell division. Source data are available online for this figure.

Tandazo et al, 2021; Castillo-Tandazo et al, 2019). This resulted in the identification of loss of KLHDC3 as a strong suppressor, including for in vivo phenotypes. KLHDC3 is a substrate recognition component of the CRL E3 ubiquitination system (Pilcher et al, 2025; Rusnac et al, 2018; Timms et al, 2023; Yeh et al, 2021; Zhang et al, 2023). KLHDC3 targets proteins that have RxxxG C-end degron motifs, promoting their ubiquitination and degradation (Timms et al, 2023; Yeh et al, 2021; Zhang et al, 2023). Further confirming the specificity of the KLHDC3 enrichment, we failed to observe any significant enrichment of sgRNA against other closely related KLHDC family members, such as KLHDC2 or KLHDC10, or against the scaffold components that the KLHDC proteins interact with (e.g. Elongin B/D, Cullin 2). However, this rescue was specific to the design of the floxed *Recql4* allele in our mice (Fig. 6E) (Smeets et al, 2014). Although this limited the generalisability and involvement of KHLDC3 in the fundamental biological roles of RECQL4, and by extension in the pathology of RTS, our finding that minute amounts of a severely truncated RECQL4 protein fragment largely support its essential role in DNA replication allows for important general conclusions.

The data indicate that a significantly truncated RECQL4 protein, only retaining the first in-frame 480aa, was present in the rescued cells

at less than 10% of the levels of WT. The truncated protein was found, near exclusively, in the chromatin fraction of the nucleus suggesting that it remains predominantly associated with chromatin at comparable molar levels to chromatin-bound wild-type full-length RECQL4. The capacity of this residual level of RECQL4 to be functionally relevant is consistent with the results obtained in *Drosophila* where mutants that retained less than 7.0% of RECQL4 protein remained largely normal (Wu et al, 2008). This suggests that only a small fraction of the endogenous level of RECQL4 is required to support DNA replication and homeostasis. This has important implications for studies using siRNA and other efficient, but not absolute, means to reduce expression of *Recql4* (Padayachy et al, 2024; Terui et al, 2024). This is also the case for experimental systems that utilise depletion, where even trace amounts of functionally active RECQL4 may confound interpretation of results. An exception to the conclusion that at least some level of RECQL4 protein, independent of the ATP-dependent helicase activity, was necessary for DNA replication was the recent report describing the apparent normality of RECQL4 deficiency in the human osteosarcoma-derived U2OS cell line generated using CRISPR/Cas9 mediated knock-out (Padayachy et al, 2024). However, the findings of this report contrast with prior independent work generating RECQL4 null U2OS, where these cells showed significant proliferation and viability defects

(Duan et al, 2020). Notably, when we previously assessed the role of *Recql4* in a tractable in vivo mouse osteosarcoma model (Ng et al, 2015; Walkley et al, 2008), we could not recover any tumours that were *Recql4* deficient. Additional work in mouse models, by our group and others using independent alleles, also supports the conclusion that *Recql4* is required for DNA replication in vivo in multiple cell types (Ichikawa et al, 2002; Lu et al, 2015; Ng et al, 2015; Smeets et al, 2014).

The predicted protein produced from the deleted *Recql4* allele retains the N-terminal in-frame up to aa 480, including regions that would contain the phosphorylation sites recently identified in RECQL4 to regulate baseline and dormant replication origins (Thakur et al, 2025). We could show that homozygous RECQL4 p.E481GfsX48 (the protein product from the recombined *Recql4* floxed allele) was sufficient for normal DNA replication and prevented BM failure in vivo if stabilised by the loss of KLHDC3. Our previous results showed that RECQL4 p.G522EfsX43 and RECQL4 p.R347X variants alone did not rescue this in vivo phenotype (Castillo-Tandazo et al, 2021; Castillo-Tandazo et al, 2019). The G522Efs protein product, longer in predicted region of retained RECQL4 in-frame sequence, is not stable and the R347X protein, although stable and localised in the nucleus, lacks the Zn-knuckle motif and most of its upstream region previously shown to be required for ssDNA and fork DNA substrate binding (Marino et al, 2016). In contrast, retention of a minute level of stable protein expression encoding the first 480aa of RECQL4 was able to substitute for full-length RECQL4 in many settings.

The mechanism via which the RECQL4 p.E481GfsX48 enables relatively normal DNA replication is still to be fully resolved. Our analysis indicate that this small RECQL4 fragment supports complete DNA replication, but delays completion of DNA synthesis with a shift towards late S phase and even into M phase as evidenced by EdU. However, using whole-genome replication timing analysis, we did not observe a global replication timing difference in bulk cell analysis. The differences between the flow cytometry measurements (EdU/Ki67) and the normal replication profiles could be due to a lower origin firing efficiency genome-wide, while the relative timing is preserved, and/or to increased variation between individual cells, with each cell having a different set of origins firing later or less efficiently or not firing at all. Our bulk replication timing analysis is not an absolute measurement and would not reveal such changes. Single-cell or molecule approaches may be able to resolve this, however, those are also limited in their accuracy or comprehensiveness at assessing a large range of origins, both early and late, and could equally well miss any such subtle effects. Recent work demonstrated that RECQL4 was required for replication origin choice, and the choice between dormant vs. baseline origins and the replication stress response (Thakur et al, 2025). The work proposed that the RECQL4 helicase domain normally acts to limit how many origins are initiated and that RECQL4 acts to promote the efficient eviction of DONSON from CMG (Terui et al, 2024). The increased amount of loaded MCM3 observed in G1/early S-phase suggests increased licensing, possibly as a compensatory mechanism for slower DNA replication in the absence of the helicase domain. An alternative explanation would be that inefficient replication initiation caused by the RECQL4 truncation simply results in a slower clearance of the pre-replication complex (MCM2-7) from the DNA and therefore a relative accumulation of MCM3 in early S-phase (Polasek-Sedlackova et al, 2022).

Remarkably, the low levels of RECQL4 p.E481GfsX48 product of the *Recql4*^Δ allele in the absence of KLHDC3 were fully sufficient to support

adult haematopoiesis and prevent BM failure resulting from loss of *Recql4* in vivo (Smeets et al, 2014). In contrast, while it significantly extends the embryonic viability of *Recql4*-mutated embryos, it does not support full completion of embryonic development. This difference in the extent of in vivo rescue may reflect the need for highly proportional and synchronised cell proliferation during embryonic development that may be less tolerant of altered, even subtly, replication timing than the more asynchronous process of haematopoiesis. The unanticipated generation of a highly specific C-terminal degron sequence as favoured by KLHDC3 could not have been predicted when the *Recql4*^fl allele was made, as KLHDC3 and the C-end degron model had not been described at the time (Koren et al, 2018; Lin et al, 2018; Smeets et al, 2014; Timms and Koren, 2020). This would mean that in the *Recql4*^fl/R347X cells used for the original screen, the RECQL4 p.E481GfsX48 product of the *Recql4*^Δ allele was responsible for mediating the rescue even in the presence of the p.R347X protein product. Our findings emphasise that the nature and basis of genetic interactions observed in genome-wide screens, both in vitro as here but increasingly in vivo, needs to be carefully validated and considered to demonstrate that it is a specific finding related to biological function, not one related to the specific model or genetic modifications.

Another key finding of this study is that using a genome-wide screening we were unable to identify any additional loss of function alleles, aside from KLHDC3 and the mutant RECQL4 protein product mechanism described, that could rescue the proliferation and viability defects of cells expressing a RECQL4 mutant protein similar to those reported in patients with RTS. This result indicates that there are unlikely to be monogenic loss of function rescue alleles for RECQL4 mutation, suggesting that RECQL4 is an essential and non-redundant regulator of DNA replication and viability. Therefore, it is improbable that there will be genetic means to bypass the molecular consequences that occur when RECQL4 is mutated as in RTS patients. This has significant implications for how we consider future therapy for RTS patients with RECQL4 mutations and indicates that direct correction of the mutations in RECQL4 is likely the most feasible path to treating or curing these patients. Heterozygosity for RECQL4 mutation is well tolerated, based on the absence of phenotypes in the parents of RTS patients, suggesting correction of only one allele of RECQL4 is likely to be significantly beneficial. The rapid advances in the potential of genetic corrective therapy, such as CRISPR or redirected RNA editing, offer potential pathways to explore. Collectively the present results, together with our previous work, do not support an essential function for the ATP-dependent helicase activity of RECQL4 in DNA replication.

## Methods

**Reagents and tools table**

| Reagent/Resource | Reference or Source | Identifier or Catalog Number |
|---|---|---|
| **Experimental models** | | |
| C57BL/6-*Recql4*^R347X (*M. musculus*) | (Castillo-Tandazo et al, 2019). Available from the Australian Phenome Bank (APB) | *Recql4*^m2Anu/AnuCrwApb APB ID#7986 MGI:6342636 |
| C57BL/6-*Recql4*^G522EfsX43 (*M. musculus*) | (Castillo-Tandazo et al, 2019). Available from the APB | *Recql4*^tmSCrw APB ID#7985 MGI:6342634 |

| Reagent/Resource | Reference or Source | Identifier or Catalog Number |
|---|---|---|
| C57BL/6-*Recql4*<sup>fl/fl</sup> (*M. musculus*) | (Smeets et al, 2014). Taconic Artemis GmbH, Cologne, Germany; Available from APB | *Recql4*<sup>tm2272Arte</sup> APB ID#7263 MGI:5645319 |
| *Rosa26*-CreER<sup>T2</sup> (B6.129-Gt(ROSA) 26Sor<sup>tm1(cre/ERT2)Tyj</sup>/J (*M. musclus*) | The Jackson Laboratory | Stock Number: 008463 |
| C57BL/6-*Klhdc3*<sup>fl/fl</sup> (*M. musculus*) | This paper; Walkley lab, generated by Monash Genome Modification platform. This study. Available from Carl Walkley or APB. | *Klhdc3*<sup>em2Crwy/Apb</sup> APB ID#10400 |
| BHK-HM5 cells (GM-CSF producing cells) | (Gupta et al, 2014) | Prof. S Collins (deceased), Fred Hutchinson Cancer Research Center, Seattle, USA |
| HEK-293T cells (*H. sapiens*) | ATCC | CRL-3216 |
| HEK-293T *KLHDC3*<sup>-/-</sup> cells (*H. sapiens*) | This study | N/A |
| Hoxb8 *R26*CreER<sup>T2</sup> *Recql4*<sup>fl/+</sup>, *Recql4*<sup>fl/R347X</sup>, *Recql4*<sup>G522EfsX43</sup> and *Recql4*<sup>fl/fl</sup> cells (*M. musculus*) | (Castillo-Tandazo et al, 2019; Castillo-Tandazo et al, 2021) | N/A |
| Cas9 Hoxb8 *R26*CreER<sup>T2</sup> *Recql4*<sup>fl/+</sup> and *Recql4*<sup>fl/R347X</sup> cells (*M. musculus*) | This study | N/A |
| **Recombinant DNA** | | |
| pMSCVneo-Flag-mHoxb8 | (Wang et al, 2006) | Prof. Mark Kamps, UCSD, USA |
| pLentiCas9-Blast | Addgene/Feng Zhang | 52962 |
| pLentiGuide-Puro | Addgene/Feng Zhang | 52963 |
| Mouse sgRNA library Brie in pLentiGuide-Puro | Addgene/John Doench, David Root | 73633 |
| pLentiCRISPRv2-Puro | Addgene/Brett Stringer | 98290 |
| pLentiCRISPRv2-Hygro | Addgene/Brett Stringer | 98291 |
| pLenti-SFFV-IRES-Puro-V5-APEX2-DN-mKlhdc3 and plasmids derived thereof | Twist Biosciences and this study | pTwist Lenti SFFV Puro WPRE |
| pAIP (pLenti-SFFV-IRES-Puro) | Addgene/Jeremy Luban | Cat#74171 |
| pAIP and pLenti-SFFV-SV40-Blast-3xFlag-hKLHDC3 | This study | N/A |
| pLenti-SFFV-SV40-Blast and pLenti-SFFV-SV40-Zeo-mCherry-mRecql4 | This study and (Castillo-Tandazo et al, 2021) | N/A |
| pLenti-GPS (pDEST-GPS) | (Lin et al, 2018) and (Huang et al, 2023) | Prof. Hsueh-Chi S. Yen, IMB, Taiwan |
| pLenti-GPS-mRecql4<sup>V359-E481EfsX48</sup> | This study | N/A |

| Reagent/Resource | Reference or Source | Identifier or Catalog Number |
|---|---|---|
| **Antibodies** | | |
| Rat anti-h/mCD44 PE-Cy7 | Thermo Fisher Scientific | 25-0441-82 |
| Rat anti-mRECQL4 | WEHI Antibody Services, Melbourne (Castillo-Tandazo et al, 2019) | 18/18-3B10-1-1 (antigen: N-200 mRECQL4) |
| Mouse anti-β-Actin | Sigma-Aldrich | A1978 |
| Rabbit anti-Histone H3 | Abcam | ab1719 |
| Rabbit anti-SC-35 | Abcam | ab204916 |
| Goat anti-rat IgG secondary antibody HRP | Thermo Fisher Scientific | 31470: RRID: AB228356 |
| Goat anti-mouse IgG secondary antibody HRP | Thermo Fisher Scientific | 31444: RRID: AB228321 |
| Rat anti-h/mKi-67 PE-Cy7 | Thermo Fisher Scientific | 25-5698-82 |
| Mouse anti-MCM3 | Bio-Strategy | sc-390480 |
| Goat anti-mouse IgG secondary F(ab')<sub>2</sub> fragment Alexa Fluor 594 | Cell Signaling Technology | 8890 |
| Rat anti-h/mCD45R (B220) APC-eFluor 780 | Thermo Fisher Scientific | 47-0452-82 |
| Rat anti-mCD11b (Mac-1) PE | Thermo Fisher Scientific | 12-0112-82 |
| Rat anti-mLy-6G/Ly-6C (Gr-1) PE-Cy7 | Thermo Fisher Scientific | 25-5931-82 |
| Rat anti-mF4/80 APC | Tonbo/Cytek Biosciences | 20-4801-U100 |
| Rat anti-mCD4 violetFluor 450 | Tonbo/Cytek Biosciences | 75-0042-U100 |
| Rat anti-mCD8a PerCP-Cy5.5 | Thermo Fisher Scientific | 45-0081-82 |
| Rat anti-mTer-119 PE | Thermo Fisher Scientific | 12-5921-83 |
| Rat anti-mCD71 APC | Thermo Fisher Scientific | 17-0711-82 |
| Rat anti-h/mCD44 PE-Cy7 | Thermo Fisher Scientific | 25-0441-82 |
| Rat antimLy6A/E (Sca-1) PerCP-Cy5.5 | Thermo Fisher Scientific | 45-5981-82 |
| Rat anti-h/mCD117 (c-Kit) APC-eFluor 780 | Thermo Fisher Scientific | 47-1171-82 |
| Rat anti-mCD150 (SLAM) PE | BioLegend | 115904 |
| Hamster anti-mCD48 PE-Cy7 | BD Biosciences | 560731 |
| Rat anti-mCD34 eFluor 660 | Thermo Fisher Scientific | 50-0341-82 |
| Rat anti-mCD16/CD32 eFluor 450 | Thermo Fisher Scientific | 48-0161-82 |
| Rat anti-mCD2 Biotin | Thermo Fisher Scientific | 13-0021-82 |
| Hamster anti-mCD3e Biotin | Thermo Fisher Scientific | 13-0031-85 |
| Rat anti-mCD4 Biotin | Thermo Fisher Scientific | 13-0041-85 |

| Reagent/Resource | Reference or Source | Identifier or Catalog Number |
|---|---|---|
| Rat anti-mCD5 Biotin | Thermo Fisher Scientific | 13-0051-85 |
| Rat anti-mCD8a Biotin | Tonbo/Cytek Biosciences | 30-0081-U500 |
| Rat anti-h/mCD45R (B220) Biotin | Thermo Fisher Scientific | 13-0452-85 |
| Rat anti-mLy-6G/Ly-6C (Gr-1) Biotin | Thermo Fisher Scientific | 13-5931-85 |
| Rat anti-mCD11b (Mac-1) Biotin | Thermo Fisher Scientific | 13-0112-85 |
| Rat anti-mTer-119 Biotin | Thermo Fisher Scientific | 13-5921-85 |
| **Oligonucleotides and other sequence-based reagents** | | |
| sgRNA sequences and PCR primers | This study | See Appendix |
| Genotyping primers | This study | See Appendix |
| **Chemicals, Enzymes and other reagents** | | |
| Streptavidin Brilliant Violet-605 | BD Biosciences | 563260 |
| Tamoxifen Citrate (for chow) | Selleckchem | S1972 |
| 4-Hydroxytamoxifen (for in vitro use) | Selleckchem or Merck | S7827 or 579002 |
| Recombinant mouse SCF | PeproTech | 250-03 |
| Recombinant mouse IL-3 | PeproTech | 213-13 |
| Recombinant human IL-6 | Amgen (gift) | N/A |
| IMDM | Sigma-Aldrich | I3390-500ML |
| Foetal Bovine Serum (FBS, Australian origin; not heat inactivated) | Bovogen Biologicals | FBSAU-2007A |
| Blasticidin | InvivoGen | Ant-bl-05 |
| Puromycin | Thermo Fisher Scientific | A1113803 |
| Hygromycin | Thermo Fisher Scientific | 10687-010 |
| Gentra Puregene Cell Kit | Qiagen | 158767 |
| ISOLATE II Plasmid Mini Kit | Bioline | BIO-52057 |
| T7 endonuclease I | New England Biolabs | M0302S |
| ISOLATE II RNA Mini Kit | Bioline | BIO-52073 |
| Tetro cDNA Synthesis Kit | Bioline | BIO-65043 |
| SYBR Select Mastermix | Thermo Fisher Scientific | 4472908 |
| HALT protease and phosphatase inhibitor | Thermo Fisher Scientific | 78440 |
| 4-(2-Aminoethyl)-benzenesulfonyl fluoride hydrochloride (AEBSF) | Sigma-Aldrich | A8456 |
| MG132 proteasome inhibitor | Selleckchem | S2619 |

| Reagent/Resource | Reference or Source | Identifier or Catalog Number |
|---|---|---|
| Pierce BCA protein assay kit | Thermo Fisher Scientific | 23227 |
| Amersham ECL Prime Western blotting detection reagents | Cytiva Life Sciences | RPN2232 |
| Click-iT® EdU Alexa Fluor® 488 Flow Cytometry Assay Kit | Life Technologies Thermo Fisher Scientific | C10425 |
| AZdye 488-Azide (structural analogue of AF488-Azide) | Click Chemistry Tools | 1275 |
| Alexa Fluor 647-Azide | Lumiprobe Life Science Solutions | A6830 |
| DAPI | Molecular Probes Thermo Fisher Scientific | D1306 |
| AffinityScript cDNA Synthesis Kit | Agilent Technologies | 600559 |
| Ex Taq DNA polymerase | Takara | RR001 |
| **Software** | | |
| TIDE | https://tide.nki.nl/ (Brinkman et al, 2018) | |
| ICE | https://www.synthego.com/products/bioinformatics/analysis (Conant et al, 2022) | |
| CRISPRBetaBinomial (CB²) | https://liuzlab.org/software/crisprbetabinomial-cb2/ (Jeong et al, 2019) | |
| biomaRt | https://bioconductor.org/packages/release/bioc/html/biomaRt.html | |
| ggplot2 | https://ggplot2.tidyverse.org/ | |
| Fiji (ImageJ2) | https://imagej.net (Schindelin et al, 2012) | |
| FlowJo™ Software, version 9 or 10 | Becton, Dickinson and Company; 2023 https://www.flowjo.com | |
| cellSens Software | Olympus – Evident Scientific https://evidentscientific.com | |
| GraphPad Prism Software, version 9 or 10 | GraphPad; San Diego, CA, USA https://www.graphpad.com | |
| **Other** | | |
| Mouse chow containing 400 mg/kg tamoxifen citrate; irradiated | Compounded by Specialty Feeds, Perth, Australia | |
| Brie CRISPR knockout pooled library virus aliquots | Made by the Victorian Centre for Functional Genomics, Peter MacCallum Cancer Centre, Australia | RRID:SCR_025582 |

| Reagent/Resource | Reference or Source | Identifier or Catalog Number |
|---|---|---|
| Sequencing services (various) | Novogene, Singapore | |
| Whole-genome sequencing | BGI Tech Solutions, Hong Kong (150PE, DNBSEQ) | |
| Sanger sequencing | AGRF, Parkville or MicroMon, Clayton, Australia | |

## Preparation of experimental models and subject details

### Ethics statement

All animal experiments were performed according to the National Health and Medical Research Council Act 1992 and the Australian Code for the Care and Use of Animals for Scientific Purposes (2013). The procedures were approved by the Animal Ethics Committee, St. Vincent's Hospital, Melbourne, Australia (#011/15, and 015/17). Animals were euthanased by $CO_2$ asphyxiation or cervical dislocation.

## Mice

All animals were housed at the Bioresources Centre (BRC) located at St. Vincent's Hospital, Melbourne. Mice were maintained and bred in microisolators under specific pathogen-free conditions with 12-h light-dark cycle and with autoclaved food and acidified water provided ad libitum. The Recql4R347X and Recql4G522Efs mutations and Recql4fl/fl mice (C57BL/6-Recql4tm2272Arte) have been all been previously described (Castillo-Tandazo et al, 2021; Castillo-Tandazo et al, 2019; Ng et al, 2015; Smeets et al, 2014). Rosa26-CreERT2 mice on a C57Bl/6 background were purchased from The Jackson Laboratory (B6.129-Gt(ROSA)26Sortm1(cre/ERT2)Tyj/J, Stock Number: 008463) and have been previously described (Smeets et al, 2014). All genotyping was performed as previously described (Castillo-Tandazo et al, 2021; Castillo-Tandazo et al, 2019; Ng et al, 2015; Smeets et al, 2014). ENU mutants were outcrossed at least 6 times and assessed across multiple generations to eliminate effects of any additional mutations. All lines were on a backcrossed C57Bl/6 background.

Klhdc3 conditional (Klhdc3fl/fl) mice were generated by the Monash Genome Modification Platform, Monash University by CRISPR/Cas9 mediated insertion of loxP elements flanking Klhdc3 exons 4 and 5. A ssDNA repair template was prepared for the required sequence to insert. Cas9 protein (80 ng/µl; Integrated DNA Technologies (IDT)) was incubated with sgRNA (80 ng/µl; crRNA and tracrRNA; IDTDna) to generate RNP. The RNP and the ssDNA repair template (50 ng/ml) were microinjected into C57BL/6 J zygotes at the pronuclei stage. Microinjected zygotes were transferred into the uterus of pseudo pregnant F1 females. Correct insertion was screened by genomic DNA PCR and then digital droplet PCR (Bio-Rad). Correctly targeted males were identified and bred to C57BL/6 J females to ensure germ-line transmission. During screening of progeny, a male with a deletion encompassing the loxP flanked region was identified (referred to as Klhdc3+/-). The male was bred to C57BL/6 J females to ensure germ-

line transmission. All lines were confirmed to have the expected mutations/inserted elements by Sanger sequencing of the loxP elements and the deletion event, respectively. Genotyping primers and product sizes as described in the Appendix Table S1 and a full description of the Klhdc3 targeted allele and phenotype is provided in a separate report (Buco et al, 2026).

Where indicated, tamoxifen containing diet was prepared by Specialty Feeds (Perth, Australia) containing 400 mg/kg tamoxifen citrate (Selleckchem) in a base of standard mouse chow (irradiated) (Smeets et al, 2014). When used, tamoxifen containing chow was fed ad libitum for the duration of treatment.

## Generation of retrovirus/lentivirus

Retrovirus or lentivirus was produced using transient transfection of HEK293T cells as previously described (Heraud-Farlow et al, 2024; Smeets et al, 2014; Xu et al, 2022). Virus containing supernatant was collected at 48 and 72 h in 1.0 mL aliquots and stored at −80 °C.

## Generation of Hoxb8 immortalised and Cas9 expressing cell lines

### Hoxb8 immortalised cell lines

Bone marrow cells were collected from R26-CreERT2 Recql4fl/+ (control, became Recql4 heterozygous after tamoxifen treatment), R26-CreERT2 Recql4fl/fl (Recql4 deficient) and R26-CreERT2 Recql4fl/R347X (expressed only the Recql4 R347X allele after tamoxifen treatment), Ficoll density gradient separated and cultured for 48 h in complete IMDM (20% FBS, 1% Penicillin/Streptomycin, 1% glutamine) containing recombinant mouse stem cell factor (rmSCF, 50 ng/mL, Peprotech), recombinant mouse interleukin 3 (rmIL-3, 10 ng/mL, Peprotech) and recombinant human interleukin 6 (rhIL-6, 10 ng/mL, Amgen). After 48 h culture, $1 \times 10^6$ cells were spin-infected with Hoxb8 retrovirus (Wang et al, 2006; Xu et al, 2022) (Hoxb8 plasmids generously provided by Mark Kamps, UCSD) and polybrene at $1100 \times g$ for 90 min. After 48 h, the non-adherent cells were passaged into complete IMDM (10% FBS, 1% Penicillin/Streptomycin, 1% glutamine) with rmSCF, rmIL-3, rhIL-6 and 1% granulocyte-macrophage colony-stimulating factor (GM-CSF) conditioned medium (from BHK-HM5 cell conditioned medium) (Xu et al, 2022). After 1 to 2 passages, cells were cultured in complete IMDM with only 1% GM-CSF conditioned media. The cells were fully immortalised after 4 weeks and were GM-CSF dependent.

### Cas9 expressing cell lines

Following establishment, Hoxb8 immortalised cell lines were infected with Cas9 expressing lentivirus (p.lentiCas9-Blast was a gift from Feng Zhang; Addgene plasmid # 52962) (Sanjana et al, 2014) by spin-infection at $1100 \times g$ for 90 min. The infected cells were then cultured in media (IMDM, 10%FBS, 1% GM-CSF and 3 µg/mL blasticidin) for 2 weeks to select for a Cas9-expressing population. To assess the target editing efficiency of the Cas9 Hoxb8 R26CreERT2 Recql4fl/+ and R26CreERT2 Recql4fl/R347X cell lines, stable Cas9 cells were transduced with pLentiGuide (VCFG, from Addgene) expressing sgRNA targeting CD44, a cell surface marker highly expressed on all immortalised Hoxb8 cells. Forty-eight hours after infection, puromycin (0.25 µg/mL) was added to select for pLentiGuide sgCd44 expression. Four days

later, $0.5–1 \times 10^6$ cells were stained with a PE-Cy7 conjugated anti-CD44 antibody (1:400) and analysed by flow cytometry using the FACS BD LSR II Fortessa system.

Oligonucleotide sequences (sgRNAs) and genotyping primer sequences are provided in Appendix Table S2. Cell lines described in this study were routinely tested for mycoplasma by PCR analysis (performed by the Genotyping Core, Peter MacCallum Cancer Centre). Authentication testing was not performed.

## CRISPR knockout pooled library screen

For loss of function rescue screening, $80 \times 10^6$ Cas9 Hoxb8 $R26$CreER$^{T2}$ $Recql4^{fl/+}$ and $R26$CreER$^{T2}$ $Recql4^{fl/R347X}$ cell lines were infected with the Brie CRISPR knockout pooled library (Mouse Brie CRISPR knockout in lentiGuide-Puro pooled library was a gift from David Root and John Doench (Addgene #73633; http://n2t.net/addgene:73633; RRID:Addgene_73633) (Doench et al, 2016). Brie CRISPR knockout pooled library virus was generated by the Victorian Centre for Functional Genomics (RRID:SCR_025582) and provided as aliquots (Heraud-Farlow et al, 2024; Xu et al, 2022). The number of cells infected was calculated by multiplying the number of guides in the library (78,637) by the desired number of cells for each guide at the start of the experiment (aiming for 400 copies per sgRNA), divided by the multiplicity of infection required (MOI = 0.3). The infection mix was centrifuged at $1100 \times g$ and 25 °C for 1.5 h and then returned to the 37 °C incubator. At the end of the day, each flask was topped up with 30 mL of pre-warmed medium (IMDM, 10%FBS, 1% GM-CSF and 3 μg/mL blasticidin). On day 3, cells were counted and expanded to T175 flasks containing 125 mL of medium at a concentration of $1 \times 10^6$ cells/mL. Puromycin was added at a dose of 0.25 μg/mL. On day 4, cells were further diluted to reach a concentration of $3.3 \times 10^5$ cells/mL of medium plus puromycin. On day 7, puromycin was removed from the medium and cells were re-seeded at $2 \times 10^5$ cells/mL in tamoxifen containing medium (400 nM/mL) to induce CreER-mediated recombination (Day 0 designated the day that tamoxifen was added). During the next 14 days, cells were passaged every 2 or 3 days at a density to maintain at least 400 cells per sgRNA in the presence of tamoxifen, which corresponded to 2 x T175 flasks per cell line with 250 mL total at a concentration of 1 or $2 \times 10^5$ cells/mL. Additionally, at each indicated timepoint (Day 0, 4 and 9 post tamoxifen addition), pellets of 2 and 25 million cells were kept for DNA recombination analysis and gDNA isolation for library sequencing, respectively (Heraud-Farlow et al, 2024; Xu et al, 2022).

For synthetic lethal screening (Xu et al, 2022), the experimental procedure was similar to that described above for loss of function rescue screening with the following modifications. The Brie CRISPR knockout sgRNA pooled library virus used. The cell lines used for synthetic lethal screening were Cas9 Hoxb8 $R26$CreER$^{T2}$ $Recql4^{\Delta/\Delta}$ sg$Klhdc3$ clone 185K1B12 (see generation of clonal cell lines) and clone 185K1B12 complemented with mCherry-$Recql4$. $130 \times 10^6$ cells per cell line were infected with the Brie CRISPR sgRNA library at an MOI of 0.3 to have 500 cells per sgRNA and cells were maintained at ≥500 cells per sgRNA by passaging $45 \times 10^6$ cells at 2 or $~0.8 \times 10^5$ cells/mL every two or three days. At least 2 pellets of $25 \times 10^6$ cells each cell line were collected on days 0, 2, 5 and 8 (Day 0 being the third day after puromycin selection) for gDNA to assess sgRNA copy number.

## CRIPSR screen sample preparation and sgRNA library sequencing

Genomic DNA was extracted from the cell pellets using the Gentra Puregene kit (Qiagen). DNA was quantified on a Nanodrop spectrophotometer. DNA libraries were generated by PCR amplification of the integrated sgRNA constructs (as described in https://portals.broadinstitute.org/gpp/public/resources/protocols). The resultant libraries were generated and sequenced on the Illumina platform using 150 bp paired-end reads by Novogene (Singapore).

CRISPR screen analysis was performed using the CRISPRBeta-Binomial package (CB$^2$) (v 1.3.0) (Heraud-Farlow et al, 2024; Jeong et al, 2019; Li et al, 2015; Xu et al, 2022). Briefly, raw reads were mapped to the broadgpp_brie_crispr library with run_sgrna_quant function and read counts were normalised with the get_CPM function. Gene level statistics were calculated using the measure_gene_stats function. Gene level statistics were calculated using measure_gene_stats function. Annotation with ENSEMBL was performed with biomaRt (Durinck et al, 2009a; Durinck et al, 2009b). Plots were generated using ggplot2 (Wickham, 2016). Core essential genes version 2 (CEG2) was obtained from CEG2 Supplemental Table 2 "List of the human core essential gene set version 2" (Hart et al, 2017).

## Loss of function screen validation

For loss of function rescue validation, four individual sgRNAs targeting $Klhdc3$ were selected and cloned into the pLentiGuide, pLentiCrispr puromycin or pLentiCrispr hygromycin vectors (sgRNA from IDTDna; Appendix Table S2). mCherry-tagged codon optimised mouse $Recql4$ cDNA (Castillo-Tandazo et al, 2021) was cloned into pLenti SFFV SV40 blasticidin or zeocin vectors (derived from pTwist Lenti SFFV Puro WPRE, Twist Bioscience). Plasmid DNA was purified using the Isolate II Plasmid Mini kit (Bioline) and quantified using the Nanodrop spectrophotometer system for use in viral production and subsequent cell proliferation assays. The sgRNA plasmids targeting $Klhdc3$ and control sgRNAs were individually transfected into HEK293T cells and transduced in Cas9 Hoxb8 $R26$-CreER$^{T2}$ $Recql4^{fl/R347X}$ cells (Re-validated with two guides in fl/+, fl/R and fl/fl). Three days after transduction, puromycin was added at a dose of 0.25 μg/mL for four days for positive selection. After generating cells expressing lentivirus containing sgRNAs targeting the $Klhdc3$ gene, cells received tamoxifen (400 nM/mL) for 14 days in a cell proliferation assay. Every two to three days, cells were counted, and the corresponding volume for the number of cells needed to re-seed $1–2 \times 10^5$ cells/mL were centrifuged, the media discarded, and the cell pellet was resuspended in 5 mL fresh tamoxifen containing medium. Cells were treated for 14 days, and at each time point, cell pellets were collected for DNA recombination/$Recql4$ deletion analysis. As a control, untreated cells were kept in parallel and seeded at the same concentration.

CRISPR-Cas9 cutting and KO efficiency was determined following PCR amplification of the targeted genomic region by T7 Endonuclease I assay (T7E1, NEB M0302) according to the NEB protocol and by Sanger sequencing plus TIDE analysis (tide.nki.nl) (Brinkman et al, 2018). Reduced expression levels were confirmed by RT-qPCR. All primers are described in the Appendix Table S1.

For synthetic lethal validation, three guides targeting *Ccnf* and *Mcm10* (sequences in Appendix Table S2) were cloned into pLentiCrispr puromycin vectors and transduced in Cas9 Hoxb8 *R26*-CreER^T2 *Recql4*^fl/fl control cells and *Recql4*^Δ/Δ sg*Klhdc3* clones 185K1B12 (-/+ mCherry-*Recql4*) and 186K1E1 (see below). Two days after infection, puromycin was added to the cells for 3 days to eliminate non-infected cells. From then on cells were counted and passaged every two to three days for 11 days, and cell pellets were collected for gDNA extraction and assessment of CRISPR-induced knockout frequency by TIDE (Brinkman et al, 2018) and ICE (accessed via Synthego) (Conant et al, 2022).

## Generation of clonal cell lines

Immortalised mouse myeloid *Recql4*^Δ/Δ sg*Klhdc3* clonal cell lines were generated at the end of a 14-day tamoxifen treatment as described in Heraud-Farlow et al (2024). After Sanger sequencing, mutant clones containing homozygous indels predicted to result in loss of function, were selected for further experiments. Specifically, clone 185K1B12 containing a 38 bp deletion in exon 7 predicted to result in KLHDC3 p.I216LfsX7 and clone 186K1E1, with a 391 bp deletion in intron 5- exon 7, resulting in KLHDC3 p.A150X.

Human HEK293T cells transduced with pLentiCrispr sg*KLHDC3* were single-cell cloned by limiting dilution in 96-well multi-well plates after 14 days of hygromycin selection, then expanded and analysed as above. Clones 293T K1E3, K2D8 and K2D10 are described in Fig. EV4.

## Mouse KLHDC3 re-expression experiments

Mouse codon-optimised dominant negative (DN) *Klhdc3* was obtained from Twist Bioscience as a clonal gene in pLenti SFFV IRES puromycin with an N-terminal V5-APEX2 fusion, a L338A mutation in the BC box and S354A, C355A, L356A and P357A mutations in the CUL2 box. These mutations have been shown to abolish Elongin B/C binding in other BC box proteins or CUL2 binding in KLHDC3 specifically (Mahrour et al, 2008). Twist gene fragments were subsequently used to create a codon-optimised wild-type (WT) and a C-terminally truncated (p.S335X) *Klhdc3*, lacking the BC box, CUL2 box and last 17aa. These plasmids were then used to construct V5-only and mCherry-tagged versions by standard cloning techniques.

For proliferation assays, two parental *Recql4*^fl/fl cell lines (185 and 186) and *Recql4*^Δ/Δ sg*Klhdc3* clones 185K1B12 and 186K1E1 were transduced with V5- or mCherry tagged WT, DN and ΔC-terminal KLHDC3 expressing constructs. Puromycin was added 2 days after infection for 3 days and cells were subsequently passaged and counted every two to three days to assess survival and proliferation. For western blot analysis *Recql4*^+/Δ *Klhdc3*^-/- cells were used to keep the cells alive after re-expression of functional KLHDC3 by one copy of WT *Recql4*. Cells were selected with puromycin 2 days after transduction and collected for protein as described below.

## Protein extraction and western blotting

Cell lysates were prepared in RIPA buffer (50 mM Tris, 150 mM NaCl, 1% NP-40, 0.5% sodium deoxycholate, 0.1% SDS, pH 8.0) plus 1x Halt™ (Thermo Fisher Scientific, 78440) protease and phosphatase inhibitors. Myeloid cells were pre-treated with 4-(2-aminoethyl) Benzenesulfonyl fluoride hydrochloride (AEBSF, 0.5 mM) for 30 min in culture medium before lysis or alternatively $2 \times 10^6$ cells were prepared directly in 100 μL NuPAGE LDS sample buffer (1x in RIPA buffer plus inhibitors) and heated at 70 °C for 10 min. Where indicated, the proteasome inhibitor MG132 was added to the culture medium at a concentration of 0.25 μM for 2 h. Chromatin-bound proteins were isolated from subcellular fractions according to BioProtocol 8-9-3035 (Gillotin, 2018). In brief, $6–10 \times 10^6$ myeloid cells were pre-treated with AEBSF as above, then washed with ice-cold PBS and centrifuged at 4 °C, followed by sequential extraction of cytoplasmic proteins in E1 buffer (50 mM HEPES-KOH, 140 mM NaCl, 1 mM EDTA, 10% glycerol, 0.5% NP-40, 0.25% Triton X-100, 0.5 mM TCEP), nuclear proteins in E2 buffer (10 mM Tris-HCl, 200 mM NaCl, 1 mM EDTA, 05 mM EGTA) and chromatin bound proteins in E3 buffer (500 mM Tris-HCl, 500 mM NaCl). All lysates were sonicated in a Bioruptor sonicating water bath at 4 °C, with 5 min, 30 s on/off pulses on power high. Protein concentrations were measured with Pierce BCA assay kit and, unless otherwise indicated, 25 μg protein extract was separated on pre-cast NuPAGE™ BOLT 8% Bis-Tris poly-acrylamide gels (Invitrogen) and transferred onto Immobilon-P PVDF membranes (Merck Millipore). Membranes were blocked with 5% milk in TBST and incubated overnight with rat monoclonal anti-mouse RECQL4 antibody (clone 3B10, made by WEHI Antibody Services, Melbourne) (Castillo-Tandazo et al, 2019), mouse anti-β-Actin (Sigma-Aldrich, A1978), rabbit anti-Histone H3 (Abcam, ab1719) or rabbit anti-SC-35 (Abcam, ab204916). Membranes were then probed with HRP-conjugated goat anti-rat (Thermo Fisher Scientific, 31470) or anti-mouse (Thermo Fisher Scientific, 31444) secondary antibodies and visualized using ECL Prime Substrate (Amersham). Images were acquired on X-ray film (Fujifilm) or digitally with the iBright Imaging System (Thermo Fisher Scientific) and analysed using ImageJ or iBright software.

## Cell cycle flow cytometry

For DNA replication analysis, cells were treated with 10 μM EdU (Life Technologies) for 30 min at 37 °C. Cells were then harvested, washed in PBS with 1% BSA, and fixed with 2% paraformaldehyde for 15 min at room temperature in the dark. Cells were washed once with PBS 1% BSA and stored short-term at 4 °C. To detect incorporated EdU, cells were processed as described by the manufacturer (EdU labelling kit, Life Technologies) using Alexa Fluor 488 Azide or Alexa Fluor 647 Azide. Cells were subsequently stained with anti-Ki-67 PECy7 antibody (eBioscience 25-5698-82) for 30 min, then washed with PBS + 1% BSA and resuspended in 1 μg/mL DAPI (Molecular Probes). The various stages of the cell cycle were analysed by flow cytometry as described (Vignon et al, 2013).

MCM loading was determined according to Matson et al (Matson and Cook, 2020; Matson et al, 2017). In short, cells were pulsed with EdU as above, pre-extracted with CSK buffer (10 mM PIPES pH 7.0, 300 mM sucrose, 100 mM NaCl, 3 mM $MgCl_2$) + 0.5% (V/V) Triton X-100 + protease/phosphatase inhibitors for 5 min on ice, followed by fixation in 4% PFA and washing with PBS 1% BSA. EdU was detected as above and cells were then incubated with mouse anti-MCM3 antibody (Santa Cruz sc-390480) in 0.1% NP40 (Sigma-Aldrich) - PBS 1% BSA. After secondary antibody-

staining with anti-mouse Alexa Fluor 594 F(ab')2 fragments (Cell Signaling Technology #8890), cells were kept in 0.1% NP40 - PBS 1% BSA supplemented with DAPI (1 μg/mL) and RNase A (100 μg/mL). Cells were analysed by flow cytometry using the FACS BD LSR II Fortessa system and FlowJo software version 10 (BD Bioscience).

## Peripheral blood analysis

Peripheral blood was analysed on a hematological analyser (Sysmex KX-21N, Roche Diagnostics). For flow cytometric analysis, red blood cells were lysed using a red blood cell lysis buffer (150 mM $NH_4Cl$, 10 mM $KHCO_3$, 0.1 mM $Na_2EDTA$, pH 7.3).

## Flow cytometry analysis of mouse tissues

Femurs were flushed (2 femurs in 2 mL PBS/2%FBS), spleens (5 mL PBS/2%FCS) and thymus (2 mL PBS/2%FCS) crushed, and single-cell suspensions were prepared in PBS containing 2% FBS. Antibodies against murine B220 (APC-eFluor780), CD11b/Mac-1 (PE), Gr1 (PE-Cy7), F4/80 (APC), CD4 (violetFluor450) and CD8a (PerCP-Cy5.5), Ter119 (PE), CD71 (APC), CD44 (PE-Cy7), Sca-1 (PerCP-Cy5.5), c-Kit (APC-eFluor780), CD150 (PE), CD48 (PE-Cy7), CD34 (eFluor660), CD16/32 (eFluor450) and biotinylated antibodies (CD2, CD3e, CD4, CD5, CD8a, B220, Gr-1, CD11b/Mac1, Ter-119) were used. The biotinylated antibodies were detected with streptavidin-conjugated Brilliant Violet-605 (Liddicoat et al, 2015; Singbrant et al, 2011; Smeets et al, 2014). 30,000–500,000 cells were acquired on a BD LSRII Fortessa and analysed with FlowJo software Version 9 or 10.0 (Treestar).

## Global protein stability (GPS) assays

The empty GPS reporter construct and constructs containing USP49, PPP1R15A, NS3, NS3GA and NS8 were a kind gift from Prof. Hsueh-Chi S. Yen (IMB, Academia Sinica, Taipei, Taiwan) and have been described before (Lin et al, 2018; Yen and Elledge, 2008; Yen et al, 2008). The original pDEST-GPS Gateway vector (Huang et al, 2023) was adapted to allow for standard cloning by replacing a 1721 bp BsrG1 fragment with a BsrG1-NotI-coCenpw-XhoI-BsrGI gene fragment. The cDNA encoding the predicted last 172aa of the recombined/deleted *Recql4* allele was synthesised from *Recql4*$^{\Delta/\Delta}$ *sgKlhdc3* derived mRNA and cloned between NotI and XhoI in-frame with the GFP. Three clones containing the correct C-terminal sequence of this short product (S4, S5 and S6) were transduced individually in HEK293T WT and KLHDC3 KO cells. Human codon-optimised *KLHDC3* was obtained from Twist Bioscience as a gene fragment with an N-terminal 3xFlag tag and cloned into the pAIP (pLenti SFFV IRES puromycin, Plasmid #74171, Addgene) and pLenti SFFV SV40 blasticidin vectors. HEK293T KLHDC3 WT or KO cells were transduced with GPS constructs and selected with 1 μg/mL puromycin. Where indicated, KLHDC3 was re- or over-expressed by transduction with pLenti SFFV 3xFlag-coKLHDC3 and selection with 10 μg/mL blasticidin. Protein stability was visualised by microscopy on an inverted fluorescence microscope (Olympus IX81) using cellSens (Olympus Lifescience) software for image acquisition and analysed by flow cytometry using a BD LSRII Fortessa for acquisition and FlowJo

software Version 10.0 (BD Biosciences) to determine the ratio of GFP/DsRed for each reporter construct (Lin et al, 2018).

## DNA replication timing profiling by whole-genome sequencing

Extraction of genomic DNA was performed using the Gentra Puregene kit (Qiagen) following the manufacturer's instructions. The purified DNA was then sent to BGI Tech Solutions (Hong Kong, China) for library preparation and Whole Genome Sequencing (150PE, DNBSEQ). DNA replication timing profiles across the genome were obtained using the Timing Inferred from Genome Replication (TIGER) methodology as described (Bracci et al, 2023; Koren et al, 2021). We assessed the data in 10 kb windows and with the remaining settings as default parameters.

## Statistical analysis

To determine statistical significance, log-rank tests, t-tests and ordinary one-way ANOVA with multiple comparison correction were conducted in GraphPad Prism software version 9 or 10 (GraphPad, San Diego, CA, USA). Statistical significance is indicated as: $*p < 0.05$; $**p < 0.01$; $***p < 0.001$ and $****p < 0.0001$, and data are presented as mean ± SEM unless otherwise indicated. The number of samples used for each experiment is indicated in the figure panels or described in the corresponding figure legends.

For cell line experiments: No statistical method was used to pre-determine sample size required. Experiments were repeated three times independently using multiple independent sgRNA targeting different sites of the targeted gene or with biologically independent cell lines (for mouse cell lines; generated from different bone marrow donors). For animal experiments: The sample sizes were based on previous experience in similar studies with the same model and upon genotypes available at time of initiation to ensure statistical power while adhering to the 3Rs (Reduction) principle and to the National Health and Medical Research Council Act 1992 and the Australian Code for the Care and Use of Animals for Scientific Purposes (2013) and the animal numbers approved by the Animal Ethics Committee. Sample blinding was not applied; no samples were omitted from the study based on inclusion/exclusion criteria.

# Data availability

Datasets are contained in the supplemental files and related sequencing files have been deposited in GEO under accession numbers GSE273174 (screen in Fig. 1), GSE273006 (synthetic lethal screen) and GSE272599 (whole genome sequencing dataset used to calculate replication timing). PCR primer and sgRNA sequences used in the study are listed in Appendix Tables S1 and S2. All reagents are available upon request to Carl Walkley.

The source data of this paper are collected in the following database record: biostudies:S-SCDT-10_1038-S44319-026-00727-2.

## Peer review information

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

## Acknowledgements

The authors thank L Purton, JX Xu and N Hoch for comments and discussion; K Simpson, T Gulati and H Beetham (Victorian Centre for Functional Genomics, Peter MacCallum Cancer Centre, Australia) for providing Brie sgRNA lentiviral library aliquots; WEHI Antibody Facility and Monash Antibody Technology Facility (MATF), Monash University for purification of RECQL4 antibody; Monash Genomics & Bioinformatics Platform, Monash University and AGRF for Sanger sequencing and plasmid sequencing services; BGI and Novogene for sequencing services; H-C S Yen (Academia Sinica Institute of Molecular Biology, Taipei) for GPS reporter plasmids; M Kamps (UCSD, USA) for HoxB8 plasmid; Addgene for plasmid distribution as noted; St. Vincent's Hospital Bioresource's Centre for care of experimental animals; S Taylor, A Goradia and E Tonkin for technical assistance. The authors acknowledge the facilities, and the scientific and technical assistance of the following Phenomics Australia nodes: Victorian Centre for Functional Genomics, Peter MacCallum Cancer Centre; and the Rodent Histopathology Service, University of Melbourne. The *Klhdc3*<sup>fl/fl</sup> mice were produced via CRISPR genome editing by the Monash Genome Modification Platform (MGMP), Monash University is a node of Phenomics Australia. Schematic figures were made BioRender.com and NIH Bioart. This work was supported by the National Health and Medical Research Council Australia (NHMRC; GNT2018098; CRW), Medical Research Future Fund (MRFF) – Emerging Priorities and Consumer Driven Research Initiative - 2020 Childhood Cancer Research Grant (MRF2007435; CRW, MFS, JKH, TYT), RMIT Vice Chancellor's Fellowship (JKH), RMIT postgraduate scholarship (CP) and St Vincent's Institute top-up scholarship (CP); a Melbourne Research Scholarship (WC-T; University of Melbourne), National Institutes of Health grant R35GM148071 (AK). The Victorian Centre for Functional Genomics is funded by the Australian Cancer Research Foundation (ACRF), Phenomics Australia, through funding from the Australian Government's National Collaborative Research Infrastructure Strategy (NCRIS) program, the Peter MacCallum Cancer Centre Foundation and the University of Melbourne Collaborative Research Infrastructure Program. Phenomics Australia (nodes utilised: Monash Genome Modification Platform (MGMP), Monash University; Rodent Histopathology Service, University of Melbourne) is supported by the Australian Government Department of Education through the National Collaborative Research Infrastructure Strategy, the Super Science Initiative and the National Collaborative Research Infrastructure Scheme (NCRIS). The work was supported in part by the Victorian State Government Operational Infrastructure Support Scheme to St Vincent's Institute and Hudson Institute of Medical Research. Work described was conducted on Wurundjeri, Boonwurrung and Wurundjeri Woi-wurrung lands of the Kulin Nation. The funders had no role in study design, data collection and analysis, decision to publish or preparation of the manuscript.

## Author contributions

**Paula Armina V Buco**: Formal analysis; Investigation; Visualization; Methodology; Writing—review and editing. **Wilson Castillo-Tandazo**: Formal analysis; Investigation; Visualization; Methodology; Writing—review and editing. **Alistair M Chalk**: Formal analysis; Investigation; Visualization; Methodology; Writing—review and editing. **Courtney Pilcher**: Investigation; Visualization. **Jessica K Holien**: Supervision; Funding acquisition; Investigation. **Jörg Heierhorst**: Visualization; Methodology. **Tiong Y Tan**: Conceptualization; Funding acquisition; Writing—review and editing. **Amnon Koren**: Formal analysis; Investigation; Visualization; Methodology. **Monique F Smeets**: Conceptualization; Formal analysis; Supervision; Investigation; Visualization; Methodology; Writing—original draft; Writing—review and editing. **Carl R Walkley**: Conceptualization; Formal analysis; Supervision; Funding acquisition; Investigation; Visualization; Methodology; Writing—original draft; Writing—review and editing.

Source data underlying figure panels in this paper may have individual authorship assigned. Where available, figure panel/source data authorship is listed in the following database record: biostudies:S-SCDT-10_1038-S44319-026-00727-2.

## Disclosure and competing interests statement

The authors declare no competing interests.

# Expanded View Figures

**Figure EV1.   Cell cycle kinetics of WT, point mutant and *Recql4*-deficient myeloid cells.**

(**A**) Representative flow cytometry plots assessing DNA replication rates following pulse labelling of asynchronous cultures of myeloid cells with EdU incorporation and DAPI (DNA stain). Genotypes assessed as *Recql4* wild-type (WT), *Recql4*$^{fl/G522Efs}$, *Recql4*$^{fl/R347X}$ and *Recql4*$^{fl/fl}$. All cell lines are *R26*-CreER$^{T2ki/+}$ and *Klhdc3* wild-type. Day after tamoxifen addition as indicated. (**B**) Genomic PCR showing recombination of the floxed *Recql4* allele at the indicated days after the addition of tamoxifen in each respective genotype. All cells are *Klhdc3*$^{+/+}$. (**C**) Schematic of the predicted protein products from the different Recql4 alleles used in this study.

▶

**A.**

All are *R26*-CreER^T2KI/+ *Recql4*

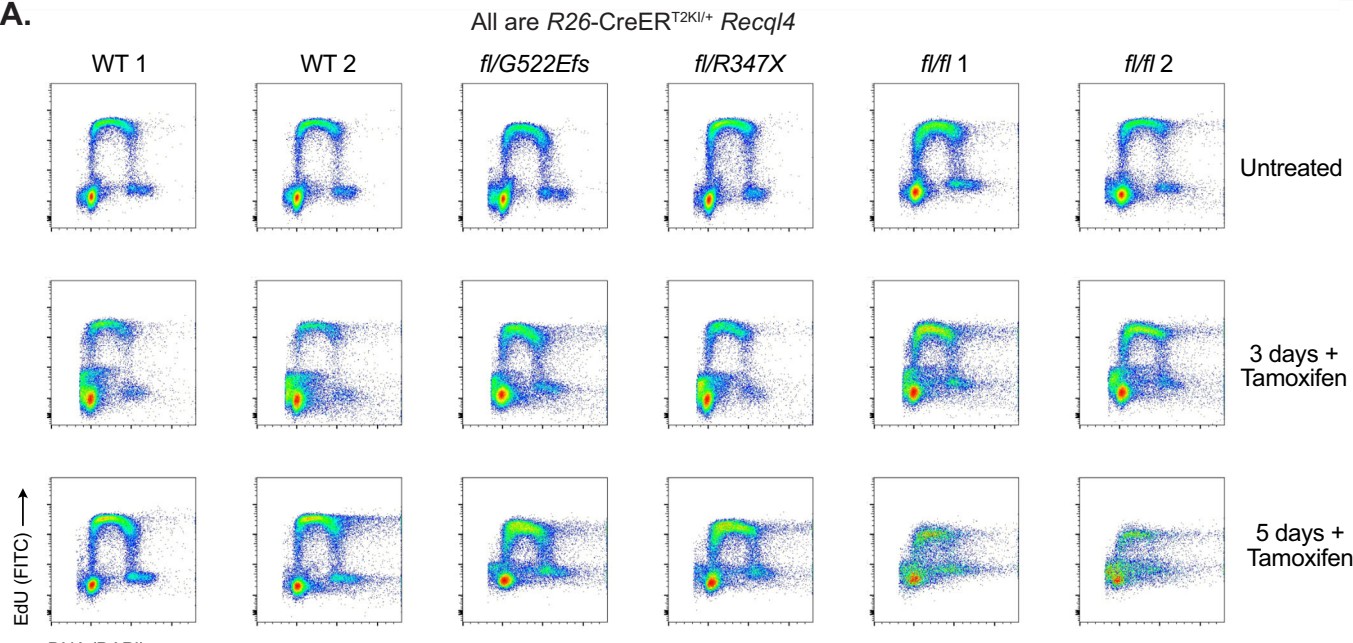

**B.**

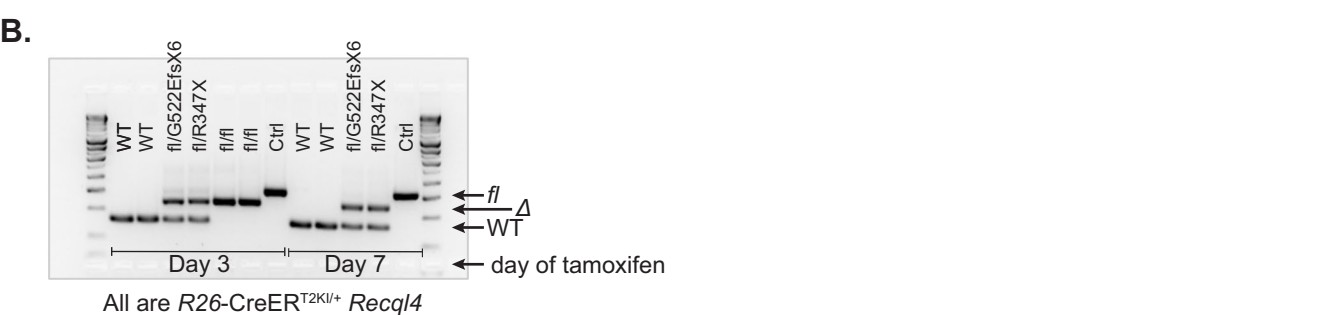

All are *R26*-CreER^T2KI/+ *Recql4*

**C.**

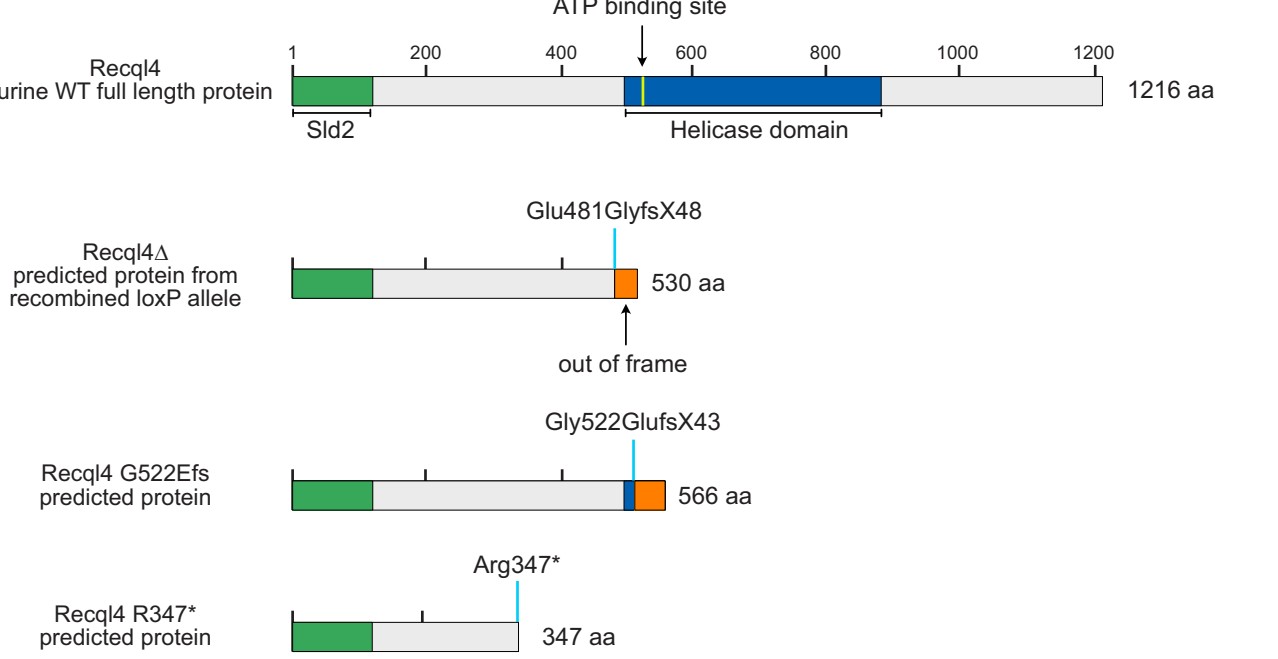

## A.

### E12.5 $Recql4^{+/-}Klhdc3^{+/-}$ inbreeding

| Mouse # | No. of Embryos | $R^{+/+}K^{+/+}$ | $R^{+/+}K^{+/-}$ | $R^{+/+}K^{-/-}$ | $R^{+/-}K^{+/+}$ | $R^{+/-}K^{+/-}$ | $R^{+/-}K^{-/-}$ | $R^{-/-}K^{-/-}$ |
|---|---|---|---|---|---|---|---|---|
| F120 | 7 | 1 | 0 | 0 | 0 | 5 | 1 | 0 |
| F128 | 6 | 0 | 3 | 0 | 1 | 0 | 0 | 2 |
| F125 | 8 | 3 | 1 | 1 | 1 | 1 | 1 | 0 |
| F147 | 6 | 0 | 1 | 2 | 0 | 2 | 1 | 0 |
| F135 | 5 | 2 | 1 | 1 | 0 | 1 | 0 | 0 |
| F182 | 7 | 0 | 2 | 1 | 1 | 2 | 1 | 0 |
| F183 | 5 | 0 | 1 | 1 | 0 | 2 | 0 | 1 |
| **Total** | **44** | **6** | **9** | **6** | **3** | **13** | **4** | **3** |
| | | **13.64%** | **20.45%** | **13.64%** | **6.82%** | **29.55%** | **9.09%** | **6.82%** |
| Expected | | 2.75 | 5.5 | 2.75 | 5.5 | 11 | 5.5 | 2.75 |
| | | 6.25% | 12.50% | 6.25% | 12.50% | 25.00% | 12.50% | 6.25% |

## B.

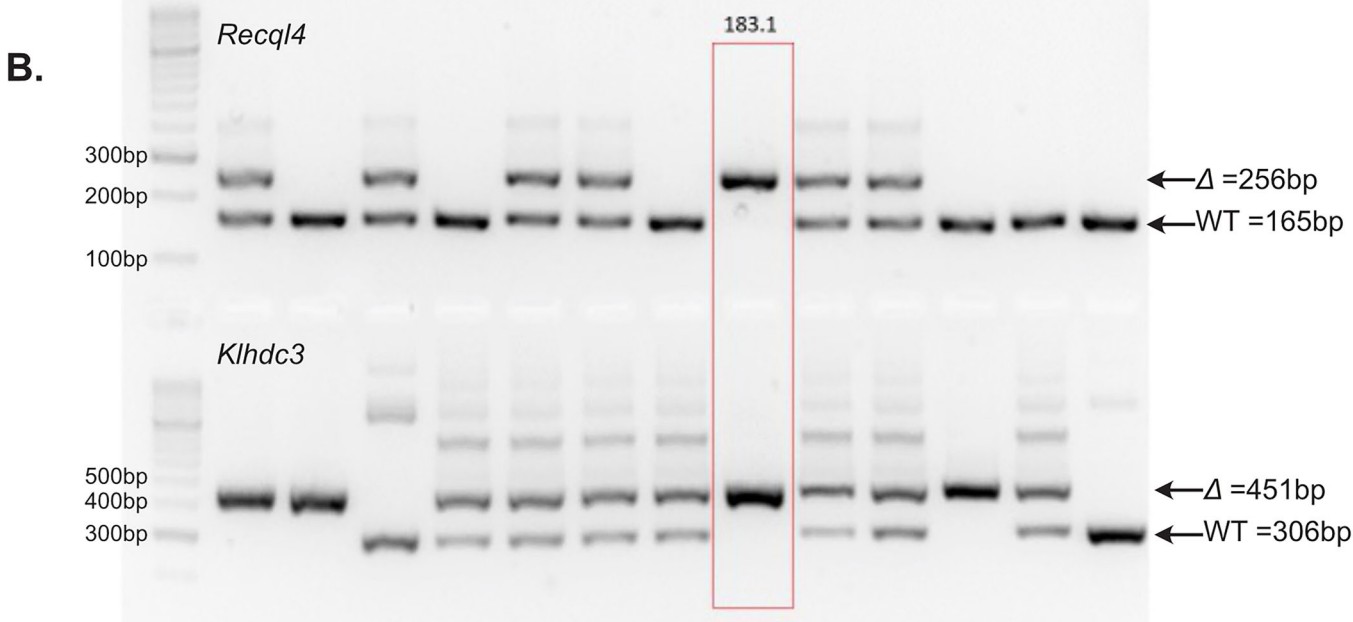

Each vertical lane represents DNA genotyping from an individual embryo

**Figure EV2. Loss of *Klhdc3* extends the survival of *Recql4* deficient embryos in vivo.**

(A) Recovery of indicated genotypes at embryonic day 12.5 (E12.5) from inbreeding of $Recql4^{+/-}Klhdc3^{+/-}$ breeding pairs. Previous analysis demonstrated that the $Recql4^{-/-}$ embryos were lethal prior to E10.5 (specific time point prior to this not determined). (B) Genomic PCR showing demonstrating recovery of a homozygous embryo.

 

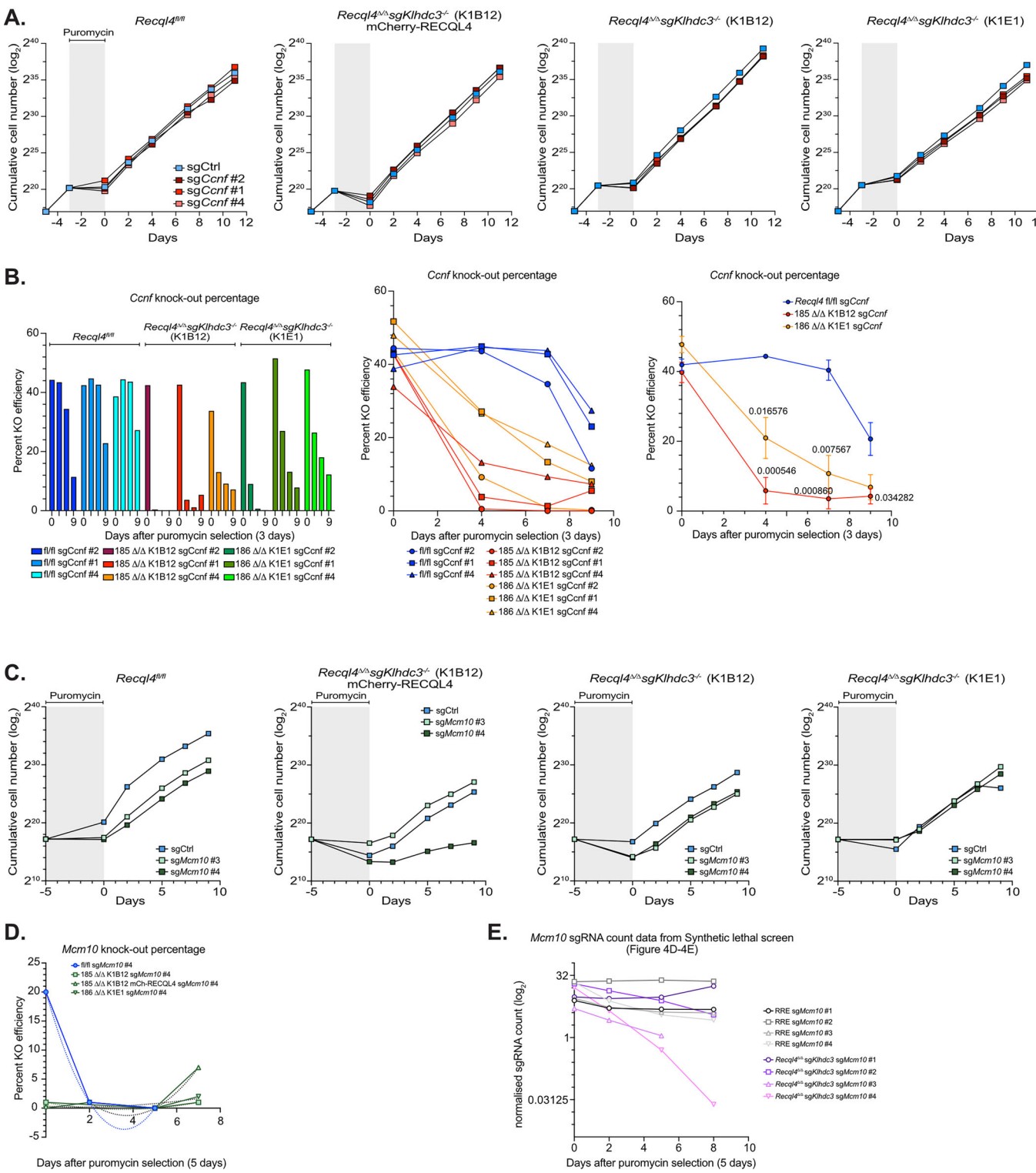

**Figure EV3. Validation that loss of Cyclin F is synthetic lethal with *Recql4* deficiency.**

(A) Proliferation curves of sgCtrl and sg*Ccnf*-targeted cell lines of the indicated genotypes. Grey shaded area indicates puromycin selection. *Recql4*$^{\Delta/\Delta}$ sg*Klhdc3* K1B12 and *Recql4*$^{\Delta/\Delta}$ sg*Klhdc3* K1E1 are independently targeted and isolated clones. (B) The knockout efficiency of *Ccnf* was measured through Sanger Sequencing and analysed by TIDE at each time point in each genotype as indicated. Data shown as each sample individually and as the mean knockout efficiency of 3 independent sg*Ccnf* guides $+/-$ SEM (right panel). Statistical analysis was done using multiple unpaired t-tests with significant p-values listed based on the nonlinear fit model of the data. (C) Proliferation curves of sgCtrl and sg*Mcm10*-targeted cell lines of the indicated genotypes. Grey shaded area indicates puromycin selection. *Recql4*$^{\Delta/\Delta}$ sg*Klhdc3* K1B12 and *Recql4*$^{\Delta/\Delta}$ sg*Klhdc3* K1E1 are independently targeted and isolated clones. (D) The knockout efficiency of *Mcm10* was measured through Sanger Sequencing and analysed by Synthego at each time point in each genotype as indicated. Data shown as each sample individually. (E) sgRNA counts from the synthetic lethal screen (Fig. 4) for *Mcm10* targeting sgRNAs in the *Recql4*$^{\Delta/\Delta}$ sg*Klhdc3* cells (labelled as DKO – double knock-out) and control *Recql4*$^{\Delta/\Delta}$ sg*Klhdc3* reconstituted with mCherry-RECQL4 (labelled as RRE – RECQL4 re-expressed).

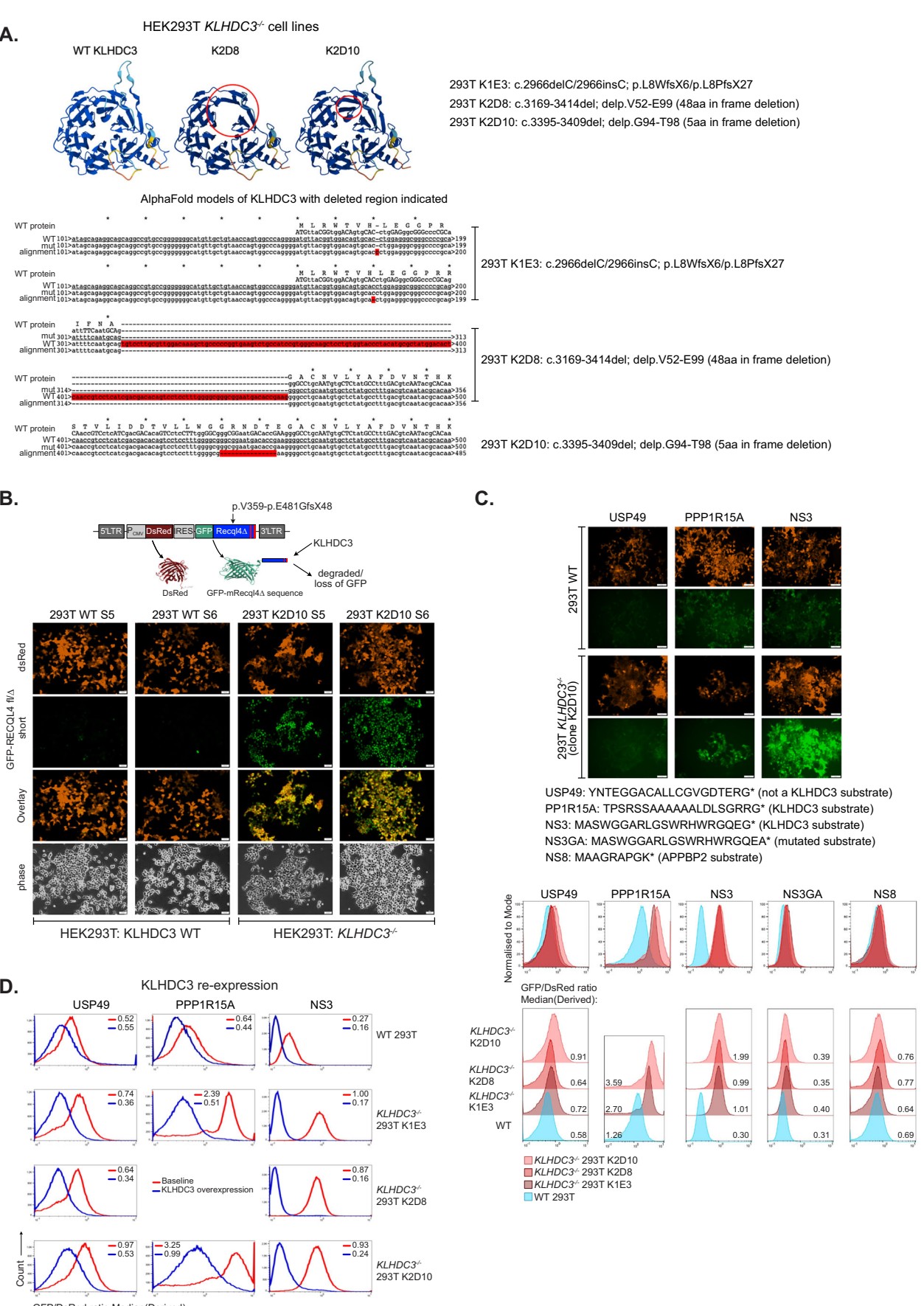

**Figure EV4.   Generation of KLHDC3⁻ᐟ⁻ 293 T cells.**

(A) Schematic and analysis of 3 independent *KLHDC3⁻ᐟ⁻* 293 T cell lines. Each line was confirmed as a homozygous mutant. (B) The C-terminal region of the *Recql4* deleted allele coding for p.V359-E481GfsX48 was cloned into the GPS-reporter plasmid. In this plasmid DsRed is constitutively expressed and GFP stability/expression is determined by the C-terminus of the fused RECQL4 deleted product. This reporter was expressed in KLHDC3 WT 293 T cells or *KLHDC3⁻ᐟ⁻* K2D10 293 T cells as indicated. Fluorescence signal was detected by live cell fluorescent imaging; images of RECQL4 p.V359-E481GfsX48 clone S5 and S6 (clone S4 shown in Fig. 5). Scale bar represents 50 µm. (C) Validation of KLHDC3 deficiency using GPS reporter assay by either live cell fluorescent microscopy of flow cytometry. Representative images of GPS reporters for USP49, PPP1R15A and NS3 (all KLHDC3 substrates) and NS3GA and NS8 (not KLHDC3 substrates) by live cell fluorescent imaging in WT and a *KLHDC3⁻ᐟ⁻* 293 T cell (clone K2D10). Scale bar represents 50 µm. Representative flow cytometric analysis of the GFP and DsRed expression in 3 independently generated *KLHDC3⁻ᐟ⁻* 293 T cells compared to KLHDC3 WT 293 T cells; Derived mean value for each sample as indicated calculated using FlowJo. (D) Re-expression of KLHDC3 in the *KLHDC3⁻ᐟ⁻* 293 T cells leads to loss of GFP signal for the known KLHDC3 substrates USP49, PP1R15A and NS3. The baseline GFP/DsRed derived median is in red, the KLHDC3 over-expressing samples are in blue. Derived mean value for each sample as indicated calculated using FlowJo. Note the left shift in the WT KLHDC3 re-expressing *KLHDC3⁻ᐟ⁻* 293 T cells indicative of destruction of the GFP-fusion protein.

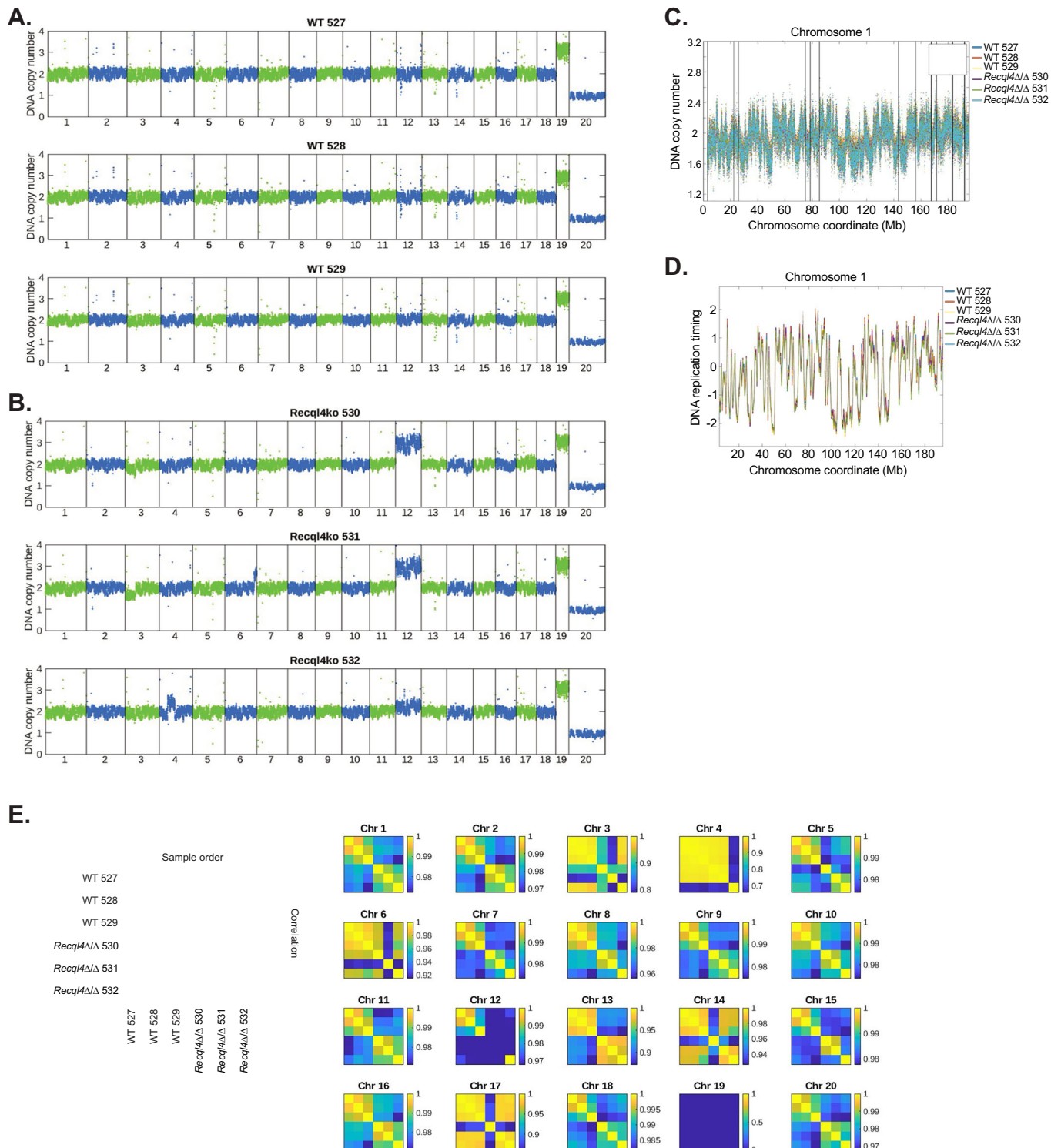

**Figure EV5. Individual whole-genome DNA copy number analysis for the myeloid cell lines used for DNA replication timing inference.**

(A) DNA copy number across all chromosomes for WT myeloid cell lines. (B) DNA copy number across all chromosomes for *Recql4^{Δ/Δ}* sg*Klhdc3* myeloid cell lines. (C) Whole-genome sequencing was used to infer DNA replication timing across the genome in 3 control and 3 *Recql4^{Δ/Δ}* sg*Klhdc3* cell lines (WT: 144 *Recql4* +/+ LCr. Hygro Ctrl C12, E12 and H11 and *Recql4* Δ/Δ: 185 *Recql4* Δ/Δ LCr. Hygro sg*Klhdc3* K2C12 and K1B12 and 186 *Recql4* Δ/Δ LCr. Hygro sg*Klhdc3* K1E1). Representative example from each sample is plotted individually for Chromosome 1. (D) Representative example of smoothed plots of DNA replication timing across chromosome 1 for each cell line. (E) Correlation analysis of DNA replication timing of each chromosome across the genome with the scale for each individual chromosome. Note that missing data, due to filtering of specific chromosomes in some samples, appears as dark blue in the correlation matrices.

