## [Peer Review File · EMBO Reports]

Minute amounts of helicase-deficient truncated RECQL4 are sufficient for DNA replication.

Paula Armina Bucu, Wilson Castillo-Tandazo, Alistair Chalk, Courtney Pilcher, Jessica Holien, Jörg Heierhorst, Tiong Tan, Amnon Koren, Monique Smeets, and Carl Walkley

Corresponding author(s): Carl Walkley (carl.walkley@hudson.org.au)

Review Timeline:

Submission Date:	27th Jul 25
Editorial Decision:	12th Sep 25
Revision Received:	3rd Dec 25
Editorial Decision:	22nd Jan 26
Revision Received:	27th Jan 26
Accepted:	9th Feb 26

Editor: Esther Schnapp

Transaction Report:

Dear Dr. Walkley,

Thank you for the submission of your manuscript to EMBO reports. We have now received the full set of referee reports as well as referee cross-comments that are pasted below.

As you will see, the referees acknowledge that the findings are potentially interesting. Referee 1 suggests that you analyse the cell line from the Halazonetis Lab, however, referee 2 does not think that this is helpful and we agree that this point can be discussed in the ms text. All other referee concerns will need to be addressed for the publication of your study by EMBO reports. Referee 2 also suggests that you test whether RECQL4-mutant cells are sensitive to MCM10 knockdown, and I think this is a good idea, if you agree.

Please let me know in case you have any comments, and we can discuss the exact revision requirements further, also in a video chat, if you like.

I would thus like to invite you to revise your manuscript with the understanding that the referee concerns must be fully addressed and their suggestions taken on board. Please address all referee concerns in a complete point-by-point response. Acceptance of the manuscript will depend on a positive outcome of a second round of review. It is EMBO reports policy to allow a single round of major revision only and acceptance or rejection of the manuscript will therefore depend on the completeness of your responses included in the next, final version of the manuscript.

We realize that it is difficult to revise to a specific deadline. In the interest of protecting the conceptual advance provided by the work, we recommend a revision within 3 months (13th Dec 2025). Please discuss the revision progress ahead of this time with the editor if you require more time to complete the revisions.

- 1) A data availability section providing access to data deposited in public databases is missing. If you have not deposited any data, please add a sentence to the data availability section that explains that.
- 2) Your manuscript contains statistics and error bars based on $n=2$. Please use scatter blots in these cases. No statistics should be calculated if $n=2$.

5) a complete author checklist, which you can download from our author guidelines <https://www.embopress.org/page/journal/14693178/authorguide>. Please insert information in the checklist that is also reflected in the manuscript. The completed author checklist will also be part of the RPF.

6) Please note that all corresponding authors are required to supply an ORCID ID for their name upon submission of a revised

manuscript (<<https://orcid.org/>>). Please find instructions on how to link your ORCID ID to your account in our manuscript tracking system in our Author guidelines
<<https://www.embopress.org/page/journal/14693178/authorguide#authorshipguidelines>>

12) All Materials and Methods need to be described in the main text using our 'Structured Methods' format, which is required for all research articles. According to this format, the Methods section includes a separate Reagents and Tools Table file (listing key reagents, experimental models, software and relevant equipment and including their sources and relevant identifiers) and a Methods and Protocols section describing the methods using a step-by-step protocol format. The aim is to facilitate adoption of the methodologies across labs. More information on how to adhere to this format as well as a downloadable template (.docx) for the Reagents and Tools Table can be found in our author guidelines:
<https://www.embopress.org/page/journal/14693178/authorguide#structuredmethods>.

An example of a Method paper with Structured Methods can be found here: <https://www.embopress.org/doi/full/10.1038/s44320-024-00037-6#sec-4>

As part of the EMBO publication's Transparent Editorial Process, EMBO reports publishes online a Review Process File (RPF)

to accompany accepted manuscripts. This File will be published in conjunction with your paper and will include the referee reports, your point-by-point response and all pertinent correspondence relating to the manuscript.

I look forward to seeing a revised form of your manuscript when it is ready.

Yours sincerely,

Referee #1:

The article by Buco et al. describes a detailed analysis of an issue that has been debated extensively in the DNA replication field; namely, is RECQL4 protein required for replication origin firing like its yeast ortholog? The authors conclude that some of the controversy has resulted from the fact that only trace amounts of RECQL4 are required to support DNA replication. In some previous studies, incomplete depletion of RECQL4 using siRNAs might have given misleading results. In addition, the authors have identified an interesting regulatory pathway involving the KLHDC3 protein that forms a component of an E3 ubiquitin ligase system. Overall, the study has been conducted thoroughly, and the findings would be of interest to many groups working on genome stability and DNA replication.

The 'elephant in the room' is the fact that the Halazonetis group reported last year that the human RECQL4 gene can be inactivated without the cells having any major defect in replication origin firing or in cell cycle progression. While this published paper is cited by the authors in their manuscript (Padayachty et al) it is done so in a way that is not appropriate in my view. At the bottom of page 12 it is stated that 'this has important implications for studies using siRNA and other efficient, but not absolute, means to reduce expression of RECQL4'. This remark is, at best, misleading given that the published paper claims to have generated a full gene knockout with the cells not expressing any protein. Those published data seem at odds with the data in this manuscript to such an extent that it is hard for this referee to know what is correct and what is not, or indeed whether both findings might be correct due to cell line differences etc. Nevertheless, the superficial and misleading way in which the Halazonetis work is described gives the impression of an attempt to ignore the already published findings. One obvious solution would be for the authors to obtain the cell line from the Halazonetis Lab and define whether it is a true KO or expresses some truncated protein. The position of the guide RNAs used in the Halazonetis paper should not have resulted in the production of any truncated protein of a significant size, as far as I can ascertain, but errors do occur during the generation of KO cell lines of course.

Referee #2:

RecQL4 is a genome maintenance factor involved in replication initiation and DNA damage response, whose mutations are linked to Rothmund-Thompson, Baller-Gerold, and RAPADILINO syndromes. How and why RecQL4 mutations cause these diseases remains poorly understood and has been the focus of many studies over the last ~25 years. The manuscript titled "Minute amounts of helicase-deficient truncated RECQL4 are sufficient for DNA replication." by Buco and Castillo-Tandazo et al. used a very elegant screen-based strategy to identify factors whose loss could rescue the proliferation defect of mouse cells bearing a hypomorphic RecQL4 mutation (truncation) similar to those that have been reported in human patients. Specifically, the RecQL4 truncation used in this study lacks the C-terminal portion of RecQL4 that contains an ATPase domain. Importantly, this mutation also destabilized the truncated RecQL4 protein product - which is also thought to happen with many RecQL4 truncations in human patients.

This manuscript has several interesting findings, but I would like to highlight a few that stood out:

1. The study identified a subunit of an E3 ligase (KLHDC3) whose loss prevents the most severe defects caused by the RecQL4 truncation (including bone marrow failure).
2. The study showed that the loss of KLHDC3 stabilized the truncated RecQL4 protein fragment by preventing its degradation.
3. The study showed that very low levels of truncated RecQL4 (5-7% of that measured for full-length RecQL4 in a control cell line) are sufficient to support relatively normal cell cycle progression.

4. The most impactful finding of this study is that "there are unlikely to be monogenic loss of function rescue alleles for RECQL4 mutation", and all the data presented here strongly supports this conclusion.

Overall the work is rigorous and significantly advances our understanding of RECQL4 mutations, their effect on cell proliferation, and RECQL4's role in human disease. It deserves to be published in EMBO reports, with a few minor edits as indicated below.

I have two major questions/criticisms about the interpretation of data. I hope they can be resolved in the final version of the manuscript.

1. The flow cytometry data (Figure 2A-B) indicate that cells with truncated RECQL4 incorporate less EdU, consistent with the idea that they replicate DNA less efficiently (presumably by firing fewer replication origins). The authors write "The mechanism via which the RECQL4 p.E481GfsX48 enables relatively normal DNA replication is still to be fully resolved. Our analysis indicate that this small fragment supports complete DNA replication, but delays completion of DNA synthesis with a shift towards late S phase and even into M phase. This shifted DNA synthesis, evidenced by EdU, presumably depends on MiDAS contributing to the ability of the cells to complete replication". At the same time, the Origin Firing Timing data (Figure 6A-D) did not reveal any major differences between the cells bearing WT RECQL4 and truncated RECQL4. The authors concluded that "This analysis demonstrated that there were, at most, subtle differences between cells with and without the ATP-dependent helicase and C-terminal domains of RECQL4 (Figure 6D)." The replication timing analysis shows which regions of the genome are replicated early and which are replicated late. In other words, replication timing provides a relative quantification of early/late replicating regions. As far as I understand, replication timing analysis does not provide an absolute measure of origin firing efficiency. In other words, it is possible for the RECQL4 truncation to have overall less efficient origin firing while maintaining the same origin firing timing program. Measuring the absolute efficiency of origin firing in cells is technically challenging. One could measure the inter-origin distance (IOD) using fiber analysis - IOD is expected to increase if origin firing efficiency decreases. However, I do realize that this may be outside the scope of this manuscript.

2. The authors measured the amount of MCM3 (a subunit of the MCM2-7 complex) and observed that there was more chromatin-bound MCM3 in cell lines with truncated RECQL4 compared to control cell lines (Figure 2D). The authors provide the following interpretation of this observation: "The small RECQL4 fragment is associated with an increase in licensed replication origins. This may be because the helicase domain normally acts to limit how many origins are licensed (Terui et al., 2024). As a consequence, more active origins during early S phase with increased dNTP depletion may cause the replication stress that ultimately delays complete replication into M phase. Alternatively, increased origin licensing could also be a compensatory response where more active origins result in shorter replicons making up for slower replication in the absence of the helicase domain."

I have a few objections to this interpretation by Buco et al.: (a) Terui et al 2024 do not provide any data on RECQL4 modulating licensing (they talk about CMG helicase activation). In fact RECQL4 has not been previously implicated in licensing (which involves ORC, CDT1, CDC6 and MCM2-7, for the most recent review, pls see <https://doi.org/10.1042/BST20220917>). (b) the authors invoke more active origins and dNTP depletion in early s-phase with accompanying replication stress, but do not provide any evidence for this claim. This line of reasoning is a bit too speculative. (c) increased origin licensing could be plausible, perhaps due to some adaptation of this cell line. However, another interpretation of the data (perhaps the most parsimonious) is that the cell line with the RECQL4 truncation replicates its DNA slowly (consistent with the data from Buco et al), which causes a relatively slow clearance of pre-replication complexes (MCM2-7) from DNA. The vast majority of MCM2-7 loaded on DNA never initiate replication. Instead, they are cleared from DNA by active replication forks (as shown by Jean Cook's lab and Jiri Lukas' lab previously, see for example <https://www.nature.com/articles/s41467-022-33887-5>). The decrease in MCM3 signal throughout S-phase corresponds to this "clearing" of MCM2-7 from chromatin. Inefficient replication initiation by the RECQL4 truncation can therefore explain the higher levels of MCM3 during early S-phase. As for the higher levels observed in G1-phase - it's difficult to distinguish very early S-phase from G1 on the FACS plots.

I have a few minor comments about how the authors interpret some of their observations, as well as data from other studies. All of these can be addressed by small edits to the text and more clear explanation in some cases.

1. The authors state that "This result demonstrated that the ATP-dependent helicase activity of RECQL4 was not essential *in vivo* for DNA replication or murine homeostasis, consistent with analysis in human cell line models (Padayachy et al, 2024)." I think it's worth explicitly mentioning in the revised manuscript that Padayachy et al 2024 showed that knocking out RECQL4 entirely in a human cell line still allowed those cells to proliferate relatively normally. That study suggests that in some cell lines in cell culture conditions, RECQL4 is not essential for viability. Another study which was recently posted on BiorXiv (<https://www.biorxiv.org/content/10.1101/2025.08.05.668837v1>) reports that RECQL4 is partially redundant with MCM10 when it comes to promoting origin firing. This explains why in some circumstances cells can proliferate without RECQL4 (Padayachy et al 2024). This also explains why depleting RECQL4 from Xenopus egg extract does not fully inhibit DNA replication (as reported in Sangrithi et al 2005, Matsuno et al, 2006, Terui et al, 2024).

2. Interestingly, Padayachy et al found that RECQL4-KO (knock-out) and RECQL4-HD (helicase dead) cell lines exhibited similar origin firing profiles (i.e. they have similar replication timing profiles). However, Padayachy et al reported that the RECQL4-KO

cells exhibited higher replication fork rates which are consistent with decreased origin firing efficiency. This echoes my major criticism above about interpreting replication timing data.

3. The authors repeatedly state that the ATP-dependent helicase activity of RECQL4 is not essential for DNA replication in mammals. They contrast this with a recent *Xenopus* study (Terui et al, 2024), which reported that RECQL4's ATPase activity does play a role in replication initiation. In fact, a few older *Xenopus* studies (Sangrithi et al, 2005 and Matsuno et al, 2006) also reported that RECQL4's ATPase plays a role in replication initiation. All aforementioned *Xenopus* studies do report that ATPase-deficient RECQL4 mutants can partially rescue RECQL4 depletions. So everyone agrees that RECQL4 can support origin firing without its ATPase function (i.e. the ATPase is not strictly essential for replication). However, the *Xenopus* studies do provide evidence that RECQL4's ATPase activity further boosts RECQL4's ability to initiate replication.

4. In the introduction, the authors write "Although RECQL4 was long considered the mammalian homologue of the essential yeast DNA replication factor Sld2, it is now clear that a structurally unrelated protein DONSON has evolved to carry out the Sld2-like function during mammalian DNA replication (Cvetkovic et al, 2023; Evrin et al, 2023; Hashimoto et al, 2023; Kingsley et al, 2023; Lim et al, 2023). As a result, the role and function of RECQL4 in normal DNA replication remains a major knowledge gap in our understanding of this fundamental cellular process." The statement that DONSON is the functional counterpart of SLD2 in mammals is just a speculation entertained in some studies which reported that DONSON is a novel replication initiation factor in metazoa. However, this is not yet widely accepted and is an area of active investigation. I suggest changing "it is now clear" to "it has been proposed" or "it has been speculated".

Referee #3:

The manuscript by Bucu et al. investigates the role of a truncated RECQL4 product in supporting cellular proliferation both in cellulo and in vivo in mouse models. The authors deleted *Recql4* exons 9-10 via Cre-Lox recombination, which normally results in bone marrow failure and cell death. Using forward genetic CRISPR/Cas9 screens, they identified KLHDC3 as the sole suppressor of this phenotype. Mechanistic studies revealed that recombination generated a truncated RECQL4 fragment (aa 1-480) fused to a 50 aa C-terminal tail (frameshifted), producing a neo-degron recognised by KLHDC3 and targeted for proteasomal degradation. In the absence of KLHDC3, residual truncated RECQL4 stabilises, maintaining sufficient function to support cell proliferation-suggesting that the ATPase and helicase domains are dispensable for RECQL4's essential role. This is an important and timely study, as it helps reconcile contradictory findings from human, mouse, and *Xenopus* studies regarding RECQL4's role in DNA replication. Nonetheless, several issues need to be addressed before the manuscript is suitable for publication.

Major Comments:

- 1) The truncation arising from the Cre/LoxP floxed allele should be characterised more thoroughly. Given the suggested importance of the RECQL4 N-terminal domain, this mutant could serve as an informative control in downstream assays. Alternatively, the authors should clarify why it was not explored further (e.g. sensitivity to replication stress, DNA fibre assays for fork restart or origin firing).
- 2) The R347X mutant is included in the initial screen but not utilised in subsequent experiments. A concise summary of its known properties in the main text (beyond citation) would improve readability and contextualise the results.
- 3) Indicators of statistical testing (p-values, error bars, replicates) are reported inconsistently across figures. Standardising presentation is essential to support key conclusions, as some claims currently lack sufficient data.
- 4) Several panels (e.g. Fig. 1) are shown as overlay histograms without numerical summaries. Quantitative plots with error bars and replicate numbers should be included for reproducibility.
- 5) Ensure that all supplementary figures and datasets are explicitly cited in the main text, to improve accessibility.
- 6) The authors should comment on why no other CRL E3 ligase components or proteasome regulators emerged from the suppressor screen.
- 7) Do HEK293T cells recapitulate the murine phenotypes upon RECQL4 depletion? Extending the findings to human cells would strengthen the conclusions.
- 8) Both text and figures require simplification and consistency. Examples include:
 - consistent labelling of alleles and truncations (Δ , R347X, truncation);
 - legends that match figure elements (e.g. missing MG132 annotation; different symbols for V5-KLHDC3 variants across figures; unexplained yellow asterisk in Fig. 5A);
 - poor visibility of Fig. 4E;
 - overly complex graphs (e.g. Fig. 4B);
 - unclear flow cytometry plots (Figs. 2C-D, colours not matched to cell-cycle phases);
 - insufficient detail in Fig. 5D, which requires a close-up.

Minor Comments:

- 1) State the number of biological replicates for key assays (e.g. replication timing WGS, GPS stability assays).
- 2) Expand the description of TIGER analysis to assist readers less familiar with replication timing approaches.
- 3) The discussion could briefly address how these findings integrate with recent studies implicating RECQL4 in chromosome

segregation, where similar truncations were analysed.

4) Fig. S2a likely mislabels fl/+ as Δ /+.

5) Ensure consistent terminology for "Fig." vs "Figure" (e.g. p. 6).

6) The suggestion to apply MIDAS should be either experimentally supported or toned down in the text.

7) A schematic summarising all RECQL4 truncations used in this study would greatly aid clarity and unify nomenclature across the manuscript.

In summary, this study addresses an important question in DNA replication biology and provides mechanistic insight into RECQL4 function. However, additional clarification, consistent data presentation, and figure/text improvements are needed to make the manuscript fully convincing and accessible to readers.

Cross-comment from referee 2:

We know that RECQL4's ATPase is not essential for viability because human RTS, BGS, and RAPADILINO patients who lack the ATPase domain are alive, but clearly, their genome maintenance is compromised. In my review of Buco et al. I suggested that Buco et al address these discrepancies in their discussion, rather than having to do a ton of extra experiments.

We now appreciate that under certain circumstances (for example specific cell types) cells can tolerate very low levels of RecQL4 (and RecQL4 mutants) without arresting the cell cycle. In this context, the work from Buco et al. provides valuable insights in a physiological context.

Regarding the comment from another reviewer about the "elephant in the room" with the paper from the Halazonetis lab where they claim that RECQL4-deficient cells doesn't cause an effect on origin firing. There are a few caveats with interpreting the paper from the Halazonetis lab:

1. Certain cell types are able to tolerate very low (and perhaps nil) levels of RecQL4 because MCM10 is partially functionally redundant with RecQL4 during origin firing.

2. CRISPR knock-outs can be tricky to validate, sometimes there are alternative transcripts being made that retain some function. The Halazonetis paper did not exhaustively validate their RECQL4-KO cell line. In addition, commercial antibodies used to demonstrate the absence of RECQL4 products in the KO line may not be able to detect trace amounts of protein (or protein fragments) still present in that cell line. So in that sense, we shouldn't put the burden of proof on Buco et al to validate the cell line from the Halazonetis lab.

3. The Halazonetis lab did not do a thorough job in describing the origin firing defect caused by the KO of RecQL4 - in part because accurately measuring origin firing efficiency in cells is very hard. This is a limitation of many cell-based studies.

I agree with you that Buco et al. should address the discrepancies between different studies in their discussion section. It seems likely that some of these discrepancies are due to the partial functional redundancy between RecQL4 and MCM10.

I don't think it will help much if Buco et al repeated experiments with the cell line generated by the Halazonetis lab because they will confront the same exact technical limitations (the difficulty in fully validating a CRISPR KO line, the limited sensitivity of commercial antibodies, the inability to accurately measure origin firing efficiency in vivo).

One prediction based on recent work in different labs is that RECQL4-mutant cells rely heavily on MCM10 to initiate replication. Therefore, they should be sensitive to knocking down MCM10. I am not sure if siRNA or sgRNA can lower MCM10 levels sufficiently to see a clear defect in cell proliferation - that is one caveat of this approach... Perhaps that could be a direction that Buco et al can explore in a future study. Perhaps they could give it a try while revising their manuscript?

Our understanding of RECQL4 and its mutations continues to evolve as we better understand the molecular mechanism of replication initiation, and studies like Buco et al provide important clues and context for elucidating the basis of RECQL4-related diseases.

EMBOR-2025-62414V1 Decision Letter (sent 12 Sept 2025)

Thank you for the submission of your manuscript to EMBO reports. We have now received the full set of referee reports as well as referee cross-comments that are pasted below.

As you will see, the referees acknowledge that the findings are potentially interesting. Referee 1 suggests that you analyse the cell line from the Halazonetis Lab, however, referee 2 does not think that this is helpful and we agree that this point can be discussed in the ms text. All other referee concerns will need to be addressed for the publication of your study by EMBO reports. Referee 2 also suggests that you test whether RECQL4-mutant cells are sensitive to MCM10 knockdown, and I think this is a good idea, if you agree.

Please let me know in case you have any comments, and we can discuss the exact revision requirements further, also in a video chat, if you like.

I would thus like to invite you to revise your manuscript with the understanding that the referee concerns must be fully addressed and their suggestions taken on board. Please address all referee concerns in a complete point-by-point response. Acceptance of the manuscript will depend on a positive outcome of a second round of review. It is EMBO reports policy to allow a single round of major revision only and acceptance or rejection of the manuscript will therefore depend on the completeness of your responses included in the next, final version of the manuscript.

We realize that it is difficult to revise to a specific deadline. In the interest of protecting the conceptual advance provided by the work, we recommend a revision within 3 months (13th Dec 2025). Please discuss the revision progress ahead of this time with the editor if you require more time to complete the revisions.

1) A data availability section providing access to data deposited in public databases is missing. If you have not deposited any data, please add a sentence to the data availability section that explains that.

Author Response: All datasets are released

2) Your manuscript contains statistics and error bars based on $n=2$. Please use scatter blots in these cases. No statistics should be calculated if $n=2$.

Author Response: We have removed statistics where $n=3$ (this appeared to be only the case for Figure 3I).

Author Response: Completed

Author Response: Completed

3) We replaced Supplementary Information with Expanded View (EV) Figures and Tables that are collapsible/expandable online. A maximum of 5 EV Figures can be typeset. EV Figures should be cited as 'Figure EV1, Figure EV2' etc... in the text and their respective legends should be included in the main text after the legends of regular figures.

- For the figures that you do NOT wish to display as Expanded View figures, they should be bundled together with their legends in a single PDF file called *Appendix*, which should start with a short Table of Content. Appendix figures should be referred to in the main text as: "Appendix Figure S1, Appendix Figure S2" etc. See detailed instructions regarding expanded view here: <https://www.embopress.org/page/journal/14693178/authorguide#expandedview&qt;>

Author Response: Figures, EV figures and appendix have been ordered as requested.

Author Response: Included.

5) a complete author checklist, which you can download from our author guidelines <https://www.embopress.org/page/journal/14693178/authorguide&qt;>. Please insert information in the checklist that is also reflected in the manuscript. The completed author checklist will also be part of the RPF.

Author Response: Completed

6) Please note that all corresponding authors are required to supply an ORCID ID for their name upon submission of a revised manuscript (<https://orcid.org/&qt;>). Please find instructions on how to link your ORCID ID to your account in our manuscript tracking system in our Author guidelines <https://www.embopress.org/page/journal/14693178/authorguide#authorshipguidelines&qt;>

Author Response: ORCID are included.

7) Before submitting your revision, primary datasets produced in this study need to be deposited in an appropriate public database (see <https://www.embopress.org/page/journal/14693178/authorguide#datadeposition>). Please remember to provide a reviewer password if the datasets are not yet public. The accession numbers and database should be listed in a formal "Data Availability" section placed after Materials & Method (see also <https://www.embopress.org/page/journal/14693178/authorguide#datadeposition>). Please note that the Data Availability Section is restricted to new primary data that are part of this study.

* Note - All links should resolve to a page where the data can be accessed. *

Author Response: Completed and all datasets are publicly released and accessible (NCBI GEO).

Author Response: Completed

9) Our journal also encourages inclusion of *data citations in the reference list* to directly cite datasets that were re-used and obtained from public databases. Data citations in the article text are distinct from normal bibliographical citations and should directly link to the database records from which the data can be accessed. In the main text, data citations are formatted as follows: "Data ref: Smith et al, 2001" or "Data ref: NCBI Sequence Read Archive PRJNA342805, 2017". In the Reference list, data citations must be labeled with "[DATASET]". A data reference must provide the database name, accession number/identifiers and a resolvable link to the landing

page from which the data can be accessed at the end of the reference. Further instructions are available at <https://www.embopress.org/page/journal/14693178/authorguide#referencesformat>

Author Response: Completed

Author Response: Competing interest statement provided.

12) All Materials and Methods need to be described in the main text using our 'Structured Methods' format, which is required for all research articles. According to this format, the Methods section includes a separate Reagents and Tools Table file (listing key reagents, experimental models, software and relevant equipment and including their sources and relevant identifiers) and a Methods and Protocols section describing the methods using a step-by-step protocol format. The aim is to facilitate adoption of the methodologies across labs. More information on how to adhere to this format as well as a downloadable template (.docx) for the Reagents and Tools Table can be found in our author guidelines: <https://www.embopress.org/page/journal/14693178/authorguide#structuredmethods>.

An example of a Method paper with Structured Methods can be found here: <https://www.embopress.org/doi/full/10.1038/s44320-024-00037-6#sec-4>

Author Response: Structured methods provided.

I look forward to seeing a revised form of your manuscript when it is ready.

Note to reviewers and editor: in the text below where previously Supplemental Figures in the original manuscript are now referred to according their location in Extended View Figures or Appendix in the revised manuscript.

Referee #1:

The article by Buco et al. describes a detailed analysis of an issue that has been debated extensively in the DNA replication field; namely, is RECQL4 protein required for replication origin firing like its yeast ortholog? The authors conclude that some of the controversy has resulted from the fact that only trace amounts of RECQL4 are required to support DNA replication. In some previous studies, incomplete depletion of RECQL4 using siRNAs might have given misleading results. In addition, the authors have identified an interesting regulatory pathway involving the KLHDC3 protein that forms a component of an E3 ubiquitin ligase system. Overall, the study has been conducted thoroughly, and the findings would be of interest to many groups working on genome stability and DNA replication.

The 'elephant in the room' is the fact that the Halazonetis group reported last year that the human RECQL4 gene can be inactivated without the cells having any major defect in replication origin firing or in cell cycle progression. While this published paper is cited by the authors in their manuscript (Padayachty et al) it is done so in a way that is not appropriate in my view. At the bottom of page 12 it is stated that 'this has important implications for studies using siRNA and other efficient, but not absolute, means to reduce expression of RECQL4'. This remark is, at best, misleading given that the published paper claims to have generated a full gene knockout with the cells not expressing any protein. Those published data seem at odds with the data in this manuscript to such an extent that it is hard for this referee to know what is correct and what is not, or indeed whether both findings might be correct due to cell line differences etc. Nevertheless, the superficial and misleading way in which the Halazonetis work is described gives the impression of an attempt to ignore the already published findings. One obvious solution would be for the authors to obtain the cell line from the Halazonetis Lab and define whether it is a true KO or expresses some truncated protein. The position of the guide RNAs used in the Halazonetis paper should not have resulted in the production of any truncated protein of a significant size, as far as I can ascertain, but errors do occur during the generation of KO cell lines of course.

Author Response:

We thank the reviewer for their consideration of our results and comments. In the revised manuscript, we have now provided more context in the way we reference the Padayachy et al. paper compared to our original manuscript.

Regarding a direct comparison it is difficult to do this as outlined both in our response below and as noted in the summary from the consultative reviewer comments. A limitation of the Padayachy et al. paper is that most of the data shown involve using a different individual "KO" clone for different experiments rather than all experiments having been performed with multiple clones throughout the manuscript for all assays. We note that even Padayachy et al. themselves acknowledge in their discussion that even though they cannot detect any RECQL4 protein in the KO clones: "it is possible that very low levels of truncated, yet functional, RECQL4 polypeptides might still be expressed in the RECQL4 KO clones described here. If this were the case, then very low levels, almost undetectable levels, of truncated RECQL4 would suffice for origin firing. In the fsN clones, the entire Sld2-like domain of RECQL4 is not deleted, but the most highly conserved residues between RECQL4 and Sld2 are targeted by the CRISPR-induced mutagenesis." This qualification is entirely consistent with the conclusion that our findings suggest that a minute amount of chromatin bound N-terminal RECQL4 is sufficient for essentially normal DNA replication.

In our mutants, which express only a very small amount of N-terminal RECQL4, we find similar to Padayachy et al., no significant defect in viability, proliferation or global DNA replication. We do, however, see a mild reduction in DNA synthesis in mid to late-S-phase, which Padayachy et al. again concede may have missed: "Another possibility is that RECQL4 is required for firing of

origins in mid or late S phase, but not in early S phase. Since the EdU-seq method identifies only the origins that fire in early S phase, our study would have missed such a function."

Overall, the majority of published studies where RECQL4 loss of function has been attempted report defects in cell proliferation and viability. This has most often been the finding in independent studies outside of the Halazonetis groups recent report as noted by the reviewer. For example, independent work from the Bohr group (*Nucleic Acids Res.* 2020 Jul 9;48(12):6530-6546. doi: 10.1093/nar/gkaa392; and this paper is not cited by Padayachy et al.,) generated CRISPR/Cas9 RECQL4 knock-out cells using U2OS cells and noted a significant reduction in proliferation, EdU incorporation and viability –phenotypes could all be rescued by re-expression of RECQL4. Whilst U2OS cells may differ between labs it does indicate that the findings reported in Padayachy et al., are not consistent with others independent studies using U2OS cell lines or with the body of work using primary and immortalised mouse models and other species.

Beyond the extensively in vitro cultured U2OS cell line, there is a vast literature about Recq14 deficiency in arguably more physiological whole animal or primary cell models:

- Germ-line loss of *Recq14* in the mouse is early embryonic lethal in both our model and independent models (see PMID: 24960165 (our previous work); PMID:25556649 (Prof Lisa Wang's independently generated *Recq14* allele that targets a different region of the gene); PMID:11979727 (originally reported model from 2002)).
- We cannot recover and establish stable *RECQL4* knock-out HEK293T cell lines using CRISPR/Cas9.
- We cannot maintain immortalise myeloid cells from our mouse models following deletion of endogenous *Recq14* unless we also have loss of *Klhdc3* (the basis of the current manuscript). This has been reproducible with both Cre-LoxP mediated loss of *Recq14* and using CRISPR/Cas9 mediated targeting of different regions of endogenous *Recq14* (using >4 different sgRNAs). This data has been generated in independent cell lines derived from different animals and using multiple independent sgRNAs targeting different regions (as indicated in Fig 4F).
- When we have used acute deletion by Cre-LoxP (see Figure 1E and Extended View Figure 2 and comparing WT1/WT2 to fl/fl 1/fl/fl 2 for EdU incorporation following acute loss of *Recq14* in the present manuscript) or in bone lineage cells both primary osteoblasts (Cre/LoxP) and immortalised (shRNA) Kusa4b10 which are a osteoblastic cell line (see Figure 5H in PMID: 25859855) – in all cases, these cells do not sustain an ability to proliferate following loss or knock-down of *Recq14*. Using an independent floxed mouse allele and *in vivo* analysis, the Wang lab demonstrated a significantly increased cell death and decreased proliferation (as measured by BrdU incorporation) of osteoblastic lineage cells *in vivo* following deletion of *Recq14* (see Fig 4 of PMID:25556649). This same effect of loss of Recq14 – failure to proliferate and cell death - was also reported in chicken DT40 cells (PMID: 21256165).
- In a validated *in vivo* primary osteosarcoma model - relevant as the U2OS cell line used by Padayachy were derived from a human osteosarcoma - we could not establish *Recq14* deficient tumours even with co-deletion of *Tp53* (PMID: 25859855). All tumours retained an undeleted allele of *Recq14* consistent with an inability to achieve a true *Recq14* null state.
- Acute loss of *Recq14* in adult mice causes a rapid and fully penetrant bone marrow failure (PMID: 24960165).

Referee #2:

RecQL4 is a genome maintenance factor involved in replication initiation and DNA damage response, whose mutations are linked to Rothmund-Thompson, Baller-Gerold, and RAPADILINO syndromes. How and why RecQL4 mutations cause these diseases remains poorly understood and has been the focus of many studies over the last ~25 years. The manuscript titled "Minute amounts of helicase-deficient truncated RECQL4 are sufficient for DNA replication." by Buco and Castillo-Tandazo et al. used a very elegant screen-based strategy to identify factors whose loss could rescue the proliferation defect of mouse cells bearing a hypomorphic RecQL4 mutation (truncation) similar to those that have been reported in human patients. Specifically, the RecQL4 truncation used in this study lacks the C-terminal portion of RecQL4 that contains an ATPase domain. Importantly, this mutation also destabilized the truncated RecQL4 protein product - which is also thought to happen with many RecQL4 truncations in human patients.

This manuscript has several interesting findings, but I would like to highlight a few that stood out:

1. The study identified a subunit of an E3 ligase (KLHDC3) whose loss prevents the most severe defects caused by the RecQL4 truncation (including bone marrow failure).
2. The study showed that the loss of KLHDC3 stabilized the truncated RecQL4 protein fragment by preventing its degradation.
3. The study showed that very low levels of truncated RecQL4 (5-7% of that measured for full-length RecQL4 in a control cell line) are sufficient to support relatively normal cell cycle progression.
4. The most impactful finding of this study is that "there are unlikely to be monogenic loss of function rescue alleles for RECQL4 mutation", and all the data presented here strongly supports this conclusion.

Overall the work is rigorous and significantly advances our understanding of RECQL4 mutations, their effect on cell proliferation, and RECQL4's role in human disease. It deserves to be published in EMBO reports, with a few minor edits as indicated below.

Author Response:

We thank the reviewer for their assessment of the work and its implications for understanding RECQL4 functions and its role in human disease.

I have two major questions/criticisms about the interpretation of data. I hope they can be resolved in the final version of the manuscript.

1. The flow cytometry data (Figure 2A-B) indicate that cells with truncated RECQL4 incorporate less EdU, consistent with the idea that they replicate DNA less efficiently (presumably by firing fewer replication origins). The authors write "The mechanism via which the RECQL4 p.E481GfsX48 enables relatively normal DNA replication is still to be fully resolved. Our analysis indicate that this small fragment supports complete DNA replication, but delays completion of DNA synthesis with a shift towards late S phase and even into M phase. This shifted DNA synthesis, evidenced by EdU, presumably depends on MiDAS contributing to the ability of the cells to complete replication". At the same time, the Origin Firing Timing data (Figure 6A-D) did not reveal any major differences between the cells bearing WT RECQL4 and truncated RECQL4. The authors concluded that "This analysis demonstrated that there were, at most, subtle differences between cells with and without the ATP-dependent helicase and C-terminal domains of RECQL4 (Figure 6D)." The replication timing analysis shows which regions of the genome are replicated early and which are replicated late. In other words, replication timing provides a relative quantification of early/late replicating regions. As far as I understand, replication timing analysis does not provide an absolute measure of origin firing efficiency. In other words, it is possible for the RECQL4 truncation to have overall less efficient origin firing while maintaining the same origin firing timing program. Measuring the absolute efficiency of origin firing in cells is technically challenging. One could measure the inter-origin distance (IOD) using fiber analysis -

IOD is expected to increase if origin firing efficiency decreases. However, I do realize that this may be outside the scope of this manuscript.

Author Response:

We thank the reviewer for this considered and thoughtful comment. We have discussed this and agree that alternative possibilities for our results exist and that with our present data we cannot definitively conclude one way or the other. The apparent discrepancy between the replication profiles and the flow cytometry data that the reviewer cited could be due to lower origin firing efficiency genome-wide (presumably affecting late S phase mostly) and/or to increased variation between individual cells, with each cell having a different set of origins firing later or less efficiently or not firing at all. Our bulk replication timing measurement would miss such subtle, stochastic effects. Single cell or molecule approaches could resolve this, however those are also limited in their accuracy or comprehensiveness and could equally well miss any such subtle effects. We have revised the text to be more inclusive of these alternative interpretations and possible mechanisms. See revised discussion.

2. The authors measured the amount of MCM3 (a subunit of the MCM2-7 complex) and observed that there was more chromatin-bound MCM3 in cell lines with truncated RECQL4 compared to control cell lines (Figure 2D). The authors provide the following interpretation of this observation: "The small RECQL4 fragment is associated with an increase in licensed replication origins. This may be because the helicase domain normally acts to limit how many origins are licensed (Terui et al., 2024). As a consequence, more active origins during early S phase with increased dNTP depletion may cause the replication stress that ultimately delays complete replication into M phase. Alternatively, increased origin licensing could also be a compensatory response where more active origins result in shorter replicons making up for slower replication in the absence of the helicase domain."

I have a few objections to this interpretation by Buco et al.: (a) Terui et al 2024 do not provide any data on RECQL4 modulating licensing (they talk about CMG helicase activation). In fact RECQL4 has not been previously implicated in licensing (which involves ORC, CDT1, CDC6 and MCM2-7, for the most recent review, pls see <https://doi.org/10.1042/BST20220917>). (b) the authors invoke more active origins and dNTP depletion in early s-phase with accompanying replication stress, but do not provide any evidence for this claim. This line of reasoning is a bit too speculative. (c) increased origin licensing could be plausible, perhaps due to some adaptation of this cell line. However, another interpretation of the data (perhaps the most parsimonious) is that the cell line with the RECQL4 truncation replicates its DNA slowly (consistent with the data from Buco et al), which causes a relatively slow clearance of pre-replication complexes (MCM2-7) from DNA. The vast majority of MCM2-7 loaded on DNA never initiate replication. Instead, they are cleared from DNA by active replication forks (as shown by Jean Cook's lab and Jiri Lukas' lab previously, see for example <https://www.nature.com/articles/s41467-022-33887-5>). The decrease in MCM3 signal throughout S-phase corresponds to this "clearing" of MCM2-7 from chromatin. Inefficient replication initiation by the RECQL4 truncation can therefore explain the higher levels of MCM3 during early S-phase. As for the higher levels observed in G1-phase - it's difficult to distinguish very early S-phase from G1 on the FACS plots.

Author Response:

We thank the reviewer for this comment and detailed information provided. We have discussed our interpretation considering the reviewers' comments and have moderated the original text to more completely capture the potential uncertainty and alternative models as proposed by the reviewer. The alternative possibilities raised by the reviewer are valid and we hope the revised text can capture this coherently. See revised discussion.

I have a few minor comments about how the authors interpret some of their observations, as well as data from other studies. All of these can be addressed by small edits to the text and more clear explanation in some cases.

1. The authors state that "This result demonstrated that the ATP-dependent helicase activity of RECQL4 was not essential in vivo for DNA replication or murine homeostasis, consistent with analysis in human cell line models (Padayachy et al, 2024)." I think it's worth explicitly mentioning in the revised manuscript that Padayachy et al 2024 showed that knocking out RECQL4 entirely in a human cell line still allowed those cells to proliferate relatively normally. That study suggests that in some cell lines in cell culture conditions, RECQL4 is not essential for viability. Another study which was recently posted on BiorXiv (<https://www.biorxiv.org/content/10.1101/2025.08.05.668837v1>) reports that RECQL4 is partially redundant with MCM10 when it comes to promoting origin firing. This explains why in some circumstances cells can proliferate without RECQL4 (Padayachy et al 2024). This also explains why depleting RECQL4 from Xenopus egg extract does not fully inhibit DNA replication (as reported in Sangrithi et al 2005, Matsuno et al, 2006, Terui et al, 2024).

Author Response:

We thank the reviewer for noting this. As noted in our response to Reviewer 1 we have amended the text and how we refer to the Padayachy et al., work reflecting that they propose to have generated RECQL4 deficient cells as well as helicase defective. We have assessed our data for MCM10 sgRNA in our suppressor and synthetic lethal screens. We see a more pronounced depletion in the synthetic lethal screen consistent with the reviewer's suggestion.

We have also now directly tested Mcm10 sgRNA in the synthetic lethal context (Extended Data Figure EV3C-3E). This data collectively is indicative of synthetic lethality and partial redundancy of RECQL4 and MCM10. These experiments were difficult as the MCM10 targeted cells were very rapidly lost and required greater expansion to get sufficient sample for genomic DNA assessment compared to the Cyclin F validation. We have also included the individual data for each of the sgRNA in the synthetic lethal screen.

2. Interestingly, Padayachy et al found that RECQL4-KO (knock-out) and RECQL4-HD (helicase dead) cell lines exhibited similar origin firing profiles (i.e. they have similar replication timing profiles). However, Padayachy et al reported that the RECQL4-KO cells exhibited higher replication fork rates which are consistent with decreased origin firing efficiency. This echoes my major criticism above about interpreting replication timing data.

Author Response:

The U2OS cell lines reported and assessed by Padayachy display a different phenotype from other U2OS RECQL4 deficient cell lines that have been reported (*Nucleic Acids Res.* 2020 Jul 9;48(12):6530-6546. doi: 10.1093/nar/gkaa392). The Bohr group have reported significantly reduced proliferation and viability when targeting RECQL4 in U2OS cell as cited above.

We acknowledge that alternative techniques to genome-wide replication timing analysis are required to measure absolute fork rate progression and origin firing efficiency, and we have amended the text to reflect this. However, Padayachy et al. only observed higher replication fork rates in the RECQL4-KO cells with their EdUseq-HU technique. The DNA combing assay performed using the same synchronisation conditions showed no difference in fork rate progression. They postulated that: "Thus, in this case, the EdU-seq experiment may be indicating changes in fork recovery after HU removal, rather than changes in fork progression rates.", adding another layer of complexity in the interpretation of replication data obtained by different techniques.

Please see response to Reviewer 1 regarding this publication.

3. The authors repeatedly state that the ATP-dependent helicase activity of RECQL4 is not essential for DNA replication in mammals. They contrast this with a recent Xenopus study (Terui et al, 2024), which reported that RECQL4's ATPase activity does play a role in replication initiation. In fact, a few older Xenopus studies (Sangrithi et al, 2005 and Matsuno et al, 2006) also reported that RECQL4's ATPase plays a role in replication initiation. All aforementioned Xenopus studies do report that ATPase-deficient RECQL4 mutants can partially rescue RECQL4 depletions. So everyone agrees that RECQL4 can support origin firing without its ATPase function (i.e. the ATPase is not strictly essential for replication). However, the Xenopus studies

do provide evidence that RECQL4's ATPase activity further boosts RECQL4's ability to initiate replication.

Author Response:

We have amended our description to more fully reflect the knowledge area including from studies outside of mammals. Our interpretation reflected our previous work with an *in vivo* mouse model where the ATP-binding by endogenous Recq14 was abolished by the K525A mutation (PMID: 31276497). We note that *Xenopus* are not mammals.

4. In the introduction, the authors write "Although RECQL4 was long considered the mammalian homologue of the essential yeast DNA replication factor Sld2, it is now clear that a structurally unrelated protein DONSON has evolved to carry out the Sld2-like function during mammalian DNA replication (Cvetkovic et al, 2023; Evrin et al, 2023; Hashimoto et al, 2023; Kingsley et al, 2023; Lim et al, 2023). As a result, the role and function of RECQL4 in normal DNA replication remains a major knowledge gap in our understanding of this fundamental cellular process." The statement that DONSON is the functional counterpart of SLD2 in mammals is just a speculation entertained in some studies which reported that DONSON is a novel replication initiation factor in metazoa. However, this is not yet widely accepted and is an area of active investigation. I suggest changing "it is now clear" to "it has been proposed" or "it has been speculated".

Author Response:

We thank the reviewer for this comment and have moderated the text as suggested.

Referee #3:

The manuscript by Buco et al. investigates the role of a truncated RECQL4 product in supporting cellular proliferation both in cellulo and in vivo in mouse models. The authors deleted Recql4 exons 9-10 via Cre-Lox recombination, which normally results in bone marrow failure and cell death. Using forward genetic CRISPR/Cas9 screens, they identified KLHDC3 as the sole suppressor of this phenotype. Mechanistic studies revealed that recombination generated a truncated RECQL4 fragment (aa 1-480) fused to a 50 aa C-terminal tail (frameshifted), producing a neo-degron recognised by KLHDC3 and targeted for proteasomal degradation. In the absence of KLHDC3, residual truncated RECQL4 stabilises, maintaining sufficient function to support cell proliferation-suggesting that the ATPase and helicase domains are dispensable for RECQL4's essential role.

This is an important and timely study, as it helps reconcile contradictory findings from human, mouse, and Xenopus studies regarding RECQL4's role in DNA replication. Nonetheless, several issues need to be addressed before the manuscript is suitable for publication.

Major Comments:

1) The truncation arising from the Cre/LoxP floxed allele should be characterised more thoroughly. Given the suggested importance of the RECQL4 N-terminal domain, this mutant could serve as an informative control in downstream assays. Alternatively, the authors should clarify why it was not explored further (e.g. sensitivity to replication stress, DNA fibre assays for fork restart or origin firing).

Author Response:

We thank the reviewer for this comment. We have not completed experiments such as those described by the reviewer because the truncation arising from the murine loxP flanked allele is specific and unique to this mouse model and not present or replicated in the human *RECQL4* patient mutation spectrum. We demonstrate that cells only expressing the truncation arising from recombination of the floxed allele do not sustain proliferation or viability (Fig 1E-G; Extended View 1C) and no longer incorporate EdU (Extended View Figure 2A). We cannot maintain cells expressing the truncated fragment without concurrent deletion of KLHDC3 which stabilizes the product (Figure 5).

Our greatest interest from these studies remains with understanding why and how *RECQL4* variants/mutations given rise to a cancer predisposition syndrome relevant to human. Whilst an interesting question, we believe that the more significant findings from our work, as noted by Reviewer 2, do not require this level of characterisation of the murine specific truncated allele.

2) The R347X mutant is included in the initial screen but not utilised in subsequent experiments. A concise summary of its known properties in the main text (beyond citation) would improve readability and contextualise the results.

Author Response:

We have included further details of the allele in the text. We have also made clear that this allele has been extensively characterised in our prior studies (PMID: 31276497 and PMID: 33361189).

3) Indicators of statistical testing (p-values, error bars, replicates) are reported inconsistently across figures. Standardising presentation is essential to support key conclusions, as some claims currently lack sufficient data.

Author Response:

We thank the reviewer for highlighting this and we have gone through the revision to ensure consistency and inclusion of this information.

4) Several panels (e.g. Fig. 1) are shown as overlay histograms without numerical summaries. Quantitative plots with error bars and replicate numbers should be included for reproducibility.

Author Response:

We do not have any histograms in Figure 1.

5) Ensure that all supplementary figures and datasets are explicitly cited in the main text, to improve accessibility.

Author Response:

We thank the reviewer for highlighting this and have made sure to do this in the revised text.

6) The authors should comment on why no other CRL E3 ligase components or proteasome regulators emerged from the suppressor screen.

Author Response:

The suppressor screen we have conducted was to identify loss-of-function alleles that were significantly enriched in the $\Delta/R347X$ cells. This was based on rescue of proliferation and viability to allow outgrowth of the rescued cells (and thus enrichment of sgRNA copy number) which is a stringent requirement and one we have used successfully in other settings and genetic backgrounds (PMID: 39576872; PMID: 34464972). In the screen we completed here we did not identify other CRL E3 ligase components or proteasome regulators as the reviewer has identified. We speculate that this reflects the highly sequence specific nature of the DesCEND pathway and, in this case, its substrate recognition receptor KLHDC3. Deletion of other CRL E3 ligase components or proteasome regulators may change the proteome more broadly but less specifically than deletion of KLHDC3. The overall effect may therefore not be strong enough to rescue the lethality of the $\Delta/R347X$ cells.

As with all screens irrespective of technology (siRNA, shRNA, CRISPR), it is not possible to draw a conclusion about the majority of genes as they do not enrich or deplete to a statistically applied cut off when the data is assessed. For the genes that do not enrich or deplete no conclusion, either positive or negative, can be drawn about the function of these genes from the screen. The only valid conclusion is that we cannot draw further understanding about this gene (usually the vast majority) from the screen as it was performed.

Related specifically to the screens we are reporting here with CRISPR/Cas9 mediated loss-of-function: The failure to enrich or deplete a given candidate can be for many reasons unrelated to biology, for example sub-optimal sgRNA design for an individual gene so no loss of function is ever achieved, very low copy number of sgRNAs in the pooled library so they never are considered due to insufficient sgRNA at time 0, the specific characteristics of the cell type used and screen design, heterozygous mutations occurring at higher frequency than homozygous deletion etc. These caveats cannot be addressed at the individual gene level when applying a whole genome library and are one of the considerations when using these libraries and interpreting the outcomes.

7) Do HEK293T cells recapitulate the murine phenotypes upon RECQL4 depletion? Extending the findings to human cells would strengthen the conclusions.

Author Response:

We have repeatedly tried to generate and test this biology in HEK293T cells but we could not generate stable *RECQL4*^{-/-} cell lines, even with concurrent loss of KLHDC3. We can successfully establish KLHDC3 knockouts in these cells so our technique is able to generate and isolate deficient cells and clones. We therefore cannot directly answer this as we have not been able to recover RECQL4 deficient HEK293T cells.

We believe the failure to establish viable RECQL4 deficient human cells is consistent with our interpretation that it is the specific C-terminal sequence generated by the deletion of the floxed allele that generates the specific targeting sequence for KLHDC3 in mouse. Human RECQL4 does not have the same predicted sequence motif arising after deletion of the homologous region so loss of KLHDC3 would not be expected to be advantageous in this setting.

This is further discussed and expanded on in response to Reviewer 1, but we have never been able to establish a *bona fide* *Recql4/RECQL4* deficient viable sample in either mouse or human cells to be able to test this.

- 8) Both text and figures require simplification and consistency. Examples include:
- consistent labelling of alleles and truncations (Δ , R347X, truncation);
 - legends that match figure elements (e.g. missing MG132 annotation; different symbols for V5-KLHDC3 variants across figures; unexplained yellow asterisk in Fig. 5A);
 - poor visibility of Fig. 4E;
 - overly complex graphs (e.g. Fig. 4B);
 - unclear flow cytometry plots (Figs. 2C-D, colours not matched to cell-cycle phases);
 - insufficient detail in Fig. 5D, which requires a close-up.

Author Response:

We have reviewed all of these aspects in the revision and have unified the presentation.

Minor Comments:

- 1) State the number of biological replicates for key assays (e.g. replication timing WGS, GPS stability assays).

Author Response:

We have added these specific details as requested.

- 2) Expand the description of TIGER analysis to assist readers less familiar with replication timing approaches.

Author Response:

We thank the reviewer for identifying this and have expanded the text to allow readers more understanding of the method used.

- 3) The discussion could briefly address how these findings integrate with recent studies implicating RECQL4 in chromosome segregation, where similar truncations were analysed.

Author Response:

We think that the reviewer is referring to the recent RECQ4-MUS81 paper (<https://doi.org/10.1038/s41467-025-56518-1>)

In this paper the authors demonstrate that: "RECQ4 physically interacts with MUS81, targeting it to specific DNA substrates and enhancing its endonuclease activity. Loss of this interaction, results in significant chromosomal segregation defects, including the accumulation of micronuclei, anaphase bridges, and ultrafine bridges (UFBs)."

They also observe that: "a mutation associated with Rothmund-Thomson syndrome, which produces a truncated RECQ4 unable to interact with MUS81, recapitulates these chromosome instability phenotypes."

We have not specifically looked for chromosomal segregation defects and have only performed one metaphase spread analysis. In that experiment we did not find any significant changes in the RECQL4 truncation mutant. Since this mutant contains the MUS81 interaction sequence (aa317-392 in mouse corresponding with aa322-400 in human), we do not expect this mutant to be defective in the RECQL4-MUS81 function described in this paper.

- 4) Fig. S2a likely mislabels fl/+ as Δ /+.

Author Response:

We have reviewed this labelling and in the original Figure S2a (now in Appendix) the labelling was correct as prior to tamoxifen treatment (annotated as -Tam) the cells are fl/+, after tamoxifen treatment (+Tam) the cells delete the floxed allele and become Δ /+. This labelling has been consistently used throughout.

- 5) Ensure consistent terminology for "Fig." vs "Figure" (e.g. p. 6).

Author Response:

We thank the reviewer for identifying this and we have reviewed the revised document to ensure consistency.

6) The suggestion to apply MIDAS should be either experimentally supported or toned down in the text.

Author Response:

We thank the reviewer for this comment and have revised the text to moderate the invocation of MIDAS.

7) A schematic summarising all RECQL4 truncations used in this study would greatly aid clarity and unify nomenclature across the manuscript.

Author Response:

For completeness we have included a figure in the Extended View Figure 1C summarising the *Recq14* alleles and also direct the reviewer/reader to our previous work (eg Fig 6A of Castillo-Tandazo et al., *PLoS Genetics* 2019; <https://doi.org/10.1371/journal.pgen.1008266>).

In summary, this study addresses an important question in DNA replication biology and provides mechanistic insight into RECQL4 function. However, additional clarification, consistent data presentation, and figure/text improvements are needed to make the manuscript fully convincing and accessible to readers.

Cross-comment from referee 2:

We know that RECQL4's ATPase is not essential for viability because human RTS, BGS, and RAPADILINO patients who lack the ATPase domain are alive, but clearly, their genome maintenance is compromised. In my review of Buco et al. I suggested that Buco et al address these discrepancies in their discussion, rather than having to do a ton of extra experiments.

We now appreciate that under certain circumstances (for example specific cell types) cells can tolerate very low levels of RecQL4 (and RecQL4 mutants) without arresting the cell cycle. In this context, the work from Buco et al. provides valuable insights in a physiological context.

Regarding the comment from another reviewer about the "elephant in the room" with the paper from the Halazonetis lab where they claim that RECQL4-deficient cells doesn't cause an effect on origin firing. There are a few caveats with interpreting the paper from the Halazonetis lab:

1. Certain cell types are able to tolerate very low (and perhaps nil) levels of RecQL4 because MCM10 is partially functionally redundant with RecQL4 during origin firing.
2. CRISPR knock-outs can be tricky to validate, sometimes there are alternative transcripts being made that retain some function. The Halazonetis paper did not exhaustively validate their RECQL4-KO cell line. In addition, commercial antibodies used to demonstrate the absence of RECQL4 products in the KO line may not be able to detect trace amounts of protein (or protein fragments) still present in that cell line. So in that sense, we shouldn't put the burden of proof on Buco et al to validate the cell line from the Halazonetis lab.

Author Response:

We have reviewed the Halazonetis paper in detail to supplement the comments raised based on the data presented in the Halazonetis groups publication:

Clone fsN1 is a compound heterozygote yielding frameshift mutations in the targeted exon. Clone fsN2 has a homozygous insertion of a TAACCC repeat sequence of unknown (telomere?) origin predicted to result in an early stop codon. Clones fsH1 and fsH2 (which has the same repeat insertion in one allele) are compound heterozygotes with frameshift mutations after L514. With the exception of the abnormal fsN2 clone, all clones are predicted to produce truncated transcripts (see Padayachy et al. Supplementary Figure 2).

The western blot using the NBP2-47310 antibody (raised against aa 190-301) shows multiple bands for the full-length RECQL4 and similar faint bands for the fsH1 and fsN1 KO clones. The truncated overexpressed RECQL4 is only detected as degradation products of what should be 34.5 and 40.2kDa products (see Padayachy et al. Supplementary Figure 3a)

3. The Halazonetis lab did not do a thorough job in describing the origin firing defect caused by the KO of RecQL4 - in part because accurately measuring origin firing efficiency in cells is very hard. This is a limitation of many cell-based studies.

I agree with you that Buco et al. should address the discrepancies between different studies in their discussion section. . It seems likely that some of these discrepancies are due to the partial functional redundancy between RecQL4 and MCM10.

Author Response:

The data from the synthetic lethal screen and when tested individually support the conclusion for partial redundancy between RECQL4 and MCM10 (Extended Data Figure EV3C-3E).

I don't think it will help much if Buco et al repeated experiments with the cell line generated by the Halazonetis lab because they will confront the same exact technical limitations (the difficulty in fully validating a CRISPR KO line, the limited sensitivity of commercial antibodies, the inability to accurately measure origin firing efficiency in vivo).

One prediction based on recent work in different labs is that RECQL4-mutant cells rely heavily on MCM10 to initiate replication. Therefore, they should be sensitive to knocking down MCM10. I am not sure if siRNA or sgRNA can lower MCM10 levels sufficiently to see a clear defect in cell proliferation - that is one caveat of this approach... Perhaps that could be a direction that Buco et al can explore in a future study. Perhaps they could give it a try while revising their manuscript?

Our understanding of RECQL4 and its mutations continues to evolve as we better understand the molecular mechanism of replication initiation, and studies like Buco et al provide important clues and context for elucidating the basis of RECQL4-related diseases.

Dear Dr. Walkley,

Thank you for the submission of your revised manuscript. We have now received the enclosed reports from the referees and I am happy to say that all support its publication now. Only a few editorial requests will need to be addressed before we can proceed with the official acceptance of your manuscript:

- Please reduce the number of keywords to 5.
- The conflict of interest subheading needs to be corrected to Disclosure and Competing Interests Statement.
- The author credits need to be removed from the ms file. All credits need to be entered during online ms submission.
- Please send us with your final ms a completed author checklist, which you can download from our author guidelines: <https://link.springer.com/journal/44319/submission-guidelines>
The completed author checklist will also be part of the transparent peer-review process file.
- You entered some of the FUNDING INFO into the Comments box online, this needs to be corrected. The funding info provided in the Comments box will not be displayed on the published ms since our publisher retrieves funding from the separate funding entries. Also, the funding info in the ms needs to be part of the Acknowledgments, the separate "Funding" section heading needs to be deleted.
- Figure S6, Figure S8 and Table S1 are not correct callouts, please correct. A callout is missing for Appendix Figure S4. I am not sure whether the uploaded Datasets are the same as the ones mentioned in the Appendix file. The Appendix should only contain figures and tables, and the legends need to follow their respective figures, and should not contain any Datasets. All Datasets need to be uploaded as individual Dataset files and should be called Dataset EV1, etc. This needs to be correct in the files themselves and in the ms.
- The Reagent Table should only be uploaded as an individual file. The Reagents info in the Appendix file needs to be moved to the Reagents and Tools table.
- Please double check that all source data are uploaded, it seems that the SD for Fig 5D still needs to be uploaded.
- The manuscript sections should be in the following order: Title page - Abstract & Keywords - Introduction - Results - Discussion - Methods - Data Availability - Acknowledgments - Disclosure Statement & Competing Interests - References - Figure Legends - (Main Tables with legends if applicable) - Expanded View Figure Legends.
- ORCIDs and the Number of figures should be removed from the title page
- The nomenclature of EV figure legends and some of the separate EV figures is not correct: needs to be Figure EV1, etc. instead of Expanded View Figure EV1, etc.
- Our image integrity officer noted that the Appendix file figures need to be uploaded at higher resolution. They are currently pixelated under analysis.
- The specific URLs for GSE273174, GSE273006, GSE272599 datasets are not provided in the data availability statement, please add.
- Please note that the exact p values are not provided in the legends of figures 2C, D. Please add exact values as reasonable.

I would like to suggest some minor changes to the abstract. Please let me know whether you agree with this :

RECQL4, a RecQ family helicase, is essential for DNA replication and genome stability. Mutations in RECQL4 cause severe human disorders yet we do not fully understand its functions, particularly regarding ATP-dependent helicase activity. To understand RECQL4's functions further, we performed a genome-wide forward genetic screen using a murine model harbouring patient-like RECQL4 mutations [OK?]. We identify KLHDC3, a substrate-binding subunit of the Cullin-RING ligase E3 complex, loss as the most significant rescue allele. KLHDC3 loss restores proliferation and replication in RECQL4-deficient cells by stabilizing trace levels of a truncated RECQL4 fragment containing the N-terminal 480 amino acids, lacking the helicase and C-terminal regions. This RECQL4 fragment forms after Cre-mediated recombination of the *Recql4*^{fl} allele and contains a neo-degron sequence specific for KLHDC3. Although this mechanism does not apply to human mutations, it demonstrates that minimal RECQL4 levels, without any ATPase domain/activity, are sufficient to support DNA replication. This demonstrates that RECQL4 is an essential and non-redundant regulator of DNA replication and cell viability and that this activity does not require

its ATP-dependent helicase activity.

EMBO press papers are accompanied online by A) a short (1-2 sentences) summary of the findings and their significance, B) 2-3 bullet points highlighting key results and C) a synopsis image that is exactly 550 pixels wide and 200-600 pixels high (the height is variable). The synopsis image should provide a sketch of the major findings, like a graphical abstract. Please note that text needs to be readable at the final size. Please send us this information along with the final manuscript.

Kind regards,
Esther

Referee #1:

I am satisfied with the responses made by the authors and I recommend that the manuscript be accepted for publication.

Referee #2:

The authors did a good job addressing my criticisms (as rev #2) and from my reading of their response to rev #1 - they handled that response well too. My initial criticism was fairly mild and was not about the data, but about the interpretation, discussion, and reconciliation with previous reports. I am happy with the changes made by the authors and I have no further criticism.

I was also happy to see the new MCM10 data.

Referee #3:

The authors have addressed both the major and minor concerns raised in the initial review and have made substantial revisions that significantly improve the overall quality of the manuscript. The study is now better aligned with recent findings regarding the role of RECQ4 in DNA replication. The authors have also provided clearer explanations of their results and have removed any over-interpretation of the implications of their data. As a result, the consistency and narrative flow of the manuscript have markedly improved.

Importantly, the authors have included new experiments confirming partial redundancy between RECQ4 and MCM10. This addition substantially strengthens the manuscript. Furthermore, the revised version now presents a clearer and more effective analysis of the key replication phenotypes observed in the presence and absence of RECQ4 and/or KLHDC3. The distinctions between the allelic variants used are now better defined, and the rationale for the inclusion or exclusion of specific variants is clearly summarised.

Interestingly, the authors note in their point-by-point response that it was not possible to generate HEK293T cells lacking RECQ4. Although this observation is not included in the manuscript itself, it is noteworthy in light of other recent reports in the field.

Overall, the quality of the manuscript has improved substantially in all respects, and in its current form it appears suitable for publication.

EMBOR-2025-62414V2 Decision Letter (sent 22 Jan 2026)

Thank you for the submission of your revised manuscript. We have now received the enclosed reports from the referees and I am happy to say that all support its publication now. Only a few editorial requests will need to be addressed before we can proceed with the official acceptance of your manuscript:

- Please reduce the number of keywords to 5.

Author Response: reduced to 5

- The conflict of interest subheading needs to be corrected to Disclosure and Competing Interests Statement.

Author Response: changed

- The author credits need to be removed from the ms file. All credits need to be entered during online ms submission.

Author Response: removed

- Please send us with your final ms a completed author checklist, which you can download from our author guidelines: <https://link.springer.com/journal/44319/submission-guidelines> The completed author checklist will also be part of the transparent peer-review process file.

Author Response: completed

- You entered some of the FUNDING INFO into the Comments box online, this needs to be corrected. The funding info provided in the Comments box will not be displayed on the published ms since our publisher retrieves funding from the separate funding entries. Also, the funding info in the ms needs to be part of the Acknowledgments, the separate "Funding" section heading needs to be deleted.

Author Response: The "funding" section heading has been removed and the funding in the ms merged with the acknowledgements section. The text in the comments box online has been removed.

- Figure S6, Figure S8 and Table S1 are not correct callouts, please correct. A callout is missing for Appendix Figure S4. I am not sure whether the uploaded Datasets are the same as the ones mentioned in the Appendix file. The Appendix should only contain figures and tables, and the legends need to follow their respective figures, and should not contain any Datasets. All Datasets need to be uploaded as individual Dataset files and should be called Dataset EV1, etc. This needs to be correct in the files themselves and in the ms.

Author Response: We have doubled checked all Appendix and EV data callouts and removed the reference to the Datasets in the Appendix.

- The Reagent Table should only be uploaded as an individual file. The Reagents info in the Appendix file needs to be moved to the Reagents and Tools table.

Author Response: removed

- Please double check that all source data are uploaded, it seems that the SD for Fig 5D still needs to be uploaded.

Author Response: this was an oversight/error on the authors part when uploading the source data. The file has been included in the revision upload, and it was a labelling error in the revision 1 submission of the source data.

- The manuscript sections should be in the following order: Title page - Abstract & Keywords - Introduction - Results - Discussion - Methods -Data Availability - Acknowledgments - Disclosure Statement & Competing Interests - References - Figure Legends - (Main Tables with legends if applicable) - Expanded View Figure Legends.

Author Response: manuscript reordered to order required

- ORCID IDs and the Number of figures should be removed from the title page

Author Response: removed

- The nomenclature of EV figure legends and some of the separate EV figures is not correct: needs to be Figure EV1, etc. instead of Expanded View Figure EV1, etc.

Author Response: corrected

- Our image integrity officer noted that the Appendix figure files need to be uploaded at higher resolution. They are currently pixelated under analysis.

Author Response: We have re-exported the Appendix figures as 400dpi PDF files and checked them. As far as we can ascertain the images appear clear on screen but please advise if they remain pixelated

- The specific URLs for GSE273174, GSE273006, GSE272599 datasets are not provided in the data availability statement, please add.

Author Response: URLs added to the data availability statement

- Please note that the exact p values are not provided in the legends of figures 2C, D. Please add exact values as reasonable.

Author Response: P values have been added to the figure

I would like to suggest some minor changes to the abstract. Please let me know whether you agree with this:

RECQL4, a RecQ family helicase, is essential for DNA replication and genome stability. Mutations in RECQL4 cause severe human disorders yet we do not fully understand its functions, particularly regarding ATP-dependent helicase activity. To understand RECQL4's functions further, we performed a genome-wide forward genetic screen using a murine model harbouring patient-like RECQL4 mutations [OK?]. We identify KLHDC3, a substrate-binding subunit of the Cullin-RING ligase E3 complex, loss as the most significant rescue allele. KLHDC3 loss restores proliferation and replication in RECQL4-deficient cells by stabilizing trace levels of a truncated RECQL4 fragment containing the N-terminal 480 amino acids, lacking the helicase and C-terminal regions. This RECQL4 fragment forms after Cre-mediated recombination of the Recql4^{fl} allele and contains a neo-degron sequence specific for KLHDC3. Although this mechanism does not apply to human mutations, it demonstrates that minimal RECQL4 levels, without any ATPase domain/activity, are sufficient to support DNA replication. This demonstrates that RECQL4 is an essential and non-redundant regulator of DNA replication and cell viability and that this activity does not require its ATP-dependent helicase activity.

Author Response: agreed and accepted in revised manuscript

EMBO press papers are accompanied online by A) a short (1-2 sentences) summary of the findings and their significance, B) 2-3 bullet points highlighting key results and C) a synopsis image that is exactly 550 pixels wide and 200-600 pixels high (the height is variable). The synopsis image should provide a sketch of the major findings, like a graphical abstract. Please note that text needs to be readable at the final size. Please send us this information along with the final manuscript.

Author Response: synopsis text and figure included in revised submission.

Dr. Carl Walkley
Hudson Institute of Medical Research
Centre for Innate Immunity and Infectious Disease
27-31 Wright St
Clayton, Victoria 3168
Australia

Dear Dr. Walkley,

I am very pleased to accept your manuscript for publication in the next available issue of EMBO reports. Thank you for your contribution to our journal.

You may qualify for financial assistance for your publication charges - either via a Springer Nature fully open access agreement or an EMBO initiative. Check your eligibility: <https://link.springer.com/journal/44319/how-to-publish-with-us>

Kind regards,
Esther

>>> Please note that it is EMBO Reports policy for the transcript of the editorial process (containing referee reports and your response letter) to be published as an online supplement to each paper. If you do NOT want this, you will need to inform the Editorial Office via email immediately. More information is available here: <https://link.springer.com/partners/embo-press/editorial-policies#Peer%20review>